# Human T$_H$17 cells engage gasdermin E pores to release IL-1α on NLRP3 inflammasome activation

Ying-Yin Chao[1,2], Alisa Puhach [1], David Frieser[2], Mahima Arunkumar [1], Laurens Lehner [1], Thomas Seeholzer [3], Albert Garcia-Lopez [4], Marlot van der Wal[5], Silvia Fibi-Smetana[6], Axel Dietschmann[7], Thomas Sommermann[1], Tamara Ćiković[8], Leila Taher [6], Mark S. Gresnigt[7], Sebastiaan J. Vastert[5], Femke van Wijk [5], Gianni Panagiotou[4], Daniel Krappmann [3], Olaf Groß[8] & Christina E. Zielinski [1,2,9,10,11] ✉

It has been shown that innate immune responses can adopt adaptive properties such as memory. Whether T cells utilize innate immune signaling pathways to diversify their repertoire of effector functions is unknown. Gasdermin E (GSDME) is a membrane pore-forming molecule that has been shown to execute pyroptotic cell death and thus to serve as a potential cancer checkpoint. In the present study, we show that human T cells express GSDME and, surprisingly, that this expression is associated with durable viability and repurposed for the release of the alarmin interleukin (IL)-1α. This property was restricted to a subset of human helper type 17 T cells with specificity for *Candida albicans* and regulated by a T cell-intrinsic NLRP3 inflammasome, and its engagement of a proteolytic cascade of successive caspase-8, caspase-3 and GSDME cleavage after T cell receptor stimulation and calcium-licensed calpain maturation of the pro-IL-1α form. Our results indicate that GSDME pore formation in T cells is a mechanism of unconventional cytokine release. This finding diversifies our understanding of the functional repertoire and mechanistic equipment of T cells and has implications for antifungal immunity.

Helper T cells (T$_H$ cells) are important enactors of antigen-specific effector responses via their secretion of distinct cytokines. Helper type 17 T cells (T$_H$17 cells), in particular, are recognized for their antifungal functions through the secretion of their signature cytokine IL-17A, which is regulated by the transcription factor RAR-related orphan receptor (ROR)-γt[1]. They are also the main culprits in the pathogenesis of autoimmune diseases[2]. T$_H$17 cells have previously been recognized to display functional heterogeneity[3]. Pro- or anti-inflammatory functions are exerted via the differential coexpression of IL-17 with either interferon (IFN)-γ or IL-10, respectively[4–7]. Overall, this has shaped the concept of a T$_H$17 cell dualism and has stimulated investigation into the signals and molecular targets that control the dichotomy between the two functional T$_H$17 cell outcomes for therapeutic applications[3,4,8,9]. However, a deep understanding of the identity and mechanistic basis of pathogenic versus immunoregulatory T$_H$17 cell fates remains elusive. Additional, yet-to-be-found effector mechanisms that go beyond IL-17 production might also operate in T$_H$17 cells with antifungal or antibacterial target specificities.

IL-1 cytokines, of which IL-1α and IL-1β represent the most prominent members, exert profound inflammatory effects. On release from antigen-presenting cells (APCs), they not only induce rapid innate inflammatory responses, but also orchestrate adaptive immunity by

promoting $T_H17$ cell polarization and T cell pathogenicity on binding to their shared IL-1R1 receptor[4,10,11]. IL-1-independent $T_H17$ cell priming, which has also been previously described, results in the production of anti-inflammatory $T_H17$ cells[4]. IL-1 from innate cellular sources therefore serves as a switch factor for the dichotomy of pro- versus anti-inflammatory $T_H17$ cell fates. Unlike most other cytokines, IL-1 cytokines lack a signal peptide and are therefore secreted by an unconventional, endoplasmic reticulum (ER)–Golgi-independent mechanism. Pro-IL-1β requires enzymatic cleavage before release into the extracellular space and engagement of its receptor. The NLRP3 inflammasome is a multimeric cytosolic protein complex that assembles on microbial infection and cellular damage and recruits caspase-1 for subsequent pro-IL-1β cleavage[12]. IL-1β exit also requires caspase-1-mediated gasdermin D (GSDMD) cleavage and pore formation in a process called pyroptosis, an inflammatory form of cell death[13,14]. IL-1α, on the other hand, is thought to be processed independently of the NLRP3 inflammasome through regulatory checkpoints that are still poorly understood[10]. Despite these completely distinct pathways for the maturation and release of IL-1β and IL-1α, both cytokines are jointly produced by cells of the innate immune system, pointing to the existence of yet-to-be-identified co-regulatory routes.

In the present study, we show that a subset of human $T_H17$ cells engages an NLRP3-dependent signaling cascade to induce membrane pore formation by GSDME, which serves the autocrine release of pro-inflammatory IL-1α. This finding reveals an unconventional mode of cytokine secretion by human T cells and thus diversifies the T cell functional and mechanistic repertoire.

## Results

### Production of IL-1α is a characteristic of human $T_H$ cells

To investigate the heterogeneity of the human $T_H17$ cell subset and to reveal distinct functions and their molecular control, we performed single-cell RNA-sequencing (scRNA-seq) of activated human $T_H17$ cells, which had been isolated ex vivo from peripheral blood according to their unique expression of chemokine receptor surface markers[15]. Exploratory analysis by uniform manifold approximation and projection (UMAP) and Leiden clustering of all $T_H17$ cells identified six individual clusters (Fig. 1a). A distinct and rare (6%) population of *IL1A*-expressing $T_H17$ cells was selectively enriched in cluster 1 (Fig. 1a,b). Comparison of all genes in cluster 1 with all other clusters revealed *IL1A* to be significantly upregulated (Supplementary Table 1). This was unexpected given that IL-1α is not considered to belong to the canonical effector cytokine repertoire of T cells, but instead represents an innate danger signal[16]. *IL1A* was not, however, among the top differentially expressed genes (DEGs) in cluster 1, which necessitated a deeper search strategy to unmask its significant upregulation in a subpopulation of $T_H17$ cells (Supplementary Fig. 1). At the protein level, $T_H17$ cell clones also segregated into distinct IL-1α$^+$ and IL-1α$^-$ T cell clones, thus supporting the heterogeneity of IL-1α protein expression at the single-cell level within the $T_H17$ cell population (Fig. 1c).

To compare *IL1A* expression by $T_H17$ cells with that in other immune cell types, particularly previously reported bona fide producers of IL-1α, we interrogated multiple public scRNA-seq datasets of human peripheral blood mononuclear cells (PBMCs). Surprisingly, this did not reveal any *IL1A* expression in various immune cell types of resting PBMCs, including T cells, B cells, natural killer (NK) cells, NKT cells, monocytes and dendritic cells (Supplementary Fig. 2a,b). Even monocytes did not display any *IL1A* expression at the single-cell level, unless a specific *IL1A*-inducing stimulus, specifically lipopolysaccharide (LPS), was applied to these cells (Supplementary Fig. 2c,d), which unmasked their *IL1A*-producing ability and the association of *IL1A* expression with an ongoing inflammatory innate immune response (Supplementary Fig. 2e). It is interesting that a single-cell transcriptomic comparison of the DEGs between *IL1A*$^+$ and *IL1A*$^-$ cells from $T_H17$ cells versus monocytes demonstrated hardly any overlap in gene coexpression (*IL1A* and *CCL3*),

which was highly suggestive of a different mode of *IL1A* regulation in T cells versus monocytes (Fig. 1d). Taken together, these results reveal the existence of a distinct subpopulation of IL-1α-expressing cells within the $T_H17$ cell subset.

### The IL-1α-producing subset of $T_H17$ cells is proinflammatory

To explore the physiological relevance of *IL1A* expression in human $T_H17$ cells, we performed an unbiased transcriptomic comparison of *IL1A*$^+$ and *IL1A*$^-$ $T_H17$ cells after scRNA-seq. Gene set enrichment analysis (GSEA) for genes coexpressed with *IL1A* in $T_H17$ cells, as well as enrichment analysis using DEGs, revealed a striking association of *IL1A* with T cell activation and proliferation after an unbiased interrogation of all available gene ontology (GO) terms (Fig. 2a and Supplementary Fig. 3). This finding challenged the previously assigned role of IL-1α in senescence and cell death in the new context of T cells[17]. The enrichment analysis also revealed a strong overrepresentation for several GO terms related to 'inflammation', suggesting that *IL1A* expression by $T_H17$ cells contributed to a pathogenic T cell identity with roles in inflammatory diseases (Fig. 2a). This idea was supported by the upregulation of genes annotated with the GO term 'cellular response to interleukin-1', considering the previously reported proinflammatory switch effect of IL-1β on the overall $T_H17$ cell functionality[4] and the suppressive effect of autocrine IL-1α on IL-10 expression (Extended Data Fig. 1a–c). A direct comparison of *IL1A*$^+$ versus *IL1A*$^-$ $T_H17$ cells across all clusters demonstrated that *IL1A*$^+$ $T_H17$ cells displayed significantly enhanced proinflammatory, but reduced anti-inflammatory, signatures (Fig. 2b)[7]. Furthermore, cluster 1, which enriched for *IL1A*$^+$ $T_H17$ cells, was significantly more proinflammatory and less anti-inflammatory than all other five clusters, as indicated by GSEA (Fig. 2b). It is of interest that a bulk transcriptomic comparison of pro- versus anti-inflammatory $T_H17$ cell subsets revealed *IL1A* to even be among the top upregulated genes in the proinflammatory $T_H17$ cell subset (Fig. 2c,d and Extended Data Fig. 2). *IL10*, instead, was highly downregulated, as expected according to previous reports[4,7]. This reciprocal correlation of IL-1α and IL-10 expression by $T_H17$ cells was also observed at the protein level by flow cytometry (Fig. 2e). Enrichment analysis with the DEGs demonstrated overrepresentation of KEGG (*Kyoto Encyclopedia of Genes and Genomes*) pathways for autoimmune diseases such as 'rheumatoid arthritis' and 'inflammatory bowel disease' (Fig. 2f), which supported the proinflammatory nature of the *IL1A*-expressing $T_H17$ cell subset. In fact, patients suffering from juvenile idiopathic arthritis (JIA), a highly inflammatory form of rheumatoid arthritis in children whose pathogenesis has previously been linked to innate IL-1β and IL-1α[18,19], revealed significantly and strongly elevated IL-1α expression by IL-17$^+$ $T_H$ cells compared with IL-17$^+$ $T_H$ cells from healthy control blood (Fig. 2g). No IFN-γ increase was observed within IL-17$^+$ $T_H$ cells from the blood of patients with JIA compared with control donor blood, instead, despite the previously reported association of IFN-γ with $T_H17$ cell pathogenicity (Fig. 2g, right panel)[4]. Furthermore, the analysis of blood-matched synovial fluid demonstrated very high frequencies of IL-1α$^+$ $T_H$ cells, suggesting that T cells, beyond innate cells, could also represent a relevant cellular source of the disease-associated IL-1α at the site of inflammation.

### IL-1α expression is regulated by a $T_H17$ program

To investigate whether IL-1α expression is a general property of T cells, we enriched individual $T_H$ cell subsets from PBMCs according to their differential expression of chemokine receptors (Extended Data Fig. 3) and compared their IL-1α secretion. IL-1α was specifically produced by the $T_H17$ cell subset but not the $T_H1$ cell subset, $T_H2$ cell subset or regulatory T cells ($T_{reg}$ cells) (Fig. 3a–c). Strikingly, its secretion level on stimulation with anti-CD3 and anti-CD28 monoclonal antibodies matched that of human monocytes stimulated with LPS and nigericin, indicating that human $T_H17$ cells, notwithstanding their adaptive immune identity, serve as a major source of the danger signal IL-1α (Fig. 3a).

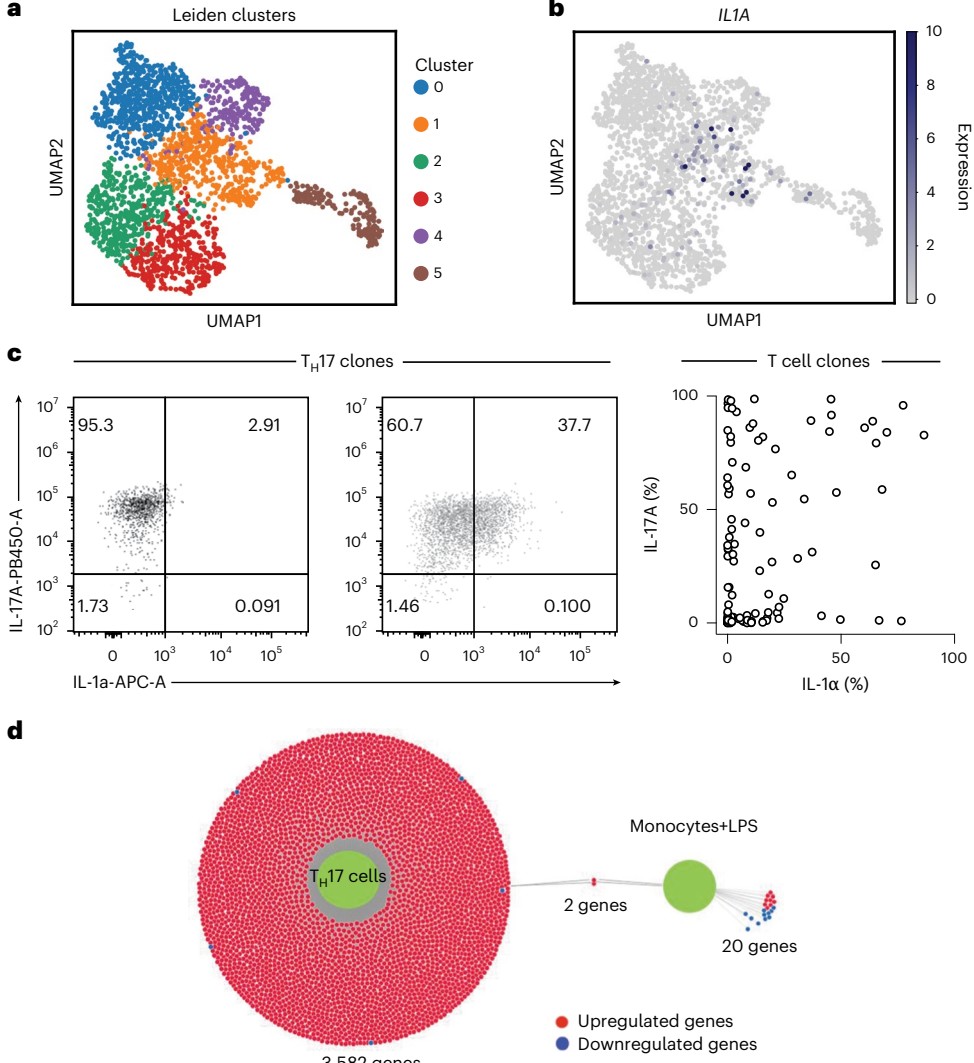

**Fig. 1 | A distinct subset of human $T_H17$ cells can express IL-1α. a**, ScRNA-seq and Leiden clustering of human $T_H17$ cells after 5 d of stimulation with anti-CD3 and anti-CD28 monoclonal antibodies. **b**, *IL1A* expression in $T_H17$ cells visualized in UMAP. **c**, Intracellular cytokine staining and flow cytometry of T cell clones generated from $T_H17$ cells that were isolated ex vivo according to their differential expression of chemokine receptors. Left, representative flow cytometric analysis of one $T_H17$ cell clone. Right, cumulative data from the blood of three healthy donors. **d**, DiVenn plot of DEGs obtained from *IL1A*+ versus *IL1A*− human $T_H17$ cells stimulated as described in **a** (shown as left green circle) and compared with *IL1A*+ versus *IL1A*− human LPS-stimulated monocytes (GEO, accession no. GSE159113) (shown as right green circle). Upregulated genes (red circles) and downregulated genes (blue circles) are connected via a gray line to either green circle, indicating its dataset of origin. Gray lines connecting both green circles depict common DEGs between both datasets.

Intracellular IL-1α protein expression was absent in freshly isolated resting $T_H$ cells but inducible on T cell receptor (TCR) activation, with significant enrichment in $T_H17$ cells compared with $T_H1$, $T_H2$ and $T_{reg}$ cell-enriched $T_H$ cells (Fig. 3b,c). These findings seem to be specific for the human immune system, because previous reports excluded IL-1α production by mouse T cells[16].

The unique association of IL-1α with the $T_H17$ cell subset prompted us to mechanistically dissect the regulation of this cytokine. It is interesting that IL-1α expression was reduced on specific inhibition of ROR-γt, the master transcription factor of $T_H17$ cells (Fig. 3d,e)[1]. These data are in line with the presence of putative binding sites for ROR-γt and ROR-α in the *IL1A* promotor and enhancer regions (Extended Data Fig. 4a,b).

The fate of a particular $T_H$ cell subset is determined by the distinct polarizing cytokine microenvironment during naive T cell stimulation. We observed the highest intracellular expression and secretion of IL-1α on naive T cell priming in $T_H17$ cell-polarizing conditions (IL-1β and

transforming growth factor (TGF)-β) (Fig. 3f,g). Single treatments with IL-1β or TGF-β alone did not, however, lead to significant IL-1α expression in naive T cells, stressing the synergistic effect of the combinatorial $T_H17$ cell-priming cytokines on de novo induction of IL-1α (Supplementary Fig. 4). $T_H1$ (IL-12) and $T_H2$ (IL-4) cell-priming conditions, in contrast, did not significantly alter IL-1α secretion compared with that under stimulation in the absence of polarizing cytokines. Together, these findings demonstrate that naive T cells acquire the capacity to produce IL-1α through $T_H17$ cell-polarizing cytokines. Memory $T_H$ cells also displayed a significant further upregulation of their IL-1α effector cytokine in IL-1β and TGF-β microenvironments (Fig. 3h).

The identity of $T_H$ cell subsets is also characterized by distinct migration properties, which are associated with the differential expression of chemokine receptors[20]. We observed that IL-1α expression was enriched in CCR6+ but not CCR6− T cells and was reduced in CXCR3− T cells compared with CXCR3+ T cells, although there was no difference in IL-1α expression between CCR4+ and CCR4− T cells (Fig. 3i). These

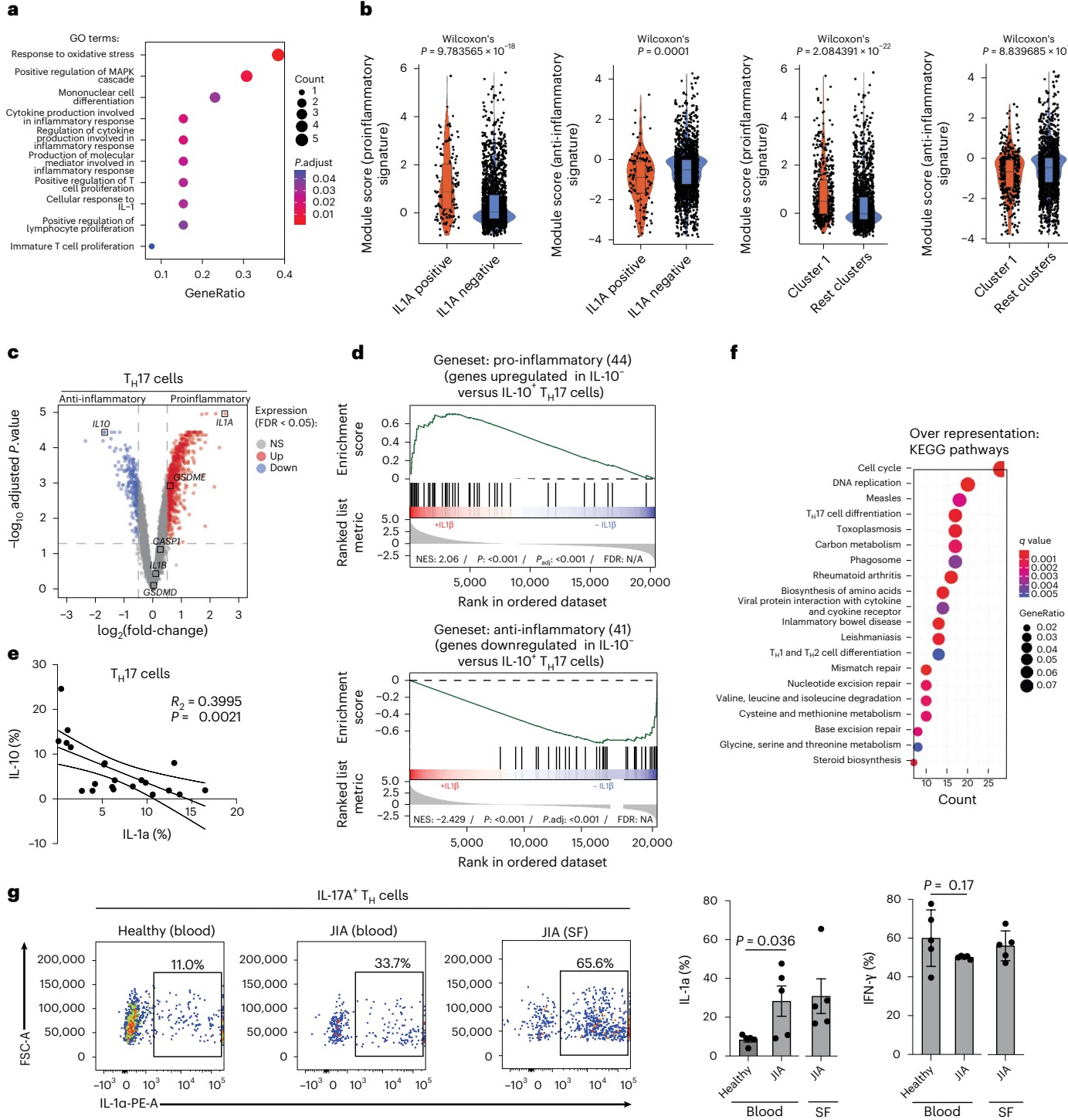

**Fig. 2 | IL-1α producing T_H17 cells are proinflammatory. a**, Enrichment analysis using clusterprofiler with genes coexpressed with *IL1A* as determined in Extended Data Fig. 3a,b. The top 10 GO terms out of 150 significant GO terms are shown. **b**, Expression of pro- and anti-inflammatory gene sets obtained from public data[7] in T_H17 cells analyzed by scRNA-seq after grouping single cells into *IL1A*⁺ and *IL1A*⁻ T_H17 cells and after Leiden clustering (Wilcoxon's rank-sum test). **c**, Transcriptome analysis showing DEGs (red, upregulated; blue, downregulated; gray, nonsignificant genes) of pro- versus anti-inflammatory T_H17 cells after 5 d of polyclonal stimulation in the presence or absence of IL-1β, respectively. **d**, GSEA of T_H17 cells from **c**. The gene sets were established from a public

dataset[7] after transcriptomic comparison of IL-10⁻ versus IL-10⁺ T_H17 cell clones. N/S, not significant; NES, normalized enrichment score. **e**, Intracellular cytokine staining and flow cytometric analysis of T_H17 cells stimulated for 5 d with anti-CD3 and anti-CD28 monoclonal antibodies. **f**, Overrepresentation of KEGG pathways within the DEGs from the transcriptomic comparison of pro- versus anti-inflammatory T_H17 cells. **g**, Intracellular cytokine staining and flow cytometry (left, representative experiment; right, cumulative data) of T cells (from blood and synovial fluid (SF) of patients suffering from JIA and healthy control donors). IL-17A⁺ gated T_H cells are shown (paired Student's *t*-test; *n* = 5 independent patients and healthy donors).

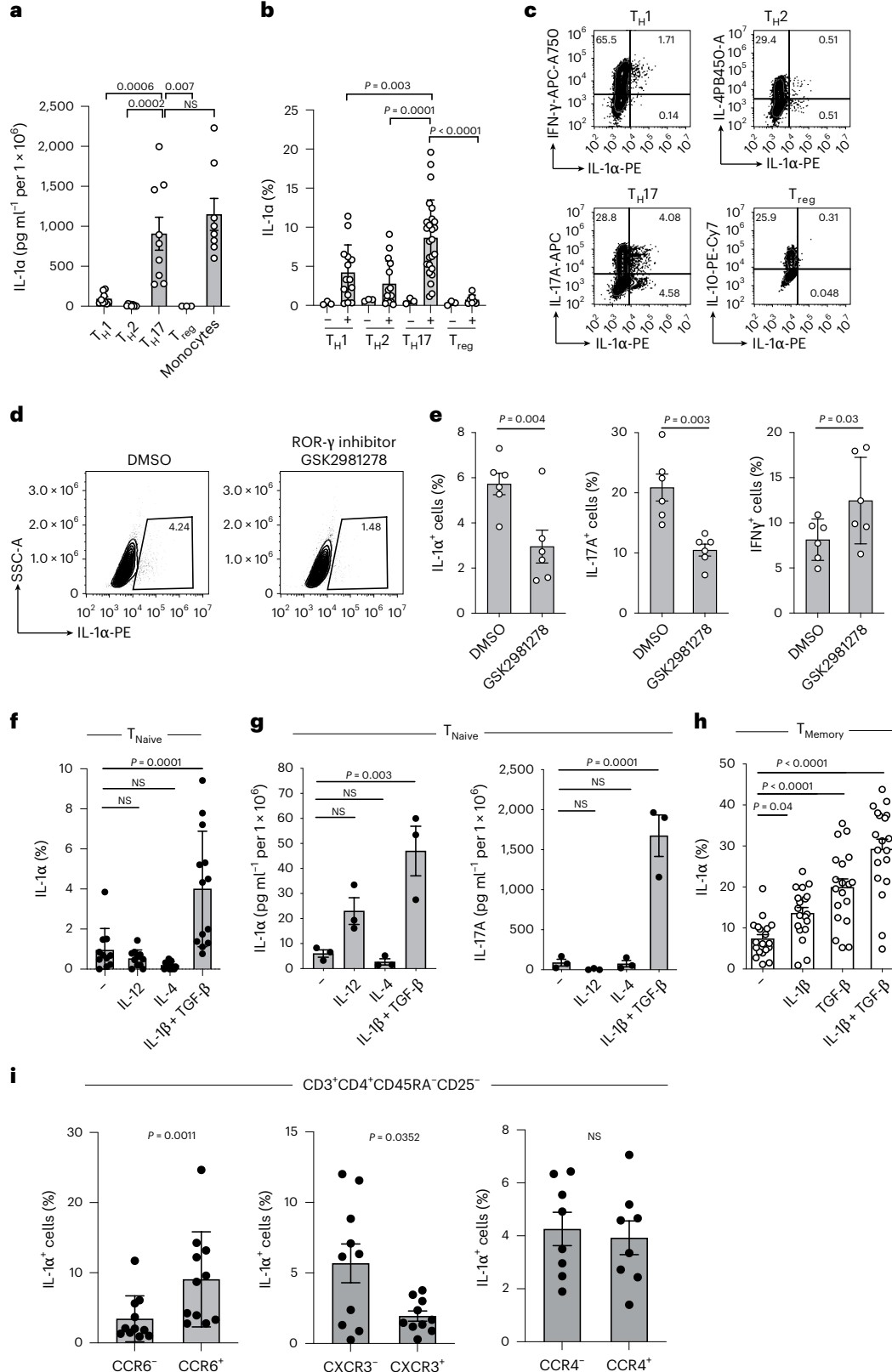

**Fig. 3 | IL-1α production by T cells is restricted to the $T_H$17 cell fate. a**, ELISA of cell culture supernatants of $T_H$ cell subsets after stimulation with anti-CD3 and anti-CD28 monoclonal antibodies for 5 d. Monocytes were stimulated with LPS for 24 h and nigericin for the last 30 min (one-way analysis of variance (ANOVA) with Dunnett's multiple-comparison test). **b–f,h**, Intracellular cytokine staining and flow cytometric analysis of cells stimulated as in **a** (one-way ANOVA with Dunnett's multiple-comparison test (**b**, **f** and **h**), two-tailed paired Student's $t$-test (**e**)). DMSO, Dimethysulfoxide; NS, not significant. **g**, ELISA of cell culture supernatants from cells stimulated as in **f** (one-way ANOVA with Dunnett's multiple-comparison test). **i**, Intracellular cytokine staining and flow cytometry of memory $T_H$ cells sorted positively and negatively for specific chemokine receptors. The analysis was performed after 5 d of stimulation with anti-CD3 and anti-CD28 monoclonal antibodies (two-tailed, paired Student's $t$-test). Data are presented as mean ± s.e.m. Each circle indicates an independent biological sample representing a blood donor.

data demonstrate that IL-1α-producing cells display the migration pattern previously assigned to IL-17-producing cells[15]. Taken together, these data consistently show that IL-1α is a unique function of human T$_H$17 cells.

### Calpain is a prerequisite for IL-1α secretion by T$_H$17 cells

To explore the mechanism of IL-1α secretion, we treated T$_H$17 cells with the protein export inhibitor brefeldin A (BFA) and found that the secretion of IL-1α was not affected, unlike that of conventional cytokines, such as IL-17A (Extended Data Fig. 5a–c). This supports the existence of an unconventional ER–Golgi-independent pathway for IL-1α secretion by T cells and is in line with previous reports for innate cell types[21,22].

A unique property that has previously been assigned to IL-1α is its simultaneous localization in the cytoplasm and the plasma membrane[23]. We found, however, that T$_H$17 cells, in contrast to monocytes, did not display membrane-bound IL-1α (Extended Data Fig. 5d). Although cleavage of pro-IL-1β is required to generate bioactive extracellular IL-1β, IL-1α is known to be passively released on cell death and to exert its bioactive potential after binding to IL-1RI in its uncleaved or cleaved form[24]. To determine whether pro-IL-1α undergoes intracellular processing for controlled release by human T cells, we evaluated the full-length and mature forms of IL-1α in the supernatant of activated T$_H$17 cells. To exclude any contaminating monocytes as a potential source of secreted IL-1α, we generated T$_H$17 cell clones over a period of 2 weeks and restimulated them with anti-CD3 and anti-CD28 monoclonal antibodies for another 5 d before immunoblotting. In all six tested T$_H$17 cell clones, we found preferential enrichment of the cleaved form of IL-1α in the culture supernatants (Fig. 4a). T$_H$17 cell lysates, in contrast, showed preferential enrichment of the uncleaved pro-IL-1α form, as expected (Supplementary Fig. 5a). These results exclude passive release of pro-IL-1α during cell necrosis as the default IL-1α exit modality in T cells and, instead, suggest that human T$_H$17 cells must possess a molecular machinery for pro-IL-1α cleavage to enable the controlled extracellular release of this potent bioactive molecule by viable T cells, in contrast to innate cells (Supplementary Fig. 5b). This does not, however, exclude an additional contribution of T cell necrosis to the liberation of the bioactive pro-form of IL-1α (Supplementary Fig. 5a).

Pro-IL-1α processing at distinct cleavage sites can be catalyzed by several proteases[17,25,26]. Calpain is a calcium-dependent cysteine protease that can give rise to the mature IL-1α p17 fragment[26], which we identified herein. We detected calpain activity in T$_H$17 cells. It increased on activation with anti-CD3 and anti-CD28 monoclonal antibodies (Fig. 4b). IL-1α secretion by T$_H$17 cells was dependent on T cell-intrinsic calpain activity because pharmacological calpain inhibition reduced IL-1α secretion by activated T$_H$17 cells into the extracellular space in a dose-dependent manner (Fig. 4c). Correspondingly, calpain inhibition resulted in intracellular accumulation of pro-IL-1α (Fig. 4d). To corroborate the dependence of IL-1α secretion on calpain, we also genetically knocked out calpain in human T$_H$17 cells using clustered regularly interspaced short palindromic repeats (CRISPR)–Cas9. This revealed a role for *CAPN2*, but not *CAPN1*, in IL-1α secretion by human T$_H$17 cells (Fig. 4e). In contrast, LPS-/nigericin-induced IL-1α secretion by monocytes was not dependent on calpain (Fig. 4f). These data were consistent with the preferential expression of *CAPN2* but not *CAPN1* by human T cell subsets in contrast to dendritic cells, which displayed a reversed calpain gene expression pattern (Supplementary Fig. 6). IL-1α secretion in T$_H$17 cells increased on intracellular accumulation of calcium and inhibition of the sarco-/ER Ca$^{2+}$ ATPase with thapsigargin (Fig. 4g) and decreased on extracellular calcium chelation with (ethylenebis(oxonitrilo))tetra-acetate (EGTA) (Fig. 4h), consistent with the calcium-dependent function of calpain and TCR-dependent IL-1α secretion[26]. Together, these data demonstrated that the proteolytic activity of the calcium-dependent protease calpain is a prerequisite for unconventional IL-1α secretion by TCR-activated human T$_H$17 cells.

### IL-1α secretion by T$_H$17 cells is regulated by NLRP3 inflammasome activation

Despite the essential role of calpain in pro-IL-1α maturation, the mechanism leading to the extracellular exit of cleaved IL-1α remained unknown and was therefore addressed next. Extracellular IL-1α release by myeloid cells has previously been associated with NLRP3 inflammasome activation, nonenzymatic activity of caspase-1 and release of IL-1β[16,27]. We tested whether human T$_H$17 cells possessed the molecular scaffold of the NLRP3 inflammasome and found protein expression of NLRP3 and the adapter molecule, apoptosis-associated speck-like protein containing a CARD (ASC), in human T$_H$17 cells (Extended Data Fig. 6a,b). We observed ongoing inflammasome activation in TCR-activated T$_H$17 cells by identification of ASC specks using ImageStream technology (Fig. 5a,b)[28,29]. Strikingly, increased frequencies of ASC specks were uniquely confined to the T$_H$17 cell subset (Fig. 5b, left). T$_H$17 cells even approximated the ASC-speck formation of LPS- and ATP-stimulated macrophages (Fig. 5b, right). T$_H$1 and T$_H$2 cell subsets, in contrast, displayed background ASC-speck levels (Fig. 5b, left). Moreover, ASC-speck and NLRP3-speck formation were completely abrogated in the presence of the specific NLRP3 inflammasome inhibitor MCC950, substantiating the existence of ongoing NLRP3 inflammasome activation in T$_H$17 cells (Fig. 5b, left and Extended Data Fig. 6c). The selective engagement of the NLRP3 inflammasome by T$_H$17 cells was further supported by the strong induction of *NLRP3* transcripts on T cell stimulation in the presence of the T$_H$17 cell-polarizing cytokines IL-1β and TGF-β (Fig. 5c). Importantly, IL-1α secretion by T$_H$17 cells was significantly reduced by specific inhibition of the NLRP3 inflammasome with MCC950 (Fig. 5d), thus demonstrating the critical role of the NLRP3 inflammasome in the secretion of IL-1α by human T$_H$17 cells.

Caspase-1 is the canonical effector protein in the NLRP3 inflammasome complex[30]. We observed pro-caspase-1 expression in activated human T$_H$17 cells (Fig. 5e). However, in contrast to the findings in LPS- and nigericin-stimulated monocytes, no cleaved caspase-1 was detected in human T$_H$17 cell lysates (Fig. 5e). This observation was corroborated by the absence of caspase-1 FLICA staining in T$_H$17 cells that were stimulated in the presence or absence of IL-1α promoting cytokine stimuli (Extended Data Fig. 7a–e). We further excluded extracellular release of caspase-1 by ELISA after stimulation of T$_H$17 cells with anti-CD3 and anti-CD28 monoclonal antibodies for 5 d (Fig. 5f). Even in the presence of the IL-1α stimulus IL-1β, caspase-1 secretion was not inducible and remained as low as that in resting monocytes (Fig. 5f). Pharmacological inhibition of caspase-1 with Ac-YVAD-CMK did not reduce IL-1α secretion in the presence or absence of IL-1α induction by IL-1β. Rather, we observed slightly elevated IL-1α secretion on caspase-1 inhibition (Supplementary Fig. 7a,b). Importantly, T$_H$17 cells did not show any alteration in IL-1α secretion on CRISPR–Cas9-engineered depletion of *CASP1* expression (Fig. 5g). In line with the absence of bioactive caspase-1, we did not observe any secretion of IL-1β by human T$_H$17 cells. This was in contrast to the case in monocytes, which demonstrated caspase-1-dependent IL-1β secretion on stimulation with LPS and ATP (Extended Data Fig. 7f). Furthermore, no intracellular IL-1β was detectable or inducible by IL-1α-polarizing cytokines in T$_H$17 cells or T$_H$17 cell clones or detectable in activated T$_H$17 cells by scRNA-seq analysis (Extended Data Fig. 7g,h,i). This was in contrast to the finding for monocytes, which coexpressed IL-1β and IL-1α at the single-cell level, consistent with the previously suggested cosecretion and putative coregulation pattern of both IL-1 cytokines (Extended Data Fig. 7g)[16,27]. Cumulatively, these data demonstrate that human T$_H$17 cells produce IL-1α independently of caspase-1 and IL-1β, unlike monocytes and presumably other innate immune cells, despite clear involvement of the NLRP3 inflammasome.

### IL-1a exits T$_H$17 cells via GSDME pores

Gasdermins belong to a family of recently identified pore-forming effector molecules that enable the release of inflammatory mediators[31].

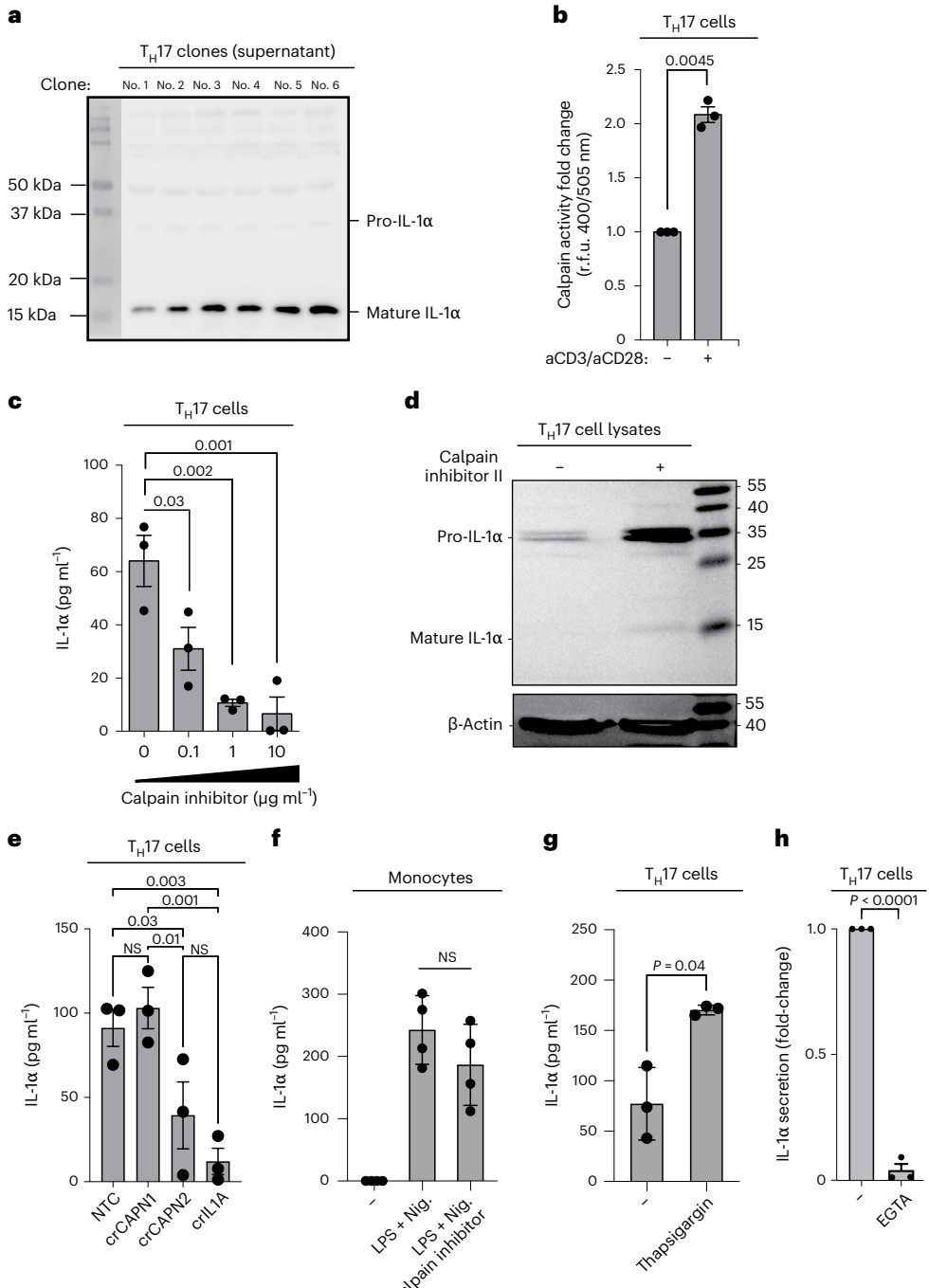

**Fig. 4 | Calpain is a prerequisite for the release of cleaved IL-1α by human T$_H$17 cells. a**, Immunoblot analysis of cell culture supernatants derived from T$_H$17 cell clones that were restimulated with anti-CD3 and anti-CD28 monoclonal antibodies for 5 d. **b**, Fold-change in relative fluorescence units (r.f.u.) after 1 h of incubation of T$_H$17 cells with the calpain substrate Ac-LLY-AFC. T$_H$17 cells were stimulated for 3 d with anti-CD3 and anti-CD28 monoclonal antibodies. **c,e,g,h**, ELISA of cell culture supernatants after stimulation of T$_H$17 cells (**c** and **e**–**g**) with anti-CD3 and anti-CD28 monoclonal antibodies for 5 d and of monocytes (**f**) with LPS (24 h) and nigericin (30 min). Thapsigargin (**g**), 1 µM, was added on days 2 and 3 and EGTA (**h**) on day 0. **d**, Immunoblot analysis of

human T$_H$17 cell lysates. Human T$_H$17 cells were stimulated with anti-CD3 and anti-CD28 monoclonal antibodies and IL-1β in the presence or absence of calpain inhibitor II (10 mM) and analyzed on day 5. The data represent two experiments with two donors. **e**, T$_H$17 cells were stimulated with anti-CD3 and anti-CD28 monoclonal antibodies for 5 d after genetic depletion of CAPN1 or CAPN2 with CRISPR–Cas9 technology. Each circle indicates an independent blood donor. The data represent three independent experiments (**b**, **c** and **e**–**h**). *P* values were calculated using one-way ANOVA with Dunnett's multiple-comparison test (**c**) or Fisher's least significance difference test (**e**) or two-tailed, paired Student's *t*-test (**b** and **f**–**h**).

Our transcriptome analysis revealed a selective upregulation of *GSDME* expression in the proinflammatory IL-1β-stimulated T$_H$17 cell subset, but no regulation of any other member of the gasdermin family (Fig. 6a). This was surprising considering that GSDME expression has never before been reported in primary T cells. In contrast, GSDMD, which is known to be regulated by the NLRP3 inflammasome and to be a target of caspase-1 (ref. 31), was not upregulated, supporting the idea of non-canonical NLRP3 inflammasome signaling. GSDME has previously been

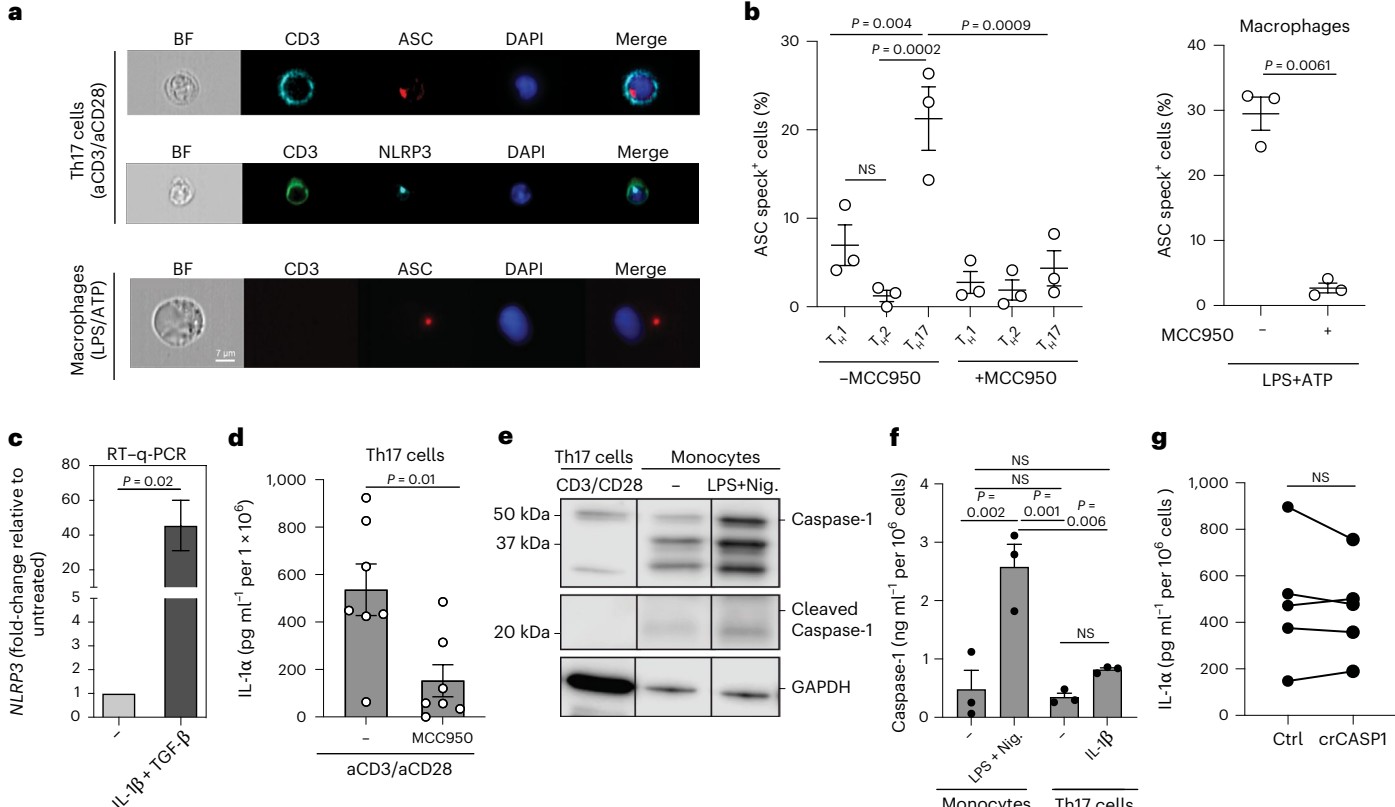

**Fig. 5 | Unconventional NLRP3 inflammasome activation regulates IL-1α production by human $T_H17$ cells. a,b,** Imaging flow cytometry with $T_H17$ cells on day 5 after stimulation with plate-bound anti-CD3 and anti-CD28 monoclonal antibodies and macrophages after 24 h of stimulation with LPS and ATP for the last 30 min. **a,** Representative experiment. BF, bright-field. **b,** Cumulative data with $n = 3$ biological samples, presented as mean ± s.e.m. Left, $P$ values calculated using one-way ANOVA with Tukey's multiple-comparison test. Right, $P$ values calculated using two-tailed, paired Student's $t$-test. **c,** RT–qPCR analysis of $T_H17$ cells stimulated as in **a** and restimulated with PMA and ionomycin for 3 h. Data represent three independent experiments with $n = 9$ biological replicates (two-tailed, paired Student's $t$-test). **d,** ELISA of cell culture supernatants after stimulation of $T_H17$ cells for 5 d with anti-CD3 and anti-CD28 monoclonal antibodies. Data represent three experiments with $n = 7$ biological replicates

(two-tailed, paired Student's $t$-test). **e,** Immunoblot analysis of cell lysates from $T_H17$ cells after 5 d of stimulation with anti-CD3 and anti-CD28 monoclonal antibodies and of monocyte lysates after stimulation with LPS for 24 h and nigericin (Nig.) for the last 30 min. The conditions from the same blot after removal of irrelevant conditions or replicates are shown. **f,** ELISA of cell culture supernatants from anti-CD3- and anti-CD28-activated $T_H17$ cells (5 d) and LPS (24 h)- and nigericin (30 min)-stimulated monocytes ($n = 3$ biological samples presented as mean ± s.e.m.; one-way ANOVA with Tukey's multiple-comparison test). **g,** ELISA of cell culture supernatants from anti-CD3- and anti-CD28-activated $T_H17$ cells after depletion of *CASP1* by CRISPR–Cas9 gene editing. Data represent five independent experiments. $P$ values were calculated using two-tailed, paired Student's $t$-test. Each circle indicates an independent blood donor.

shown to form membrane pores in innate immune cells that may serve as conduits for the extracellular release of alarmins and initiate pyroptotic cell death similar to GSDMD[32]. To test the association of GSDME with the $T_H17$ cell subset, we assessed whether $T_H17$ cell-polarizing cytokines coregulate GSDME. Only the combination of TGF-β with IL-1β ($T_H17$), but not IL-12 ($T_H1$) or IL-4 ($T_H2$), increased *GSDME* transcript levels as assessed by reverse transcription–quantitative PCR (RT–qPCR) (Fig. 6b). This was supported by the existence of putative binding sites for the $T_H17$ cell-associated transcription factors ROR-α and BATF (basic leucine zipper transcription factor, ATF-like) in the promotor regions of *GSDME* (Extended Data Fig. 8a,b)[33]. *GSDMD* expression, in contrast, was not regulated by T cell-polarizing cytokines (Supplementary Fig. 8).

This result prompted us to evaluate GSDME expression at the protein level in human $T_H17$ cells. The GSDME pro-form was inducible on TCR activation. GSDME protein induction in response to this adaptive immune signal was unexpected, but further supported by the existence of putative binding sites of TCR signaling-responsive transcription factors such as NFAT, FOS, JUN and RELA in *GSDME* promotor regions (Extended Data Fig. 8a,b). GSDME was expressed as early as 24 h after polyclonal stimulation. The cleaved N-terminal pore-forming GSDME was detectable at late time points, 3–4 d after TCR stimulation of $T_H17$

cells (Fig. 6c). Full-length GSDMD was concomitantly induced on T cell activation. In contrast, no GSDMD cleavage was observed, as predicted from the absence of caspase-1 and IL-1β secretion, leaving the role of GSDMD in $T_H17$ cells open for further analysis (Fig. 6c).

We next aimed to explore whether GSDME pores serve as conduits for the extracellular release of IL-1α in $T_H17$ cells. We therefore knocked out *GSDME* with CRISPR–Cas9 technology and monitored IL-1α release into the supernatant over time by ELISA. The absence of GSDME, but not GSDMD, significantly inhibited the release of IL-1α by $T_H17$ cells (Fig. 6d). This clearly demonstrates that GSDME pore formation serves as the mechanism for unconventional IL-1α release by human $T_H17$ cells.

## The caspase-8/3 GSDME cleavage cascade enables NLRP3-dependent IL-1α secretion

We next explored the possibility of mechanistic crosstalk between NLRP3 inflammasome activation and GSDME cleavage in human $T_H17$ cells. Caspase-3 has recently been shown to cleave GSDME, which in turn is a target of the NLRP3 inflammasome interactor caspase-8 (refs. 34,35). Indeed, both pro-caspase-8 and pro-caspase-3 were detected in $T_H17$ cells. We found that cleavage of both caspases occurred on TCR stimulation and preceded GSDME cleavage (Fig. 6e). In contrast, no cleaved

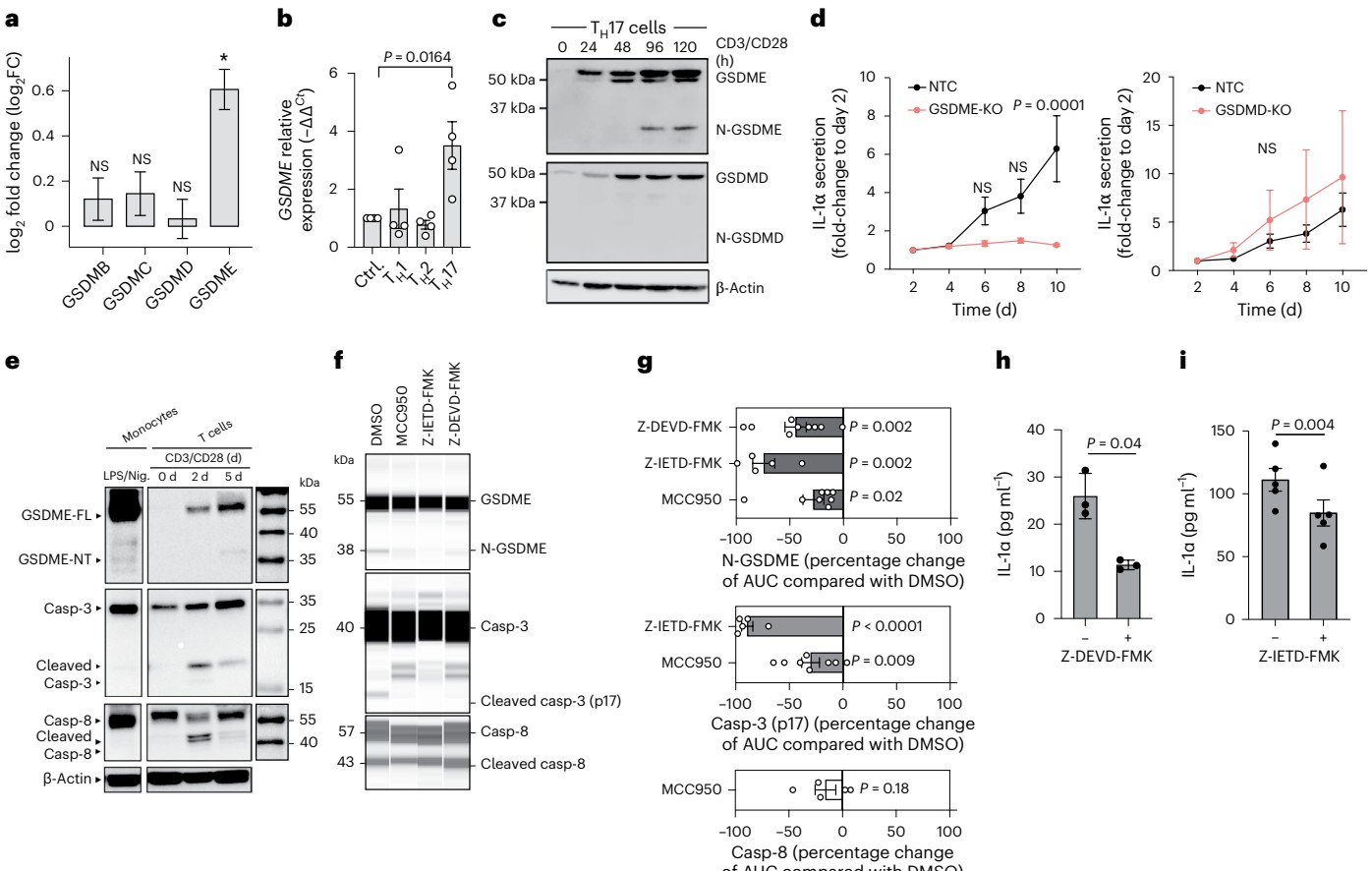

**Fig. 6 | The NLRP3–casp8/3 cleavage cascade leads to GSDME pores for IL-1α release. a**, Differential gene expression determined by transcriptome analysis of T$_H$17 cells treated as in Fig. 2c (*n* = 3 individual healthy blood donors). **b**, RT–qPCR analysis of anti-CD3 and anti-CD28 monoclonal antibody-stimulated, naive T cells in polarizing cytokine conditions (*n* = 4, one-way ANOVA with Dunnett's multiple-comparison test). **c**, Immunoblot analysis of cell lysates from T$_H$17 cells stimulated with anti-CD3 and anti-CD28 monoclonal antibodies for different durations. The data represent three experiments. **d**, ELISA of cell culture supernatants from T$_H$17 cells with and without deletion of *GSDME* (left) or *GSDMD* (right) by CRISPR–Cas9 technology. Individual experiments were normalized to the first time point of analysis on day 2 (*n* = 3 individual biological samples, two-way ANOVA with Bonferroni's multiple-comparison test). **e**, Immunoblot

analysis of cell lysates from T$_H$17 cells stimulated with anti-CD3 and anti-CD28 monoclonal antibodies for the indicated time points and of CD14$^+$ monocytes stimulated for 24 h with LPS and 30 min with nigericin. Casp, Caspase. **f**, Lane view of electropherograms obtained with a Jess Simple Western System for cell lysates of T$_H$17 cells stimulated for 5 d as in **e** in the presence or absence of the indicated inhibitors. It is a representative experiment. **g**, Cumulative data of **f** (one-sample Student's *t*-test). AUC, area under the curve. **h**, Luminex assay of the supernatants of T$_H$17 cells stimulated with plate-bound anti-CD3 (1 μg ml$^{-1}$, TR66) and phorbol-12,13-dibutyrate for 8 h on day 4 of culture (*n* = 3 individual biological samples, two-tailed, paired Student's *t*-test). **i**, ELISA of supernatants of T$_H$17 cells stimulated as in **f**. Each circle indicates an independent blood donor in **h** and **i** (*n* = 4 individual biological samples; two-tailed, paired Student's *t*-test).

products of caspase-8 and caspase-3 were detected in nigericin- and LPS-stimulated monocytes (Fig. 6e). To establish a causative role for these caspases in the cleavage of GSDME and the secretion of IL-1α by T cells, we pharmacologically blocked caspase-3 or caspase-8 activity with the inhibitor Z-DEVD-FMK or Z-IETD-FMK, respectively. Inhibition of caspase-8 activity with Z-IETD-FMK abrogated the cleavage of the downstream target caspase-3, in line with previous reports on other cell types[36]. Both treatments reduced GSDME cleavage while also abrogating IL-1α secretion (Fig. 6f–i and Extended Data Fig. 9a,b). Inhibition of caspase-1, instead, did not affect caspase-3 or GSDME cleavage (Supplementary Fig. 9). These data clearly demonstrated that the caspase-8–caspase-3–GSDME axis was operating in human T$_H$17 cells on TCR activation and that it regulated IL-1α secretion in these cells.

To finally establish the link between this proteolytic cleavage cascade and the NLRP3 inflammasome, we applied MCC950 to stimulated T$_H$17 cells, which, indeed, produced a significant reduction in caspase-3 and GSDME cleavage on day 5 (Fig. 6f,g and Extended Data Fig. 9c). A reduction in caspase-8 cleavage was, however, less pronounced at the same time point of analysis, which was in line with its earlier activation

time window (Fig. 6e,f). In summary, targeted inhibition of each individual molecular player established the NLRP3 inflammasome–caspase-8–caspase-3–GSDME cascade as the proteolytic pathway involved in the extracellular release of bioactive IL-1α by human T$_H$17 cells.

## T$_H$17 cells are resilient to pyroptosis despite GSDME pores

Gasdermin pore formation has previously been associated with pyroptotic cell death in a variety of cell types[13,14,37]. We found expression of the cleaved pore-forming N-GSDME unit in the plasma membrane but not the cytosol, which supported the pore-forming function of GSDME in T cells (Fig. 7a). Surprisingly, a transcriptomic comparison of GSDME-intact and CRISPR–Cas9 gene-edited, GSDME-deficient bulk human T$_H$17 cells by GSEA excluded differences in various forms of cell death; however, notably, it revealed that the affected processes and pathways in GSDME-deficient T$_H$17 cells were mainly related to genes controlling transmembrane transport, in line with the pore-forming conduits formed by N-GSDME (Fig. 7b and Supplementary Fig. 10). A single-cell transcriptomic comparison of individual T$_H$17 cells selected for the presence versus the absence of *GSDME* expression supported

the viability of GSDME-expressing $T_H17$ cells by their gene set enrichment for proliferation (Fig. 7c). We finally validated the resilience of human GSDME-expressing $T_H17$ cells to pyroptosis by demonstrating the absence of any difference in lactate dehydrogenase (LDH) release across CRISPR–Cas9 gene-edited, GSDME-deficient and GSDME-intact $T_H17$ cells on TCR stimulation (Fig. 7d).

We then compared IL-1α⁺ and IL-1α⁻ $T_H17$ cells with respect to their viability and proliferation potential, because $T_H17$ cells did not display any IL-1α surface expression in contrast to IL-1α-producing cells of other cellular lineages, such as monocytes (Extended Data Fig. 5d). This comparison of IL-1α⁺ and IL-1α⁻ $T_H17$ cells necessitated the establishment of a homemade IL-1α-secretion assay, enabling isolation of IL-1α⁺-viable $T_H17$ cells by capture of secreted autocrine IL-1α to the cell surface after phorbol-12-myristate-13-acetate (PMA) and ionomycin stimulation. No difference in the cloning efficiency of $T_H17$ cells that were deposited as single cells after sorting for the presence or absence of surface IL-1α was observed (Supplementary Fig. 11). This finding excluded differences in the viability and expansion of IL-1α⁺ and IL-1α⁻ $T_H17$ cells at the single-cell level.

We further recloned $T_H17$ clones that were screened, based on varying degrees of intracellular IL-1α expression, and monitored their respective cloning efficiency. If GSDME-enabled IL-1α release was associated with pyroptotic cell death, then an inverse correlation between IL-1α expression in T cell clones and the frequency of growing clones on their individual recloning (cloning efficiency) was to be expected. However, the T cell recloning efficiency was independent of IL-1α expression levels in the $T_H17$ cell clones and was instead similar to that of control T cell clones, selected on the basis of varying expression levels of IFN-γ in the absence of IL-1α coexpression (Fig. 7e). Importantly, IL-1α⁺ $T_H17$ cell clones continued to re-express IL-1α on repetitive TCR restimulation cycles (Fig. 7f). Thus, a cell death-associated loss of IL-1α-producing cells from their respective clonal T cell population on restimulation was excluded. Notably, this finding indicates a T cell cytokine memory for reinducible IL-1α.

We performed a transcriptomic comparison between bulk IL-1α⁺ and IL-1α⁻ $T_H17$ cells and observed even greater enrichment for proliferation gene sets in IL-1α⁺ compared with IL-1α⁻ $T_H17$ cell clones (Fig. 7g). Single-cell transcriptomic comparison of $IL1A$⁺ versus $IL1A$⁻ $T_H17$ cells corroborated the transcriptomic enrichment for proliferation. This was also the case for the comparison of Leiden cluster 1, which was enriched for $IL1A$-expressing $T_H17$ cells and inflammatory signatures, with all other clusters (Fig. 7h). IL-1α⁺ $T_H17$ cells displayed higher Ki67 expression according to flow cytometric analysis than their IL-1α⁻ counterparts after 5 d of polyclonal stimulation (Fig. 7i), again supporting the idea that in human $T_H17$ cells IL-1α exit is not associated with cell death, unlike in innate cells.

Cumulatively, these data exclude an association of IL-1α production with (pyroptotic) cell death even at the single-cell level and demonstrate that human T cells have a cytokine memory for IL-1α production on repetitive TCR restimulation.

## T cell-derived IL-1α contributes to antifungal host defense

Our finding that human $T_H17$ cells produce the innate danger signal IL-1α and repurpose an innate signaling machinery for its extracellular release blurs the distinction of adaptive versus innate immune responses and thus extends the overall functional repertoire of T cells. A critical feature that remains characteristic for adaptive memory responses is TCR-endowed antigen specificity. We therefore investigated whether the ability of human $T_H17$ cells to produce IL-1α is restricted to specific antigen specificities. $T_H17$ cells have previously been shown to be highly enriched within cells specific for *C. albicans* and *Staphylococcus aureus* antigens[15]. It is interesting that we observed significantly greater IL-1α expression and secretion by *C. albicans*-specific than by *S. aureus*-specific $T_H17$ cell clones (Fig. 8a). We next investigated whether naive T cells recognizing *C. albicans* rather than *S. aureus* would

be primed by their cognate antigen to selectively produce IL-1α. Naive (CD45RA⁺CCR7⁺) $T_H$ cells were cocultured with autologous monocytes pulsed with heat-inactivated *C. albicans* or *S. aureus* antigens or stimulated polyclonally with anti-CD3 and anti-CD28 monoclonal antibodies, and then cloned on day 7 with allogeneic feeder cells. All clones were restimulated on day 14 with anti-CD3 and anti-CD28 monoclonal antibodies for 5 d for ELISA of their supernatants. Strikingly, we found that *C. albicans*, but not *S. aureus* or polyclonal TCR activation, induced de novo IL-1α production in human differentiating $T_H17$ cells (Fig. 8b). Our combined ex vivo recall and in vitro priming approach therefore establishes that the ability to produce IL-1α is confined to T cells with TCR specificity for *C. albicans*.

We finally tested whether the distinctive ability of *C. albicans*-specific $T_H17$ cells to produce IL-1α is associated with a physiological role in antifungal host defense. For this, we cocultured human monocytes with supernatants from human $T_H17$ cells after their polyclonal restimulation with anti-CD3 and anti-CD28 monoclonal antibodies for 5 d. We observed significantly increased phagocytosis of FITC-labeled *C. albicans* by monocytes using flow cytometry. Importantly, the increased *C. albicans* phagocytosis by $T_H17$ cell supernatants was IL-1α dependent as shown by significant abrogation of *C. albicans* phagocytosis if $T_H17$ cell supernatants were devoid of IL-1α after immunoabsorption or CRISPR–Cas9-targeted IL-1α depletion in $T_H17$ cells (Fig. 8c). Live cell in vitro imaging further corroborated the increased uptake and elimination of *C. albicans* by monocytes in the presence of IL-1α-containing $T_H17$ cell supernatants (Fig. 8d, video data). Taken together, these findings strongly suggest that $T_H17$ cells clear *C. albicans* infections not only via their production of IL-17, as previously thought[38], but also, to a significant extent, via the ability of a unique $T_H17$ cell subset to produce IL-1α.

Cumulatively, the findings identifying GSDME pore formation in T cells as an exit strategy for proinflammatory IL-1α and the regulation of GSDME by the NLRP3 inflammasome–caspase-8–caspase-3 axis reveal a new mode of T cell cytokine secretion that is associated with a proinflammatory subset of $T_H17$ cells with antifungal TCR specificities. This provides new therapeutic targets for the modulation of human $T_H17$ cells that are relevant for antifungal host defense and might also participate in the pathogenesis of chronic inflammatory diseases.

## Discussion

In summary, our findings reveal a previously unknown biological pathway and cytokine secretion modality in human T cells that diversifies the overall functionality of the T cell population.

We found that IL-1α expression was uniquely confined to the $T_H17$ cell fate, as evidenced by its coexpression with IL-17A, regulation by ROR-γt, induction by the $T_H17$ cell-priming cytokines IL-1β and TGF-β and by its $T_H17$ cell-associated chemokine receptor expression profile. These findings are consistent with the existence of binding sites for ROR-γt and ROR-α in the $IL1A$ enhancer and promoter regions as described in the present study and with the previously reported ROR-γt-binding sites in the $NLRP3$ promotor region[39]. We further observed that $T_H17$ cell-priming cytokines increased $NLRP3$ and $GSDME$ expression. Accordingly, master regulators of the $T_H17$ cell fate, such as ROR-α and BATF, as well as multiple TCR-inducible transcription factors, displayed putative binding sites in the GSDME enhancer regions. $T_H17$ cell polarization therefore promoted not only IL-1α expression but also its extracellular exit (Extended Data Fig. 10, graphic summary).

Unexpectedly, we found the NLRP3 inflammasome to be active and repurposed for the release of IL-1α instead of IL-1β in TCR-activated $T_H17$ cells. Unlike innate cells, T cells are not specialized for innate danger sensing, which is known to trigger the assembly of NLRP3 inflammasome components. However, elevation of cytoplasmic $Ca^{2+}$ has previously been shown to bypass innate danger signaling for NLRP3 inflammasome activation[16,40]. This is consistent with our finding that

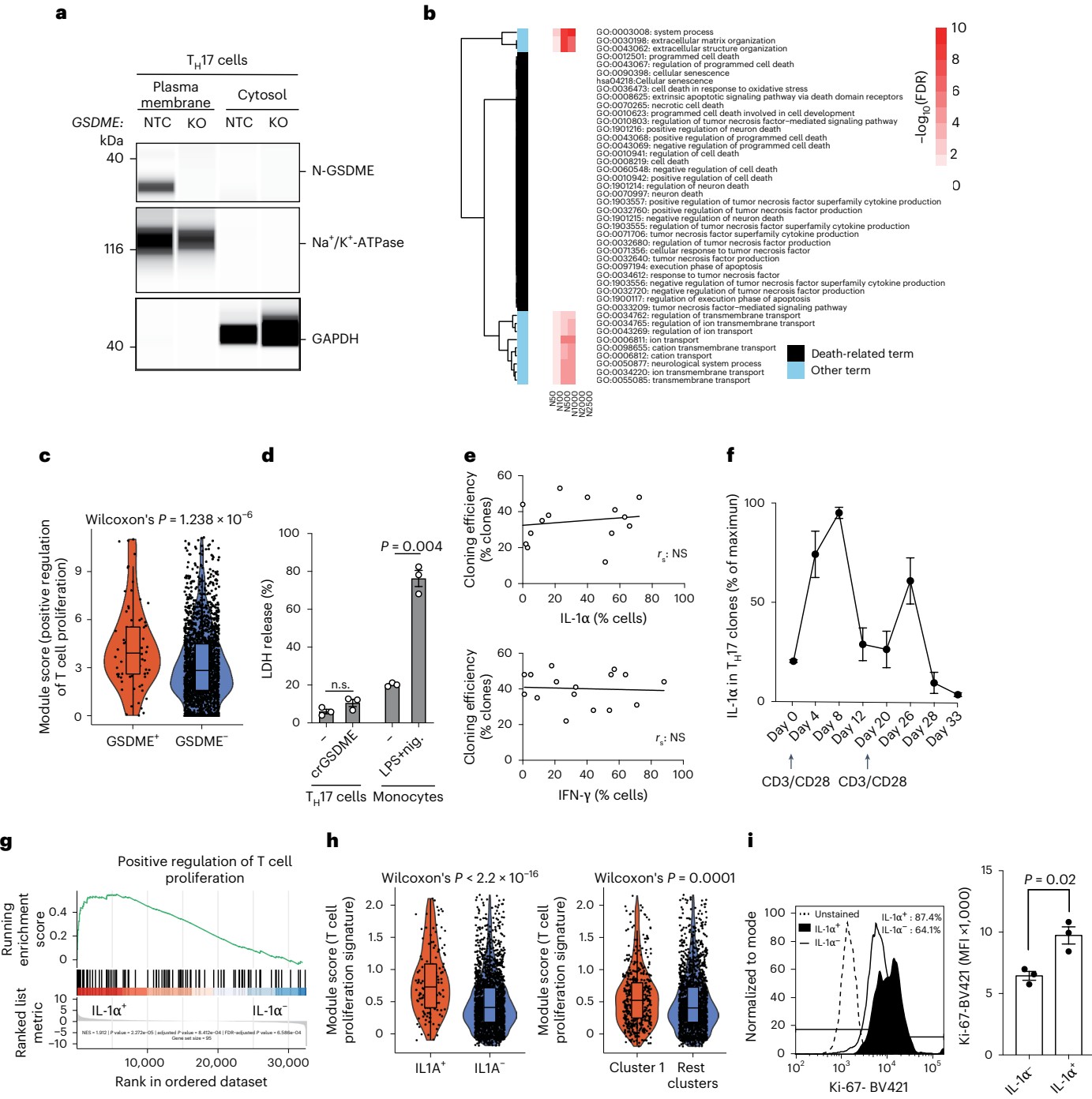

**Fig. 7 | T$_H$17 cells are resilient to pyroptosis despite GSDME plasma membrane pores. a**, Representative electropherogram obtained with a Jess Simple Western System after normalization to total protein. T$_H$17 cells were stimulated with anti-CD3 and anti-CD28 monoclonal antibodies for 48 h and then transfected with RNPs containing an NTC or crGSDME (KO). The T$_H$17 cells were then expanded for another 7 d. The data represent three experiments. **b**, Heatmap with gene sets constructed based on the fold-changes of the genes (see Supplementary Fig. 10). All annotation terms significant in at least three gene sets (FDR ≤ 5%) are shown. The observed −log$_{10}$(FDR) values were capped at 10 for ease of visualization. In addition, all cell death-associated annotation terms are shown. **c**, Gene set expression comparison using a GO term in T$_H$17 cells analyzed by scRNA-seq after grouping into *GSDME*$^+$ and *GSDME*$^-$ T$_H$17 cells (Wilcoxon's rank-sum test). **d**, CytoTox 96 Non-Radioactive Cytotoxicity Assay from T$_H$17 cells with and without CRISPR–Cas9 gene editing for *GSDME* stimulated with anti-CD3 and

anti-CD28 monoclonal antibodies or from monocytes stimulated with or without LPS and nigericin (24 h). Supernatants from washed T$_H$17 cell cultures were collected between days 4 and 5 of stimulation or from monocytes 24 h after stimulation (paired Student's *t*-test). **e**, Cloning efficiency of T$_H$17 cell clones with varying degrees of IL-1α expression (top) and of control T$_H$ cell clones with varying degrees of IFN-γ expression, but lacking IL-1α coexpression (bottom) as assessed by intracellular cytokine staining. **f**, Intracellular staining and flow cytometric analysis of T$_H$17 cell clones after repetitive restimulation with anti-CD3 and anti-CD28 monoclonal antibodies (*n* = 5 individual T$_H$17 cell clones). **g**, GSEA of IL-1α$^+$ compared with IL-1α$^-$ T$_H$17 cell clones. **h**, Gene set expression comparison after scRNA-seq as in **c**. **i**, Flow cytometric analysis of T$_H$17 cells stimulated for 5 d with anti-CD3 and anti-CD28 monoclonal antibodies. Left, representative experiment. Right, cumulative data (*n* = 3, two-tailed, paired Student's *t*-test). Each circle indicates an independent blood donor.

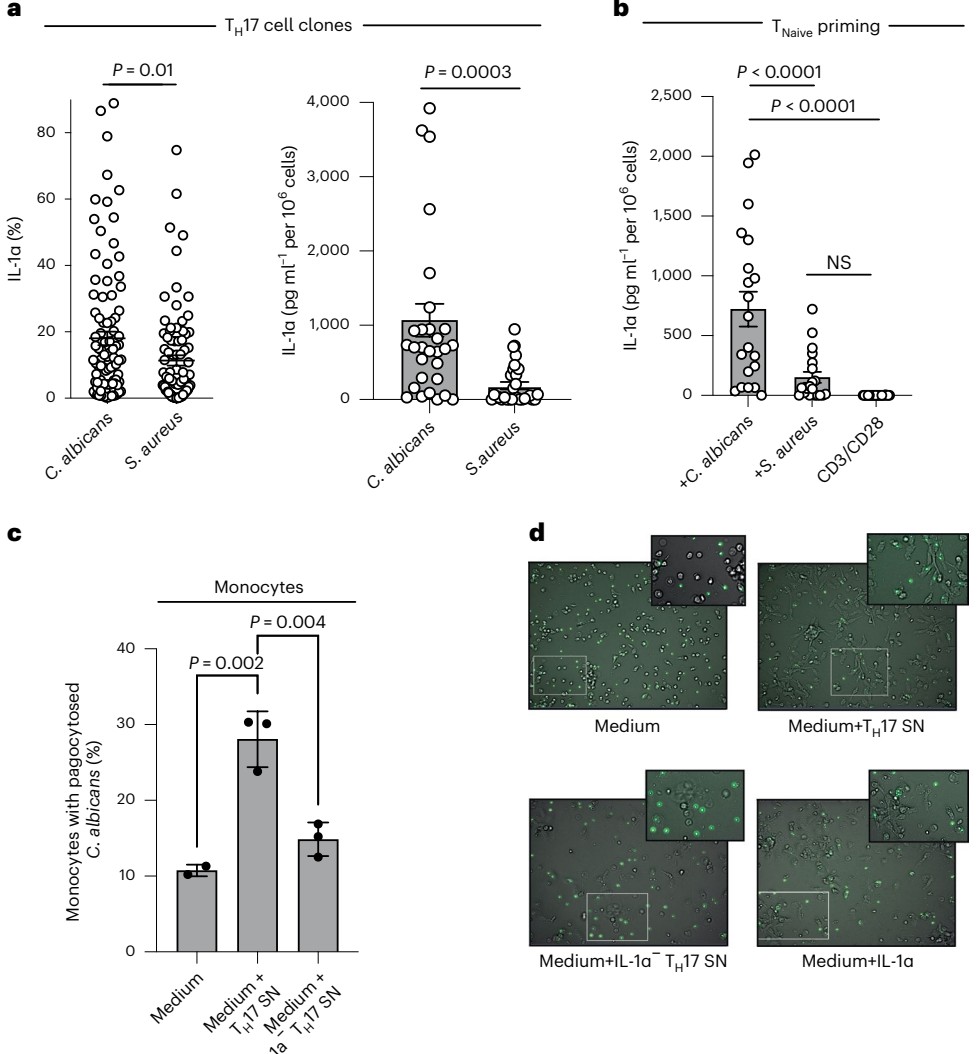

**Fig. 8 | TCR specificity controls IL-1α production contributing to *C. albicans* clearance. a**, Intracellular cytokine staining and flow cytometry (left) and ELISA (right) of *C. albicans*- versus *S. aureus*-specific T$_H$17 cell clones from three individual blood donors 14 d after single-cell T$_H$17 cell cloning with irradiated feeder cells and restimulation for 5 d with anti-CD3 and anti-CD28 monoclonal antibodies. The microbial antigen-specific T$_H$17 cells were isolated for subsequent cloning as a carboxyfluorescein succinimidyl ester (CFSE)-negative population after their restimulation with microbe-pulsed autologous monocytes. Each circle indicates an individual T cell clone (*n* = 30 T$_H$ cell clones, 10 clones per healthy blood donor). Data are presented as mean ± s.e.m. (two-tailed, unpaired Student's *t*-test). **b**, Naive T$_H$ cells were primed by either *C. albicans*- or *S. aureus*-pulsed monocytes. Each circle indicates an individual

T cell clone. The ELISA analysis of supernatants after restimulation of each clone with anti-CD3 and anti-CD28 monoclonal antibodies for 5 d is shown (*n* = 19, 4–5 clones per healthy blood donor; one-way ANOVA with Tukey's multiple-comparison test). **c**, Flow cytometric analysis of phagocytosis of FITC-labeled, heat-inactivated *C. albicans* yeast by monocytes preincubated for 18 h with IL-1α replete or depleted (immunoabsorption or CRISPR–Cas9 KO) T$_H$17 cell supernatants (*n* = 3 independent biological samples; one-way ANOVA with Tukey's multiple-comparison test). Each circle indicates an independent blood donor. **d**, Real-time live cell in vitro imaging (videos in Supplementary Video 1, time points) of monocytes in coculture with FITC-labeled *C. albicans* as in **c**. Representative snap shots with magnifications of the videos are shown at a time point 2 h after addition of *C. albicans*.

TCR activation, which is accompanied by calcium flux, is a requirement for IL-1α release by human T$_H$17 cells.

We observed that T$_H$17 cells engaged an alternative NLRP3 downstream signaling cascade via engagement of caspase-8. This might have been facilitated by the absence of caspase-1 cleavage, because competitive caspase-1 versus caspase-8 inflammasome recruitment has been demonstrated previously[34,41]. We also found pro-caspase-1 and uncleaved GSDMD in human T$_H$17 cells, but no evidence for their NLRP3 inflammasome-regulated cleavage or for IL-1β production. The expression of their precursors raises the question of whether classic NLRP3 inflammasome signaling and IL-1β release might also operate in human T$_H$17 cells if alternative yet-to-be-identified stimuli

are applied. This implies that caspase-8- versus caspase-1-dependent counterregulatory mechanisms might control a dichotomy of IL-1α versus IL-1β production by T cells, consolidating previously suggested roles for the NLRP3 inflammasome in the release of IL-1β in human T$_H$1 cells[42] and murine T$_H$17 cells[43].

An intriguing observation of our study was identification of GSDME expression and cleavage in T cells. This revealed that unconventional cytokine secretion via membrane pores can occur in T cells. Several of the functions recently assigned to GSDME have been associated with pyroptosis and subsequent enhancement of tumor cell death and an inflammatory microenvironment[32,44]. Surprisingly, we found that GSDME-expressing T$_H$17 cells instead displayed preserved viability

and continued proliferation on repetitive TCR stimulation compared with GSDME-deficient T cells. The same results were observed for IL-1α[+] compared with IL-1α[−] T cells. These findings were unexpected, considering that IL-1α production has thus far been considered a hallmark of senescence and thus of replication-arrested or dying cells[45]. This evokes the idea that the danger signal IL-1α can be part of a T cell-associated cytokine memory that is re-excitable on cognate antigen recognition[46]. Furthermore, the GSDME pores might serve a physiological function to enable $T_H17$ cells to release additional as-yet-unidentified molecules beyond IL-1α that are defined by their size or charge[47]. This would be consistent with the results of our unbiased transcriptomic comparison of GSDME-intact or CRISPR–Cas9-engineered, GSDME-deficient $T_H17$ cells, which revealed multiple roles for GSDME in transmembrane transport but not cell death. The mechanism, by which the viability and long-term IL-1α cytokine memory in T cells with GSDME pores is preserved, remains to be explored in the future.

The availability of IL-1α from different cellular sources, particularly from innate APCs, raises the question about the relative contribution of T cell-derived IL-1α to human health and disease. Although IL-1α[+] T cells constitute only a small subset within the T cell lineage, their ability to produce IL-1α was quantitatively comparable to that of LPS-stimulated monocytes, supporting the new concept that T cells can serve as a relevant source of inflammatory IL-1α. Our findings from transcriptomic and functional analyses reveal IL-1α to be associated with the proinflammatory fate of $T_H17$ cells. The IL-1α[+] subpopulation of human $T_H17$ cells displayed transcriptomic signatures of enhanced inflammatory pathogenicity and associations with chronic inflammatory diseases. The significantly enhanced expression of IL-1α by circulating $T_H17$ cells from patients with JIA and their abundant localization in the inflamed synovial fluid support this pathogenic potential of the IL-1α[+] subset of $T_H17$ cells. A rigorous causal relationship between IL-1α-producing $T_H17$ cells and the pathogenesis of JIA still remains to be established and the relative impact of IL-1α[+] $T_H17$ cells validated in future clinical trials.

A striking observation was that IL-1α production by T cells is hard-wired through their TCR specificity. Although IL-1α production by innate cellular sources is triggered by nonspecific stress stimuli[16], we found IL-1α production by human $T_H17$ cells to be associated with a TCR specificity for *C. albicans*. *S. aureus*-specific $T_H17$ cells, instead, displayed significantly reduced IL-1α production. These findings are consistent with the differential requirement of IL-1β for the generation of *C. albicans*- but not *S. aureus*-specific $T_H17$ cells, as previously reported[4] and, accordingly, with the critical role of IL-1β for the induction of IL-1α expression, as reported here. In addition, we found IL-1α secretion to be dependent on TCR stimulation and calcium signals, stressing its tight association with specific adaptive immune signaling via the TCR.

$T_H17$ cells are known to be the protagonists for the clearance of *C. albicans* infections through their secretion of IL-17, which is exemplified by *C. albicans* dysbiosis in settings of genetic or therapeutic IL-17 deficiencies[38]. We found the $T_H17$ cell product IL-1α to be involved in *C. albicans* clearance because its absence in $T_H17$ cell supernatants significantly reduced *C. albicans* phagocytosis by monocytes. This suggests that antifungal $T_H17$ cell effector functions are exerted not only through IL-17A/F, as previously suggested, but also through IL-1α in a TCR-specific manner. Whether aberrant regulation of the molecular pathway leading to IL-1α production by $T_H17$ cells could predispose to compromised antifungal host defense will therefore need to be tested in the future.

Cumulatively, our findings pave the way for a systematic investigation of the contributions of IL-1α-producing $T_H17$ cells in various inflammatory diseases and antifungal host defense. The TCR–NLRP3 inflammasome–caspase-8–caspase-3–GSDME axis not only represents a previously overlooked mode of immune signaling and fate instruction in $T_H$ cells but also provides molecular targets to either disrupt a pathogenic $T_H17$ cell identity or to harness it for host defense.

## Online content

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

[1]Department of Infection Immunology, Leibniz Institute for Natural Product Research and Infection Biology, Hans-Knöll-Institute, Jena, Germany. [2]Center for Translational Cancer Research & Institute of Virology, Technical University of Munich, Munich, Germany. [3]Research Unit Cellular Signal Integration, Molecular Targets and Therapeutics Center, Helmholtz Zentrum München—German Research Center for Environmental Health, Neuherberg, Germany. [4]Department of Systems Biology and Bioinformatics, Leibniz Institute for Natural Product Research and Infection Biology, Hans-Knöll-Institute, Jena, Germany. [5]Center for Translational Immunology, University Medical Center Utrecht and Utrecht University, Utrecht, the Netherlands. [6]Institute of Biomedical Informatics, Graz University of Technology, Graz, Austria. [7]Adaptive Pathogenicity Strategies, Leibniz Institute for Natural Product Research and Infection Biology—Hans Knöll Institute, Jena, Germany. [8]Institute of Neuropathology, Medical Center & Signalling Research Centres BIOSS and CIBSS & Center for Basics in NeuroModulation, Faculty of Medicine, University of Freiburg, Freiburg, Germany. [9]Institute of Microbiology, Faculty of Biological Sciences, Friedrich Schiller University, Jena, Germany. [10]German Center for Infection Research, Munich, Germany. [11]Department of Cellular Immunoregulation, Charité-Universitätsmedizin Berlin, Berlin, Germany. ✉e-mail: christina.zielinski@leibniz-hki.de

## Methods

### Cell purification and sorting

PBMCs were isolated by density gradient centrifugation using Ficoll-Paque Plus (GE Healthcare). CD4$^+$ T cells were isolated from fresh PBMCs by positive selection with CD4-specific MicroBeads (Miltenyi Biotec) using an autoMACS Pro Separator. T$_H$ cell subsets were sorted to at least 98% purity as follows: T$_H$1 cell subset, CXCR3$^+$CCR4$^-$CCR6$^-$CD45RA$^-$CD25$^-$CD14$^-$; T$_H$2 cell subset, CXCR3$^-$CCR4$^+$CCR6$^-$CD45RA$^-$CD25$^-$CD14$^-$; and T$_H$17 cell subset, CXCR3$^-$CCR4$^+$CCR6$^+$CD45RA$^-$CD25$^-$CD14$^-$, as described previously[4,6,48]. Memory T$_H$ cells were isolated as CD3$^+$CD14$^-$CD4$^+$CD45RA$^-$ lymphocytes, and naive T cells were isolated as CD3$^+$CD14$^-$CD4$^+$CD45RA$^+$CD45RO$^-$CCR7$^+$ lymphocytes to a purity >98%. Cells were stained with the following fluorochrome-conjugated antibodies: CCR4-PE/Cy7 (1:200), CCR6-BV421 (1:100), CD14-PacificBlue (1:200–1:400), CD3-FITC (1:150), CD3-APC (1:100), CD4-APC/Cy7 (1:300), CD45RA-FITC (1:200), CD8-PacificBlue (1:100), CCR7-PE (1:50) and CD25-BV421 (1:100) (all from BioLegend); CCR6-PE (1:50) and CXCR3-APC (1:10) (both from BD). T$_H$ cells were sorted with a BD FACSAria III (BD Biosciences) and a BD FACSAria Fusion (BD Biosciences) or an Aurora CS Sorter. Ethical approval for the use of healthy control and patient PBMCs was obtained from the Institutional Review Board of the Technical University of Munich (195/15s, 491/16S, 146/17S), the Charité-Universitätsmedizin Berlin (EA1/221/11) and the Friedrich Schiller University Jena (2020-1984_1). Synovial fluid was obtained from patients with JIA and active disease (oligoarthritis) undergoing therapeutic joint aspiration, with approval of the local ethics committee of the University Medical Center Utrecht. All experiments involving humans were carried out in accordance with the Declaration of Helsinki.

### Cell culture

Human T cells were cultured as described previously[48]. In some experiments, T cell culture was performed in the presence of recombinant cytokines (IL-6, 50 ng ml$^{-1}$; IL-12, 10 ng ml$^{-1}$; IL-4, 10 ng ml$^{-1}$; TGF-β, 10 ng ml$^{-1}$; IL-1β, 20 ng ml$^{-1}$; all from R&D Systems) or neutralizing antibodies (anti-IL-1α, 10 µg ml$^{-1}$, BD Biosciences). The cell cultures were supplemented with the following pharmacological inhibitors where indicated: Z-IETD-FMK (40 µM, R&D Systems), Z-DEVD-FMK (40 µM, R&D Systems), MCC950 (10 µM, R&D Systems), calpain inhibitor II $N$-acetyl-ʟ-leucyl-ʟ-leucyl-ʟ-methionine (0.1–10 µg ml$^{-1}$, R&D Systems), thapsigargin (1 mM, EMD Millipore), Ac-YVAD-CMK (50 µM, R&D Systems) and GSK2981278 (10 µM, Cayman Chemical). T cells were stimulated with plate-bound anti-CD3 (2 µg ml$^{-1}$, clone TR66) and anti-CD28 monoclonal antibodies (2 µg ml$^{-1}$, clone CD28.2; both from BD Biosciences) for 48 h before transfer into uncoated wells for another 3 d for a total culture period of 5 d, unless indicated otherwise in the legends. Supernatants from these 5-d cultures were used for phagocytosis assays with monocytes and heat-killed, FITC-labeled *C. albicans* yeast. T cell clones were generated in nonpolarizing conditions as described previously after single-cell deposition with FACS or by limiting dilution cloning (Messi, 2003 no. 70). Human monocytes were isolated from PBMCs by positive selection with CD14-specific MicroBeads (Miltenyi Biotec). Cells were stimulated with or without 1 µg ml$^{-1}$ of ultrapure LPS-EB (catalog no. tlrl-3pelps, InvivoGen) for 24 h and nigericin (10 µg ml$^{-1}$, InvivoGen) or ATP (5 mM, Thermo Fisher Scientific) for the last 30 min. In some experiments, CD14$^+$ magnetic activated cell sorting (MACS)-sorted monocytes were differentiated into macrophages for 7 d in the presence of granulocyte–macrophage colony-stimulating factor (R&D Systems).

### Pathogen-specific assays

*C. albicans* and *S. aureus* lysates were prepared as described previously[9,49]. Autologous monocytes were isolated by positive selection with CD14-specific microbeads (Miltenyi Biotec) and pulsed with the pathogen lysates for 3 h. T$_H$17 cells or naive CD4$^+$ T cells were isolated as described above and labeled with CellTrace Violet (CTV; Invitrogen) according to the manufacturer's recommendations and cocultured with pathogen-pulsed monocytes at a ratio of 2:1 for 7 d. CTV$^-$ T$_H$17 cells were FACS sorted and cloned by limiting dilution as described previously[4,50]. Intracellular cytokine staining of T$_H$17 cell clones and flow cytometry with a CytoFLEX (Beckman Coulter) were performed on day 14. CTV$^-$ microbe-primed T$_H$ cells originating from seeded naive T cells were FACS sorted on day 7, cloned and restimulated on day 14 with anti-CD3 and anti-CD28 monoclonal antibodies for 5 d to harvest their supernatant.

### *C. albicans* killing and phagocytosis assay

FITC-labeled, heat-killed *C. albicans* yeast was prepared as described before and cocultured at a 1:3 ratio with CD14$^+$ monocytes for 2 h after preincubation of monocytes for 18 h in the presence or absence of T$_H$17 cell supernatants, which were selectively depleted for IL-1α by immunoabsorption (R&D Systems, Human IL-1 alpha/IL-1F1 DuoSet ELISA) or by CRISPR–Cas9 engineering of T$_H$17 cells[51]. Phagocytosis by monocytes was determined by the fraction of FITC-positive staining among CD14$^+$ monocytes using flow cytometry. In addition, live cell imaging was performed using the same experimental conditions with the Celldiscoverer 7 Live Cell Imaging System (Zeiss) and an integrated AxioCam 506 using Zeiss Zen Blue software at a constant temperature set to 37 °C and 5% CO$_2$. Four independent fields per well were imaged in 10-min intervals at ×10 (numerical aperture 0.35) magnification for a period of 5 h using the bright-field channel and green fluorescence filter (full videos provided as Supplementary Video 1).

### LDH assay

LDH activity was determined with a CytoTox 96 Non-Radioactive Cytotoxicity Assay (catalog no. G1780, Promega). In short, the supernatants were collected from cells stimulated for 24 h in RPMI-1640 medium without phenol red (Gibco). Relative LDH release was calculated as follows: LDH release (%) = 100 × (Experimental LDH release (OD$_{490}$) − Unstimulated control (OD$_{490}$))/(Lysis control (OD$_{490}$) − Unstimulated control (OD$_{490}$)) where OD$_{490}$ is the optical density at 490 nm.

### CRISPR–Cas9 KO cells

Candidate genes were depleted in sorted cells by using the Alt-R CRISPR–Cas9 system (Integrated DNA Technologies (IDT)) after activation with plate-bound anti-CD3 and anti-CD28 for 3 d. In brief, cripsr (cr)RNA and *trans*-activating crRNA (tracrRNA; both from IDT) were mixed at a 1:1 ratio, heated at 95 °C for 5 min and cooled to room temperature. Then, 44 µM crRNA:tracrRNA duplex was incubated at a 1:1 ratio with 36 µM Cas9 protein (IDT) for 20 min at room temperature to form a ribonucleoprotein (RNP) complex. A total of $(5–10) × 10^6$ activated T cells were washed with phosphate-buffered saline (PBS) and resuspended in 10 µl of R buffer (Neon transfection kit, Invitrogen). The RNP complex was delivered into cells with a Neon transfection system (10 µl of sample, 1,600 V, 10-ms pulse width, 3 pulses) (Thermo Fisher Scientific). The electroporated cells were then immediately incubated with RPMI-1640 complete medium with IL-2 (500 IU). The following crRNAs were used: GTCGGACTTTGTGAAATACG (*GSDME*), ACGCGCACCCACAAGCGGGA (*GSDMD*), GTCGGAGGAGATCATCACGC (*CAPN1*), GGCTTCGAAGACTTCACCGG (*CAPN2*), GGTAGTAGCAAC CAACGGGA (*IL1A*), CGGCTTGACTTGTCCATTAT (*CASP1*) and GTAT TACTGATATTGGTGGG (control sequence, nontargeted control (NTC)). Knockout (KO) efficiency was evaluated on day 7 after electroporation by immunoblotting or ELISA.

### Cytokine and transcription factor analyses

Intracellular cytokine and transcription factor staining was performed as described before with PMA and ionomycin restimulation in the presence of BFA[48]. Cells were stained with the following antibodies:

anti-IL-1α–phycoerythrin (PE) (catalog no. 364−3B3−14, 1∶50), anti-IL-1β-Alexa Fluor-647 (catalog no. JK1B-1, 1:50), anti-IL-4-BV421 (catalog no. MP4-25D2 5, 1:200), anti-IL-17A-PacificBlue (catalog no. BL168, 1:100), anti-IFN-γ–APC-Cy7 (catalog no. 4 S.B3, 1∶300) and anti-IL-10-PE-Cy7 (catalog no. JES3-9D7, 1:50) (all from BioLegend); anti-ROR-γt–APC (catalog no. AFKJS-9, eBioscience, 1:10), anti-Ki67-BV421 (BioLegend, 1:10) and anti-IL-1R1-PE (catalog no. FAB269P, R&D Systems, 1:20,). Then, they were analyzed with a BD LSRFortessa (BD Biosciences), a CytoFLEX Flow Cytometer (Beckman Coulter) or a MACSQuant analyzer (Miltenyi Biotec). Flow cytometry data were analyzed with FlowJo software (TreeStar) or Cytobank (Cytobank Inc.). The concentrations of cytokines in cell culture supernatants were measured by ELISA (Duoset ELISA kits from R&D Systems), Human Caspase-1 SimpleStep ELISA Kit (Abcam) or Luminex assays (eBioscience) according to standard protocols as indicated in the corresponding figure legends. Counting beads (CountBright Absolute Counting Beads, Thermo Fisher Scientific) were used to normalize for cell numbers if analysis of cumulative supernatants obtained from 5-d cell cultures was performed.

## IL-1α secretion assay

The design of the IL-1α secretion assay was adapted based on a previous report[50]. $T_H17$ cells ($1 \times 10^6$) were stained with 1 mg ml$^{-1}$ of sulfo-NHS-LC-biotin (catalog no. ab145611, Abcam), incubated for 30 min at room temperature and then washed 3× with PBS (pH 8) supplemented with 100 mM glycine. The final washing of cells was performed with PBS supplemented with 0.5% bovine serum albumin. Cell surface biotinylation was validated with PE-labeled streptavidin (catalog no. 554061, BD Pharmingen). Purified anti-human IL-1α antibodies (AF-200-NA, R&D) were labeled with streptavidin using a Lightning-Link Streptavidin Conjugation kit (catalog no. ab102921, Abcam). For cytokine secretion, cells were stimulated with anti-CD3 and anti-CD28 for 72 h. The cells were collected and labeled with streptavidin-IL-1α and incubated for 24 h on the MACSmix tube rotator (Miltenyi Biotec). Recombinant IL-1α (Miltenyi Biotec) was added as a positive control. The cells were then stained with a PE-labeled IL-1α antibody (clone 364-3B3-14, BioLegend, 1:50).

## Imaging flow cytometry

Data acquisition was performed using an ImageStream X Mk II imaging flow cytometer (AMNIS, MERCK Millipore) equipped with INSPIRE software. Briefly, a ×60 magnification was used to acquire images with a minimum of 5,000 cells per sample. The following antibodies were used: anti-ASC-PE (catalog no. HASC-71, BioLegend, 1:50), anti-CD3-APC (1:100) or anti-CD3-FITC (catalog no. UCTH1, BioLegend, 1:150) and anti-NLRP3-APC (catalog no. REA668, Miltenyi Biotec, 1:50). Data analysis was performed using IDEAS 6.0 software. A compensation matrix was generated using single-stained cells. Cells that were not in the field of focus, clumped cells and debris were excluded. IDEAS software was used to design masks to define the properties of the spots. For ASC spots, a size of 1–4 μm and a signal:background ratio of 3.0–5.0 were chosen. The mask was trained on at least ten different images with spot-like structures clearly visible to refine the cutoff for the signal:background ratio. From this 'spot mask', the diameter of the mask was measured and ASC spots in the range of 1–4 μm were considered to be true spots.

## Gene expression analysis

For analysis of individual gene expression, a high-capacity complementary DNA reverse transcription kit (Applied Biosystems) was used for cDNA synthesis according to the manufacturer's protocol. The transcripts were quantified by RT−qPCR with predesigned TaqMan Gene Expression Assays (*IL1A*, catalog no. HS00174092-m1; *IL1B*, catalog no. Hs01555410_m1; *NLRP3*, catalog no. Hs00918082_m1; *CASP1*, catalog no. Hs00354836_m1, *CAPN2*, catalog no. Hs00965097_m1; *GSDMD*, catalog no. Hs00986739_g1; *DFNA5*, catalog no. Hs00903185_m1; and

18 S, catalog no. Hs03928990_g1) and reagents (Applied Biosystems). The mRNA abundance was normalized to the amount of 18S ribosomal RNA and expressed as arbitrary units (a.u.).

For microarray analysis (Gene Expression Omnibus (GEO), accession no. GSE214519), total RNA was extracted from pro- and anti-inflammatory $T_H17$ cells that were obtained on restimulation with and without IL-1β, respectively, as described previously[4,6], using an RNA MiniPrep kit (Zymo Research), and hybridized to the Human Genome U133 Plus 2 platform (Affymetrix) according to a whole-transcriptome Pico Kit. The raw signals were processed with the affy R package[52] and normalized using the robust multiarray average expression measure with background correction and crosschip quantile normalization. The limma R package[53] was applied to identify DEGs using linear model fitting, with adjustment for differences between biological replicates. Empirical Bayes statistics was used for the moderation of s.e.m. and P values were adjusted using the Benjamini–Hochberg method. A false discovery rate (FDR) <0.05 and a fold-change cutoff of 2 were used to define the DEGs. For GSEA, the top 50 upregulated genes (proinflammatory, 44 significant DEGs) and the top 50 downregulated genes (anti-inflammatory, 41 significant DEGs) genes from a transcriptomic comparison of IL-10$^+$ and IL-10$^-$ $T_H17$ cell clones from a public dataset[7] were selected as gene sets and utilized to interrogate the $T_H17$ cell transcriptomes (microarray) after cell stimulation in the presence or absence of IL-1β.

For next-generation mRNA-seq, resting T cell clones categorized as IL-1α$^+$ (>30% IL-1α expression) and IL-1α$^-$ (0% IL-1α expression) were restimulated with PMA and ionomycin (both from Sigma-Aldrich) for 3 h (GEO accession no. GSE214475). A total amount of 1 μg of RNA per sample was used as the input material for the RNA sample preparations. Sequencing libraries were generated using an NEBNext Ultra RNA Library Prep Kit for Illumina (New England Biolabs (NEB)) following the manufacturer's recommendations and index codes were added to attribute sequences to each sample. The mRNA was purified from total RNA using poly(T) oligo-attached magnetic beads. Fragmentation was carried out by using divalent cations under an elevated temperature in NEB Next First Strand Synthesis Reaction Buffer (5×) or sonication with a Diagenode Bioruptor Pico for fragmentation of RNA strands. First-strand cDNA was synthesized using random hexamer primers and M-MuL V Reverse Transcriptase (RNase H-). Second-strand cDNA synthesis was subsequently performed using DNA polymerase I and RNase H. The remaining overhangs were converted into blunt ends via exonuclease/polymerase activity. After adenylation of the 3′-ends of the DNA fragments, NEBNext adapters with a hairpin loop structure were ligated to prepare the fragments for hybridization. To preferentially select cDNA fragments 150−200 bp in length, the library fragments were purified with an AMPure XP system (Beckman Coulter). Then, 3 μl of USER Enzyme (NEB) was used with size-selected, adapter-ligated cDNA at 37 °C for 15 min followed by 5 min at 95 °C before PCR. Then PCR was performed with Phusion High-Fidelity DNA Polymerase, Universal PCR primers and Index (X) Primer. Finally, the PCR products were purified (AMPure XP system) and library quality was assessed on an Agilent Bioanalyzer 2100 system. Clustering of the index-coded samples was performed on a cBot Cluster Generation System using a PE Cluster Kit cBot-HS (Illumina) according to the manufacturer's instructions. After cluster generation, the libraries were sequenced on an Illumina platform and paired-end reads were generated (Novogene).

For comparison of GSDME-intact (NTC) and CRISPR–Cas9-deficient (KO) $T_H17$ cells (GEO accession no. GSE214292), three matched blood samples were analyzed by mRNA-seq. Quality control was performed using FastQC (v.0.11.9)[54]. STAR (v.2.7.5a) was used with the 'quantMode GeneCounts' option and the other parameters set to the default values to map the reads to the human reference genome (GRCh38; GenBank patch release 13) and count the reads mapped to each gene. Transcriptome annotation was obtained from GENCODE (v.34)[55]. Differential expression analysis was performed for protein-coding genes (retrieved from BioMart Ensembl Genes release 104)[56] using the DESeq2 (v.1.28.1) R package[57]. Specifically, the DESeq()

function was applied with the default parameters to compare the expression levels between KO and NTC samples while controlling for the donor. Significant DEGs were defined as genes with an FDR-adjusted $P$ value ≤0.05 and fold-change ≥2 or ≤0.5. To investigate whether differences in expression were associated with cell death, genes ($n = 19,622$) were sorted in decreasing order according to the absolute value of their $\log_2$(fold-change). Different numbers of genes (50, 100, 500, 1,000, 2,000 and 2,500) from the top of this list were then selected to carry out functional analysis with the DAVID API (v.2021)[58], focusing on the categories 'KEGG_PATHWAY' and 'GO_TERM_BP_FAT', with an FDR cutoff of 100%, a count threshold (minimum gene counts belonging to an annotation term) of 0 and an EASE score threshold of 1. Annotation terms with a minimum gene count of at least 5 and an FDR of 5% in ≥3 of the gene sets were deemed to be associated with expression differences between KO and NTC samples. Gene expression values were obtained using the rlogTransformation() function of the DESeq object. Principal component analysis was performed on gene expression values with the prcomp() R function.

For scRNA-seq, a library of human $T_H17$ cells that were sorted ex vivo as CCR6$^+$CCR4$^+$CXCR3$^-$ memory $T_H$ cells using FACS and then stimulated with anti-CD3 and anti-CD28 monoclonal antibodies for 4 d (2 d plate-bound) was constructed with Chromium Next GEM Single Cell 5' Reagents v.2 (Dual Index) (10x Genomics, Inc.) (GEO accession no. GSE214444). The library was sequenced on an Illumina NovaSeq 6000 Sequencing System according to the manufacturer's instructions, with 150-bp, paired-end, dual-indexing sequencing (sequencing depth: 20,000 read pairs per cell). Read alignment and gene counting of the single-cell datasets were performed with CellRanger v.6.1.1 (10x Genomics, Inc.), using the default parameters and the prebuilt human reference 2020-A (10x Genomics, Inc.) based on Ensembl GRCh38 release 98. The output filtered data were first processed with the Python package scanpy v.1.7.2 and also analyzed with the R package Seurat v.4.0.4. The total count was normalized to 10,000 reads per cell. Each gene was scaled to unit variance and values exceeding the s.d. by ten were clipped. A KNN nearest neighbor graph was constructed with a size of ten local neighboring data points. UMAP with the default settings was applied for dimensionality reduction. Clusters were identified by running the Leiden algorithm with a cluster resolution of 0.4. Differential gene expression analysis was performed using the FindAllMarkers function with the non-parametric Wilcoxon's rank-sum test from the R package Seurat v.4.0.4.

Pro- and anti-inflammatory gene sets were established from a public dataset after transcriptomic comparison of IL-10$^-$ versus IL-10$^+$ $T_H17$ cell clones[7]. For both gene sets an average expression score was calculated for each individual cell using the addModuleScore method from the R package Seurat. To compare the scores between groups of cells, Wilcoxon's rank-sum test as implemented in the R package stats was used. Similar comparisons were performed with GO terms taken from the Molecular Signatures Database (MSigDB).

### Immunoblotting
Cells were lysed in radioimmunoprecipitation buffer (50 mM Tris, 150 mM NaCl, 1 mM EDTA, 0.1% NP-40, pH 7.5) containing protease inhibitor (Roche) and PhosphoSTOP Easypack (Roche). The protein concentrations of cell lysates were determined with a Pierce BCA Protein Assay Kit (Thermo Fisher Scientific). Total protein (20-40 mg) was boiled with 4× Laemmli sample buffer (BioRad Laboratories) containing 355 mM 2-mercaptoethanol (Thermo Fisher Scientific) at 99 °C for 10 min. The supernatants and lysates were separated by sodium dodecylsulfate–polyacrylamide gel electrophoresis and transferred to a poly(vinylidene) membrane (BioRad Laboratories) by using a Mini-Protean system (BioRad Laboratories) according to the manufacturer's protocol. The following primary antibodies were used for immunoblotting: mouse anti-human caspase-8 (Cell Signaling Technology), rabbit anti-human caspase-1 (Cell Signaling Technology), rabbit anti-human IL-1α (Abcam), mouse anti-human

glyceraldehyde-2-phosphate dehydrogenase (GAPDH; Merck Millipore), mouse anti-human β-actin (Cell Signaling Technology) and rabbit anti-human GSDME (Abcam), rabbit anti-human caspase-3 (Cell Signaling Technology), mouse anti-human caspase-8 (Cell Signaling Technology), rabbit anti-human GSDMD (Cell Signaling Technology), rabbit anti-human cleaved GSDMD (Cell Signaling Technology) and rabbit anti-NLRP3 (Cell Signaling Technology). Horseradish peroxidase (HRP)-conjugated anti-mouse and anti-rabbit immunoglobulin G antibodies (Cell Signaling Technology) were used as secondary antibodies. The immunoreactive bands were detected by Pierce ECL Western Blotting Substrate or SuperSignal West Femto Maximum Sensitivity Substrate (both from Thermo Fisher Scientific). The chemiluminescence signals were recorded with an Odyssey Imaging system (LI-COR Biosciences) and analyzed on Image Studio Lite (LI-COR Biosciences). Image contrast was enhanced in a linear fashion when necessary. Protein lysates were also prepared for automated immunoblotting using a Jess System (ProteinSimple) according to the manufacturer's instructions. The following primary and secondary antibodies were used: recombinant rabbit anti-GSDME-N-terminal (Abcam), rabbit anti-GSDMD (Cell Signaling Technology), rabbit anti-caspase-3 (Cell Signaling Technology), mouse anti-caspase-8 (Cell Signaling Technology), mouse anti-ASC (Santa Cruz Biotechnology, B-3), mouse anti-NLRP3 (Novus Biologicals, catalog no. 25N10E9), rabbit recombinant anti-sodium potassium ATPase antibody (Abcam), mouse anti-GAPDH (Sigma-Aldrich) and mouse anti-β-actin (Cell Signaling Technology) primary antibodies, an anti-mouse HRP-linked secondary antibody (ProteinSimple) and an anti-rabbit HRP-linked secondary antibody (ProteinSimple).

### Extraction of plasma membrane proteins
Plasma membrane proteins were fractionated with a plasma membrane protein kit (Abcam) according to the manufacturer's protocol. In short, $(0.5–1) \times 10^7$ cells were collected, homogenized in an ice-cold Dounce homogenizer (Bellco Glass Inc.) and centrifuged at 700$g$ for 10 min. The supernatants were collected and centrifuged at 10,000$g$ for 30 min. The supernatants were collected as the cytosol fraction. The pellets were used for further extraction of plasma membrane proteins. The purified plasma membrane proteins were enriched in the upper phase solution (Abcam), whereas the lower phase solution contained the cellular organelle membranes. The cytosolic fraction was concentrated with Amicon Ultra Centrifugal Filters (10k). The lysates generated from different fractions were boiled with 4× Laemmli sample buffer (BioRad Laboratories) and subjected to immunoblotting. A rabbit anti-sodium–potassium ATPase antibody (Abcam) was used as a positive control for plasma membrane proteins.

### Calpain activity assay
Cells were harvested and washed with cold PBS. Cells were then resuspended in Extraction Buffer (Abcam) and centrifuged at 13,000$g$ for 5 min. The protein concentration in the supernatants was measured with a Pierce BCA Protein Assay Kit (Thermo Fisher Scientific). Total lysate protein, 40 μg, was used to perform a calpain activity assay (Abcam) following the manufacturer's instructions. A total of 1–2 μl of active calpain (Abcam) was used as a positive control. Calpain inhibitor Z-LLY-FMK (Abcam), 1 μl, was used as a negative control. The lysates and calpain substrate were incubated at 37 °C for 60 min. The fluorometric signal was detected at excitation/emission wavelengths of 400/505 nm with a CLARIOstar plate reader (BMG-Labtech).

### FLICA assays
A FAM-FLICA Caspase-1 Assay Kit and Caspase-8 Assay Kit (Immuno-Chemistry Technologies, LLC) were used to evaluate the presence of catalytically active forms of caspase-1 p10 and p12 and caspase-8 according to the manufacturer's instructions. Cells were incubated with 30× FAM-VAD-FMK for 30 min at 37 °C, then washed twice with

the apoptosis wash buffer (ImmunoChemistry Technologies, LLC) and analyzed by flow cytometry on a Cytoflex instrument.

## Statistical analysis

The use of the statistical tests is indicated in the respective figure legends, with the error bars indicating the mean ± s.e.m. $P$ values ≤0.05 were considered to indicate significance. Analyses were performed using GraphPad Prism v.9 or R v.4.1. No statistical methods were used to predetermine sample sizes but our sample sizes are similar to those reported in previous publications[7]. Data distribution was assumed to be normal, but this was not formally tested. No randomization was performed. No data points were excluded.

## Reporting summary

Further information on research design is available in the Nature Portfolio Reporting Summary linked to this article.

## Data availability

Transcriptomic datasets for scRNA-seq (raw scRNA-seq fastq files, count matrix, gene and barcode files) and bulk RNA-seq (raw fastq files, raw and MRN count matrices) have been deposited in the GEO under the accession nos. GSE214519, GSE214292, GSE214475 and GSE214444. Source data are provided with this paper.

## Code availability

The original code used to analyze the presented data are available at GitHub (https://github.com/thisisalbert/christina_zielinski_code_repository) and Zenodo (https://doi.org/10.5281/zenodo.7257536).

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

## Acknowledgements

We thank all past and present members of the Zielinski laboratory for fruitful discussions and technical support, in particular R. Noster, A. Burrell, F. Laudisi and S.-H. Park for technical and experimental help. We thank L. Richter and J. Klein (Core Facility Flow Cytometry, Biomedical Center, Munich) for support with imaging flow cytometry, M. Schmidt-Supprian for support with CRISPR–Cas9 gene editing and P. Wehner for support with live cell imaging analysis. This work was funded by the Deutsche Forschungsgemeinschaft (German Research Foundation) through grant no. SFB 1054 (project ID 210592381, to C.E.Z.), TRR/SFB 124 (project ID 210879364, to C.E.Z. and M.S.G.), Leibniz Center for Photonics in Infection Research (grant no. LPI-BT6, to C.E.Z.), GRK 2606 (project ID 423813989, to O.G.), Germany's Excellence Strategy (Balance of the Microverse, to C.E.Z), Emmy Noether Program (project no. 434385622/GR 5617/1-1, to M.S.G.), the German Center of Infection Research (to C.E.Z.), Carl-Zeiss Stiftung (to C.E.Z.) and by the European Research Council (grant nos. 337689 and 966687, to O.G.).

## Author contributions

Y.Y.C., A.P., D.F., T.S. and T.C. designed and performed the experiments and analyzed the data. A.G.P., M.A., S.F.S. and L.L. performed bioinformatic analyses. G.P. and L.T. supervised the bioinformatic analyses. D.K. and O.G. supervised experiments and helped to write the manuscript. M.G. provided FITC-labeled *C. albicans* and helped with live cell imaging using CellDiscoverer 7. M.W. and F.W. designed and performed experiments. S.J.V. provided JIA patient samples. A.P. analyzed the samples. C.E.Z. designed and supervised the project, analyzed the data and wrote the manuscript.

## Competing interests

A patent application has been filed by the authors. Otherwise, the authors declare no competing interests.

## Additional information

**Extended data** is available for this paper at https://doi.org/10.1038/s41590-022-01386-w.

**Correspondence and requests for materials** should be addressed to Christina E. Zielinski.

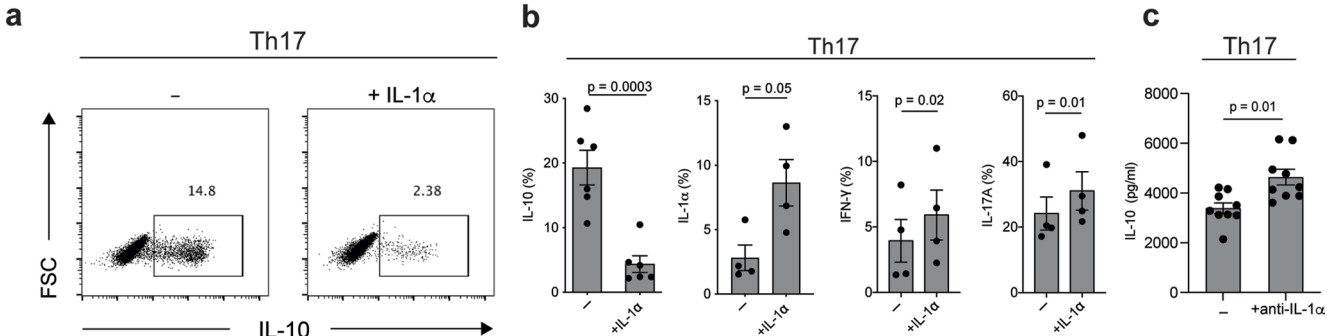

**Extended Data Fig. 1 | Autocrine IL-1α production by Th17 cells enforces a pathogenic Th17 cell identity. a**, Th17 cells were stimulated with anti-CD3 and anti-CD28 mAbs for 5 days before intracellular cytokine staining and flow cytometry following PMA and ionomycin restimulation. Representative experiment. **b**, Cumulative data for (a). Two-sided paired t-test. **c**, ELISA of supernatants from Th17 cells stimulated for 5 days with anti-CD3 and anti-CD28 mAbs. Two-sided paired t-test.

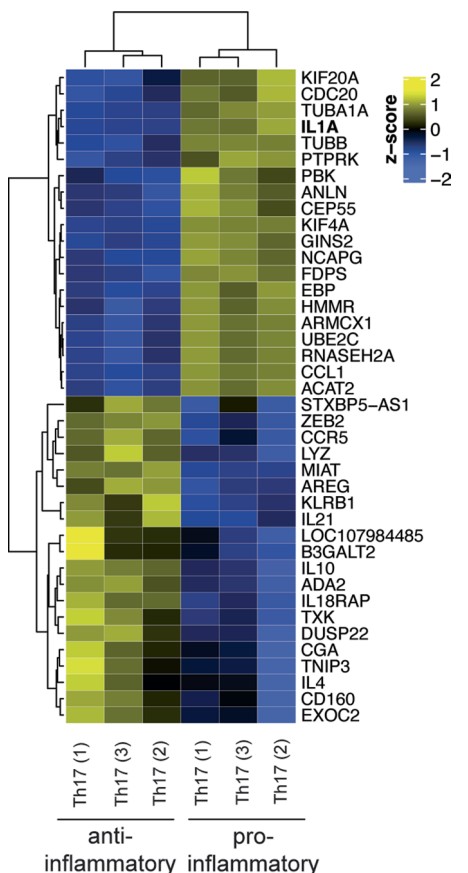

**Extended Data Fig. 2 | Differential gene expression between pro- and anti-inflammatory human Th17 cells.** Shown is a heatmap displaying the top 20 up- and downregulated genes (by order of significance) of the transcriptome following stimulation of human Th17 cells with anti-CD3 and anti-CD28mAbs for 5 days in the presence (proinflammatory) or absence (anti-inflammatory) of exogenous IL-1β. The samples and genes were clustered according to the pattern of expression using k-means clustering.

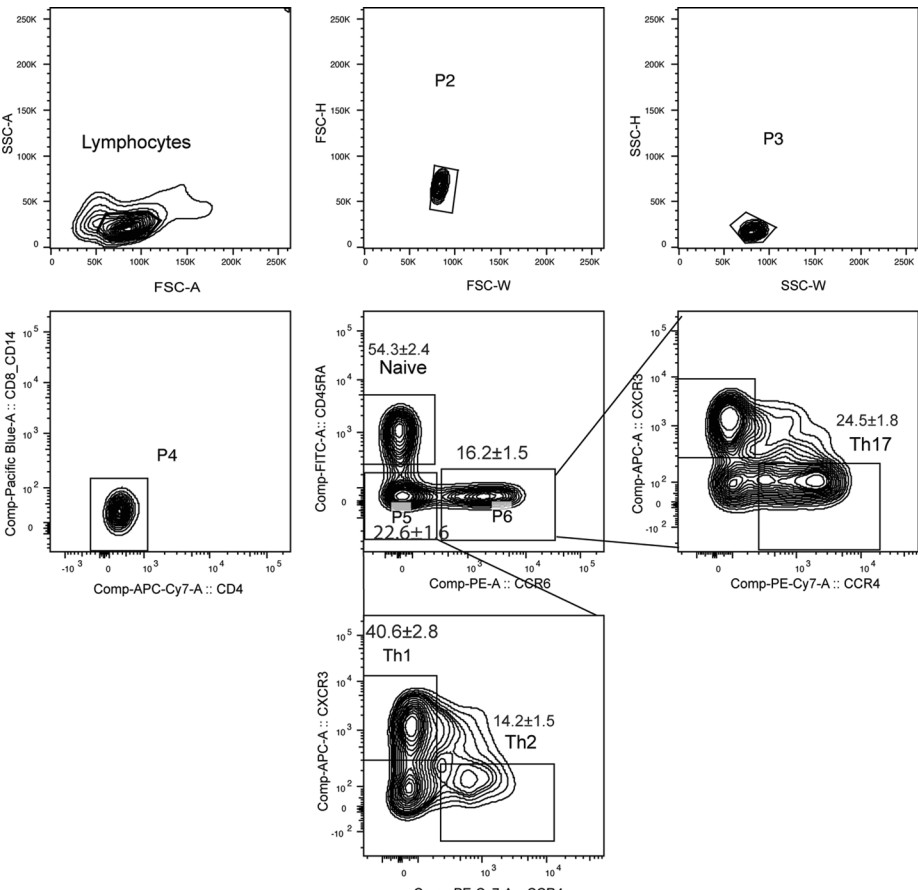

**Extended Data Fig. 3 | Gating strategy for the isolation of Th cell subsets from the blood.** Fresh PBMCs from healthy donors were enriched for CD4 expression by MACS before FACS for CD45RA⁻ Th cell subsets according to their differential expression of the chemokine receptor markers CCR6, CCR4 and CXCR3. A representative gating strategy and the percentage of Th-cell subsets in the respective gates for 33 healthy donors are shown (mean ± SEM).

**a**

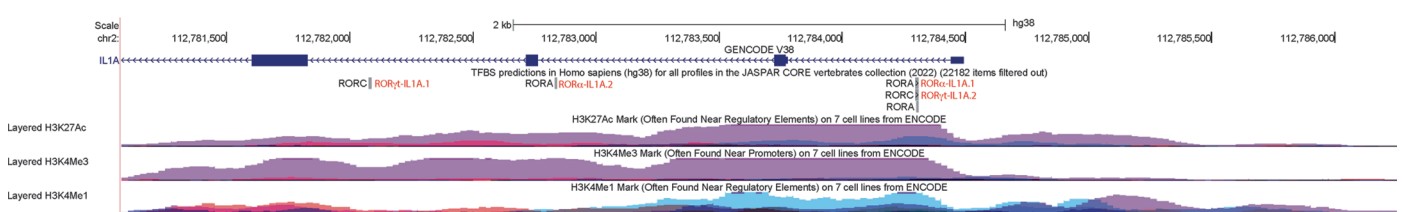

**b**

### RORα consensus binding site 1

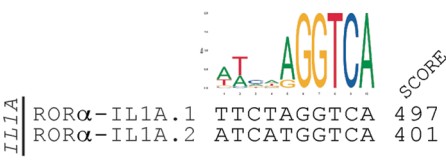

### RORα consensus binding site 2

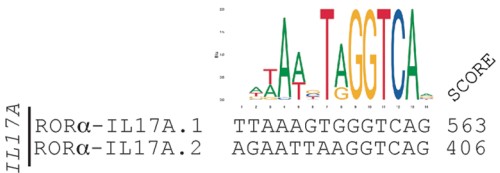

### RORγt consensus binding site

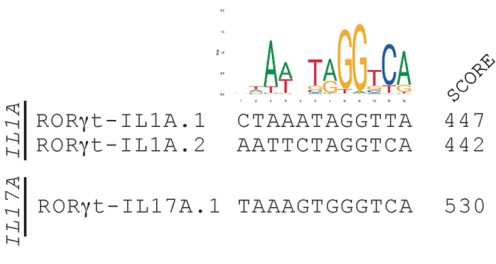

**Extended Data Fig. 4 | Consensus binding sites for the Th17 master transcription factors ROR-γt and RORα in the *IL1A* intronic enhancer region.** **a**, Locations of RORα and ROR-γt consensus binding sites in the regulatory region of *IL1A*. Binding sites were predicted using the JASPAR2022 database (Castro-Mondragon JA et al., Nucleic Acids Res 2022) and annotated with the UCSC genome browser (assembly hg38) (Kent WJ et al., Genome Res 2002). The quality cutoff was set at 400 (this score indicates a p value of p = 1*10-(score/100)). ENCODE tracks for histone marks were used to indicate the bona fide *IL1A* regulatory region (chr2:112,781,072-112,786,251) (Rosenbloom KR et al., Nucleic Acids Res 2013). **b**, Alignment of the RORα and RORγt consensus binding motif with the predicted binding sites in the *IL1A* enhancer RORα-IL1A.1 (chr2:112784300-112784309), RORα-IL1A.2 (chr2:112782834-112782843); RORγt-IL1A.1 (chr2:112,782,077-112,782,088); RORγt-IL1A.2 (chr2:112,784,298-112,784,309) and established binding sites in the *IL17A* enhancer RORα-IL17A.1 (chr6:52181117-52181130), RORα-IL17A.2 (chr6:52180979-52180992) and RORγt-IL17A.1 (chr6:52181118-52181129) (Wang X et al., Immunity. 2012). Scores for individual sites are indicated.

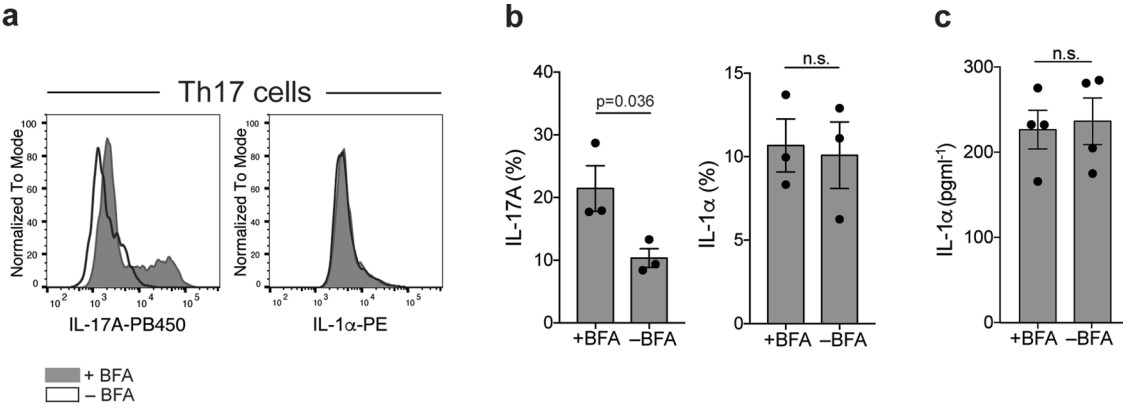

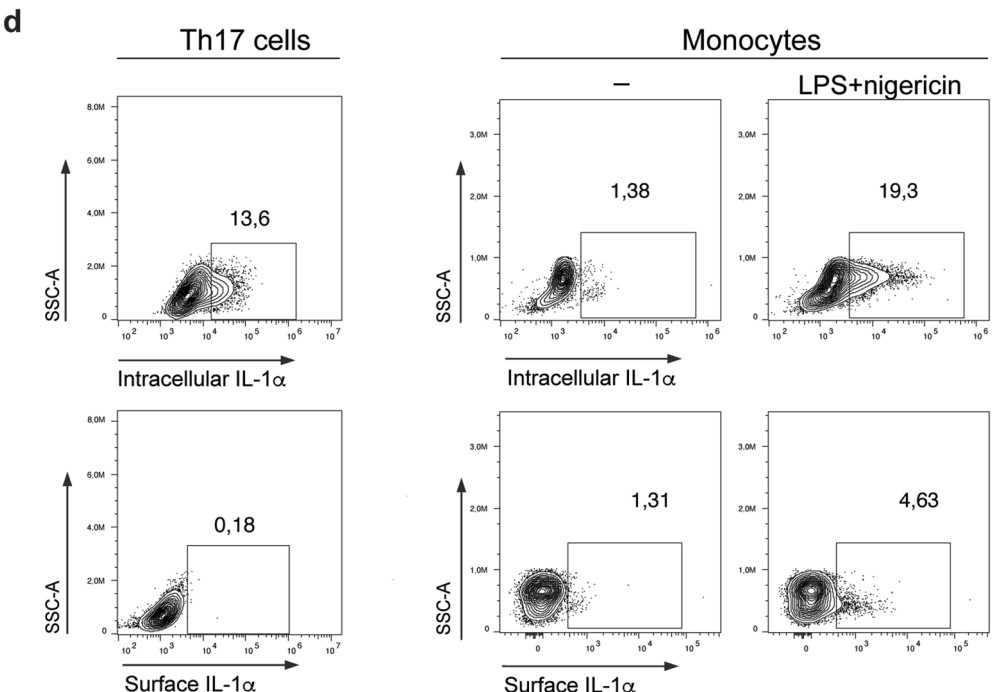

**Extended Data Fig. 5 | IL-1α is secreted by an unconventional pathway in human Th17 cells and is not expressed on the cell surface. a**, Th17 cells were stimulated for 5 days with anti-CD3 and anti-CD28 mAbs (48 h plate-bound) before intracellular cytokine staining and flow cytometric analysis on day 5 following PMA and ionomycin restimulation. Representative experiment. **b**, Cumulative data, Two-sided paired t-test. Each circle indicates an individual healthy blood donor (n = 3 examined over 2 independent experiments). Data are presented as mean ± SEM. **c**, ELISA of 24 h cumulative supernatants from day 4 to day 5 in the presence or absence of BFA. n.s., not significant. Two-sided paired t-test. Each circle indicates an individual healthy blood donor (n = 4 biologically independent samples examined over 2 independent experiments). Data are presented as mean ± SEM. **d**, Th17 cells were stimulated and analyzed as in (a). Monocytes were stimulated in the presence and absence of LPS (24 h) and nigericin (last 30 min) and analyzed by intracellular and surface staining and flow cytometry. Representative experiment (n = 3 biologically independent samples).

**a**

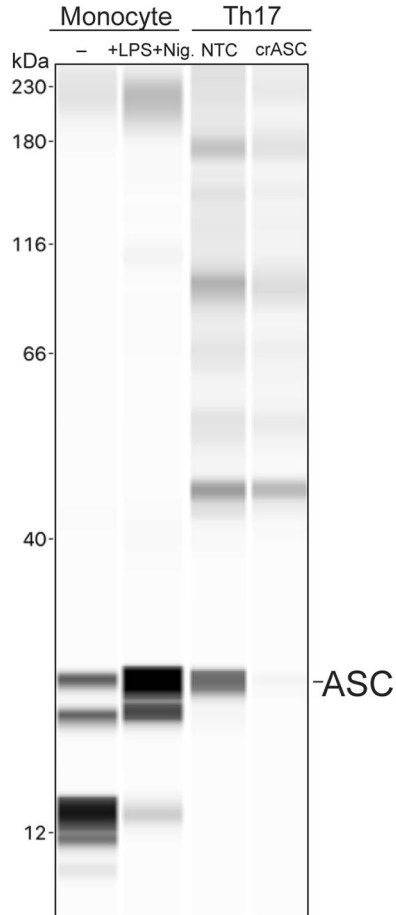

**b**

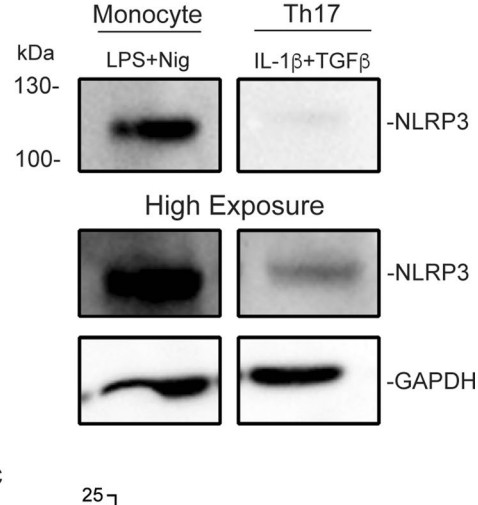

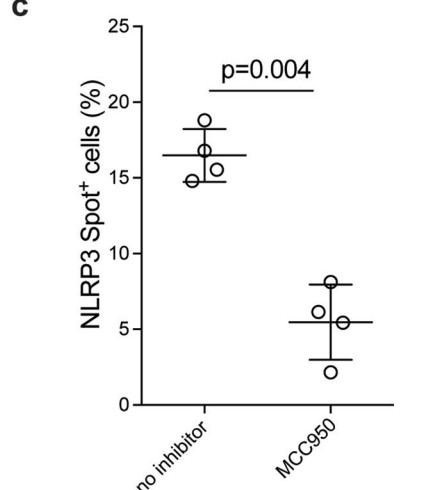

**c**

**Extended Data Fig. 6 | Human Th17 cells express the inflammasome components ASC and NLRP3. a**, Lane views of electropherograms obtained with the Jess Simple Western System (ProteinSimple). Monocytes were MACS-isolated with CD14 beads and stimulated in the presence or absence of LPS (24 h) and nigericin (Nig.) for 30 min before cell lysis. Th17 cells were FACS-sorted ex vivo from the blood of healthy donors, stimulated for 48 h and transfected with RNP containing nontargeted control (NTC) or crASC. Cell lysates were collected after 7 days. The experiment was repeated independently 3 times with similar results. **b**, Western blot analysis. Cell culture lysates derived from Th17 cells and monocytes. Th17 cells were sorted as in (a) and stimulated with anti-CD3 and anti-CD28 mAbs (48 h plate-bound) for 5 days. Monocytes were isolated and stimulated as in (a). The experiment was repeated independently 2 times with similar results. **c**, NLRP3-speck formation assessed by ImageStream in Th17 cells stimulated for 5 days with anti-CD3 and anti-CD28 mAbs. Each circle indicates an individual healthy blood donor and experiment. Two-sided paired t-test. Data are presented as mean ± SEM. n = 4 biologically independent samples examined over 2 independent experiments.

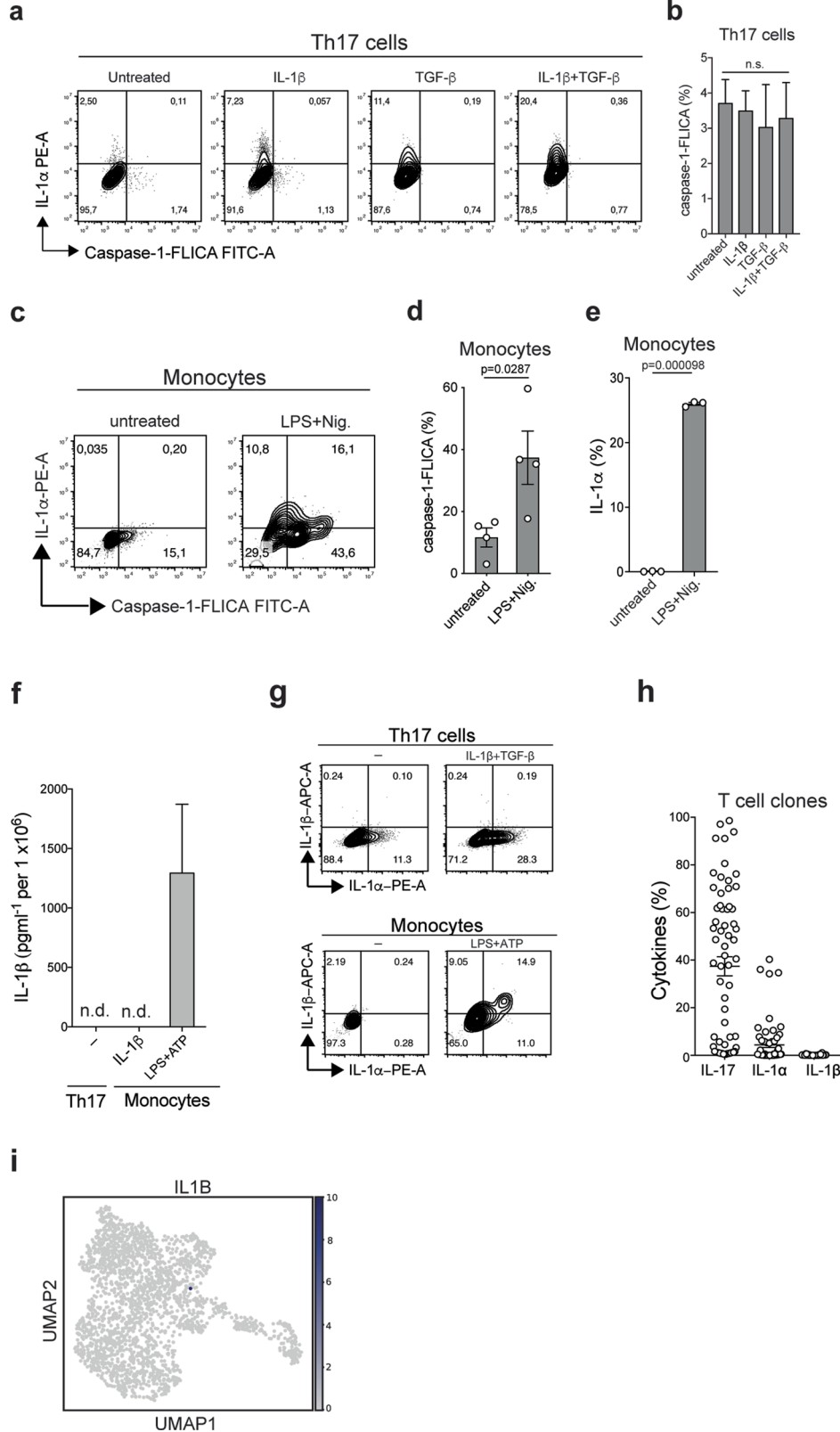

**Extended Data Fig. 7 | See next page for caption.**

**Extended Data Fig. 7 | IL-1α expression is neither regulated by active caspase-1 in human Th17 cells nor coregulated with IL-1β, in contrast to the situation in monocytes. a**, Th17 cells were isolated *ex vivo* by FACS-sorting and stimulated for 5 days with anti-CD3 and anti-CD28 mAbs (48 h plate-bound) in the presence or absence of the indicated cytokines before flow cytometric analysis of intracellular IL-1α and active caspase-1 after PMA and ionomycin restimulation. The results of one representative experiment are shown. **b**, Cumulative data for active caspase-1 as shown in (a). One-way ANOVA. n = 3 biologically independent samples examined over 3 independent experiments. Data are presented as mean ± SEM. **c**, Intracellular staining and flow cytometry of monocytes following 24 h stimulation with LPS and 30 min with nigericin (Nig.). The results shown are from one representative experiment. **d, e**, cumulative data of (d), Two-sided paired t-test. Each circle indicates an individual healthy blood donor (n = 3-4 biologically independent samples examined over 3 independent experiment). Data are presented as mean ± SEM. **f**, ELISA of Th17 cells, which were stimulated as in (a) and of monocytes were stimulated with LPS (24 h) and ATP (30 min). n = 3 biologically independent samples examined over 2 independent experiments. Data are presented as mean ± SEM. **g**, Flow cytometric analysis of cells as in (g).**h**, Intracellular cytokine staining and flow cytometric analysis of T-cell clones generated from the Th17-cell subset. Each circle indicates an individual T cell clone (n = 64). Data are presented as mean ± SEM. **i**, scRNA-seq and UMAP showing IL-1β expression in human Th17 cells stimulated for 5 days with anti-CD3 and anti-CD28 mAbs.

**a**

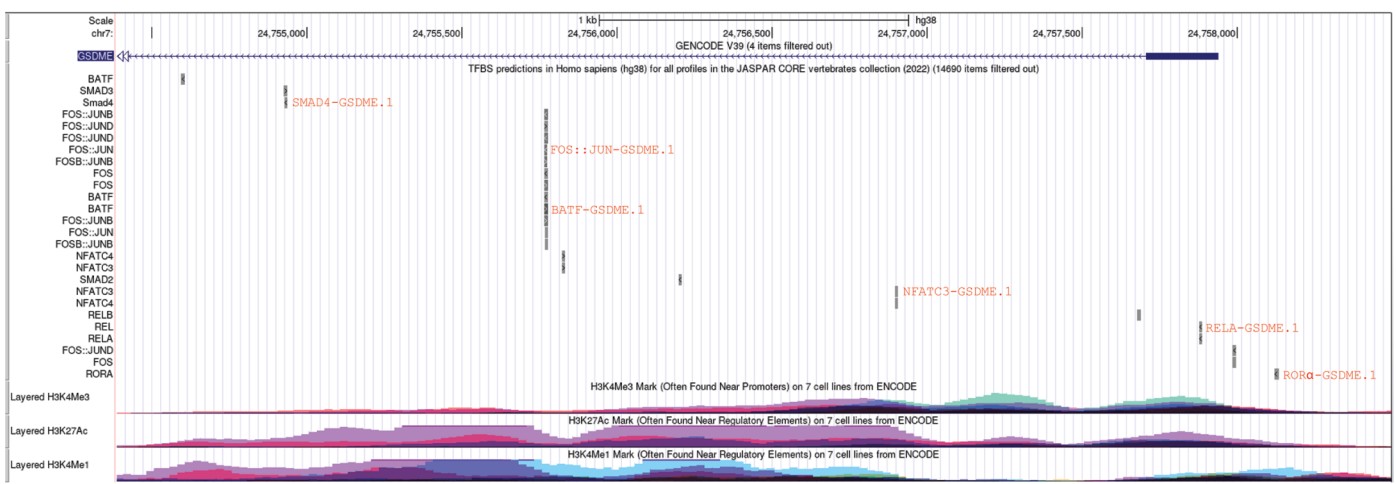

**b**

NFATC3 consensus binding site

NFATC3-GSDME.1 ACGGAAAAA 441

BATF consensus binding site

BATF-GSDME.1 TATGACTCATT 614

RELA consensus binding site

RELA-GSDME.1 TGGACTTTCC 474

SMAD4 consensus binding site

SMAD4-GSDME.1 TGTCTAGA 481

FOS::JUN consensus binding site

FOS:JUN-GSDME.1 ATGAGTCATA 571

RORα consensus binding site 2

RORα-GSDME.1 AATATTTAAGTCAG 448

**Extended Data Fig. 8 | The GSDME promoter and intronic enhancer region display consensus binding sites for TCR-induced transcription factors as well as for Th17 master regulators. a**, Locations of consensus binding sites of TCR-induced (NFATC-family, NF-κB-family, AP1-family and BATF) and TGFβ-induced (SMAD 2,3,4) transcription factors as well as the Th17 transcription factors RORα and BATF in the regulatory region of GSDME (chr7:24,754,387-24,758,496). The regulatory region and the consensus binding sites were defined and annotated as in Extended Data Fig.6. **b**, The consensus binding motives were aligned with predicted binding sites in the GSDME promotor/enhancer region. The binding sites with the highest score for each family are indicated. NFATC3-GSDME.1 (chr7:24756897-24756905), RELA-GSDME.1 (chr7:24757878-24757887), FOS::JUN (chr7:24755766-24755775), BATF-GSDME.1 (chr7:24755766-24755776), SMAD4 (chr7:24754928-24754935), RORα (chr7:24758120-24758133).

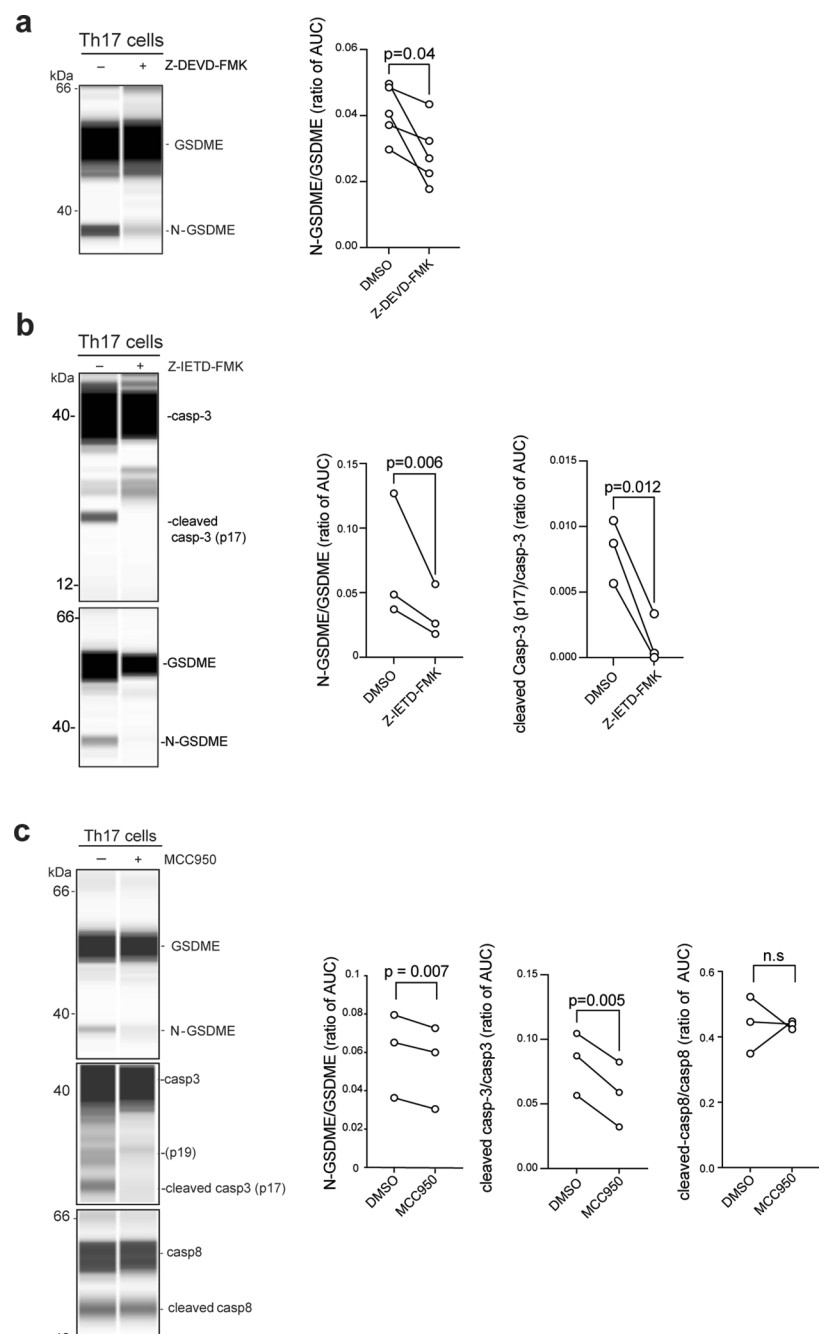

**Extended Data Fig. 9 | Pharmacological inhibition shows that GSDME pore formation is regulated by each component of the NLRP3 inflammasome – casp8 – casp3 cleavage cascade.** The lane view of electropherograms obtained with the Jess Simple Western System (ProteinSimple) is shown. Th17 cells were sorted according to the differential expression of chemokine receptors and stimulated for 5 days with anti-CD3 and anti-CD28 mAbs (48 h plate-bound) in the presence or absence of specific inhibitors for caspase-3 **(a)**, caspase-8 **(b)** or the NLRP3 inflammasome **(c)**. The inhibitors were added on day 3 of the 5-day culture period. The AUC was calculated with the Jess software Compass for SW as the ratio of the cleaved versus noncleaved inhibitor target proteins after normalization to total protein. Two-sided paired t-tests.

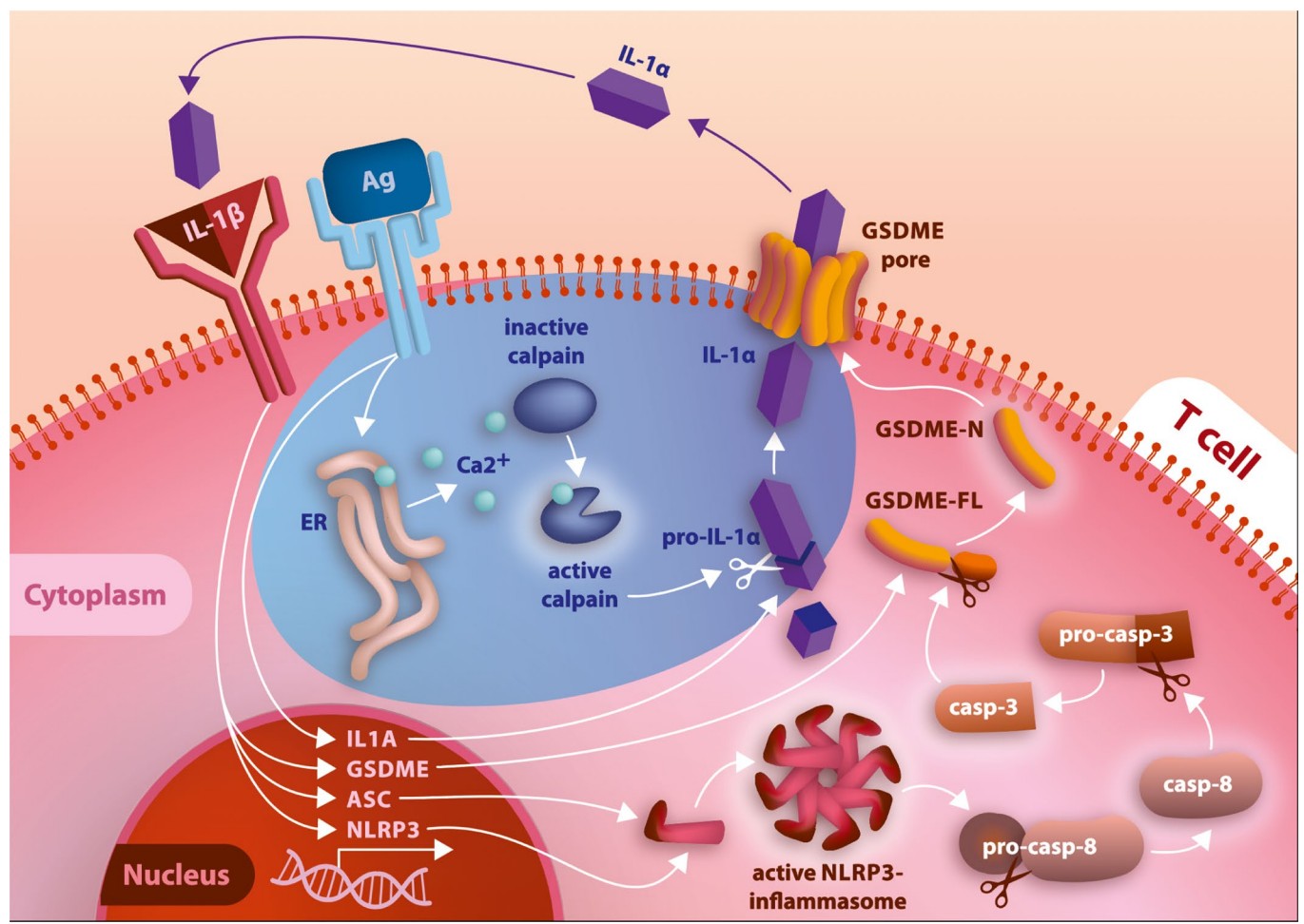

**Extended Data Fig. 10 | Mechanism of IL-1α production by human Th17 cells: Graphical summary.** The graphic was created with a commercial license from Adobe.

# Reporting Summary

## Statistics

For all statistical analyses, confirm that the following items are present in the figure legend, table legend, main text, or Methods section.

| n/a | Confirmed | |
|---|---|---|
| ☐ | ☒ | The exact sample size (*n*) for each experimental group/condition, given as a discrete number and unit of measurement |
| ☐ | ☒ | A statement on whether measurements were taken from distinct samples or whether the same sample was measured repeatedly |
| ☐ | ☒ | The statistical test(s) used AND whether they are one- or two-sided *Only common tests should be described solely by name; describe more complex techniques in the Methods section.* |
| ☒ | ☐ | A description of all covariates tested |
| ☒ | ☐ | A description of any assumptions or corrections, such as tests of normality and adjustment for multiple comparisons |
| ☐ | ☒ | A full description of the statistical parameters including central tendency (e.g. means) or other basic estimates (e.g. regression coefficient) AND variation (e.g. standard deviation) or associated estimates of uncertainty (e.g. confidence intervals) |
| ☐ | ☒ | For null hypothesis testing, the test statistic (e.g. *F*, *t*, *r*) with confidence intervals, effect sizes, degrees of freedom and *P* value noted *Give P values as exact values whenever suitable.* |
| ☒ | ☐ | For Bayesian analysis, information on the choice of priors and Markov chain Monte Carlo settings |
| ☒ | ☐ | For hierarchical and complex designs, identification of the appropriate level for tests and full reporting of outcomes |
| ☒ | ☐ | Estimates of effect sizes (e.g. Cohen's *d*, Pearson's *r*), indicating how they were calculated |

*Our web collection on statistics for biologists contains articles on many of the points above.*

## Software and code

Policy information about availability of computer code

| Data collection | BD FACSAriaTM III (BD Biosciences), BD FACSAriaTM Fusion (BD Biosciences), BD LSRFortessa (BD Biosciences), CytoFLEX Flow Cytometer (Beckman Coulter), MACSQuant Analyzer (Miltenyi Biotec), ImageStream®X Mk II imaging flow cytometer (AMNIS®; MERCK Millipore), Odyssey Imaging system (LI-COR Biosciences), Jess System (ProteinSimple) |
|---|---|
| Data analysis | All the software and their version information, when available, are shown. FlowJo (always latest version up to 10.6.1 upon completion of the study) was used for FACS analyses. Scripts for bioinformatic analyses were written by Albert Garcia and Gianni Panagiotou (coauthors) and Sivia Fibi-Smetana and Leila Taher. Codes have been deposited publicly (Github and Zenodo) as indicated in the manuscript (Code availability statement). GraphPad Prism (v.7-9) was used to analyzye data and to create plots. INSPIRE software, IDEAS software 6.2.64.0, Image Studio™ Lite (LI-COR Biosciences) 5.0, Compass software 6.0.0 (ProteinSimple), R version 4.1. |

For manuscripts utilizing custom algorithms or software that are central to the research but not yet described in published literature, software must be made available to editors and reviewers. We strongly encourage code deposition in a community repository (e.g. GitHub). See the Nature Portfolio guidelines for submitting code & software for further information.

## Data

Policy information about availability of data

All manuscripts must include a data availability statement. This statement should provide the following information, where applicable:

- Accession codes, unique identifiers, or web links for publicly available datasets
- A description of any restrictions on data availability
- For clinical datasets or third party data, please ensure that the statement adheres to our policy

Raw and processed Sequencing Data files are available  GEO. All accession numbers are provided in the manuscript. All data points for the remaining experiments are shown in the paper. All data points prepresent individual biological samples as indicated in the legends.

# Field-specific reporting

Please select the one below that is the best fit for your research. If you are not sure, read the appropriate sections before making your selection.

☒ Life sciences ☐ Behavioural & social sciences ☐ Ecological, evolutionary & environmental sciences

For a reference copy of the document with all sections, see nature.com/documents/nr-reporting-summary-flat.pdf

# Life sciences study design

All studies must disclose on these points even when the disclosure is negative.

| | |
|---|---|
| Sample size | The sample sizes are indicated in the respective figures with circles (in bar graphs) indicating individual donors and experiments. Sample sizes were based on our experience and common practice in the field of human immunology (i.e. Nat Immunol. 2018 Oct; 19(10): 1126–1136.) |
| Data exclusions | no data exclusions |
| Replication | Each data point indicates an independent blood donor. Multiple blood donors and experiments were performed to confirm the conclusions. The individual data points, which correlate with independent blood donors are shown in the respective graphs. |
| Randomization | Healthy donor blood from men and women (anonymous) was used. Patient samples (JIA)were provided solely based on the diagnosis. |
| Blinding | Blinding was not relevant for this study. For JIA patients blood collection, experiment and data analysis were done by three independent groups, respectively. |

# Reporting for specific materials, systems and methods

We require information from authors about some types of materials, experimental systems and methods used in many studies. Here, indicate whether each material, system or method listed is relevant to your study. If you are not sure if a list item applies to your research, read the appropriate section before selecting a response.

### Materials & experimental systems

| n/a | Involved in the study |
|---|---|
| ☐ | ☒ Antibodies |
| ☐ | ☒ Eukaryotic cell lines |
| ☒ | ☐ Palaeontology and archaeology |
| ☒ | ☐ Animals and other organisms |
| ☐ | ☒ Human research participants |
| ☒ | ☐ Clinical data |
| ☒ | ☐ Dual use research of concern |

### Methods

| n/a | Involved in the study |
|---|---|
| ☒ | ☐ ChIP-seq |
| ☐ | ☒ Flow cytometry |
| ☒ | ☐ MRI-based neuroimaging |

## Antibodies

| | |
|---|---|
| Antibodies used | Antigen; conjugate (if applicable); dilution; clone; vendor; Order number<br>FACS/Imaging flow cytometry<br>ASC; PE; 1:50; HASC-71; Biolegend; 653904<br>CCR4; PE/Cy7; 1:200; L291H4; Biolegend; 359410<br>CCR6; PE; 1:50; 11A9; BD; 559562<br>CD14; PacificBlue; 1:200-1:400; HCD14; Biolegend; 325616<br>CD3; FITC; 1:150; UCHT1; Biolegend; 300440<br>CD3; APC; 1:100; UCHT1; Biolegend; 300412 |

CD45RA; FITC; 1:200; HI100; Biolegend; 304106
CD8; PacificBlue; 1:100; SK1; Biolegend; 344718
CXCR3; APC; 1:10; 1C6/CXCR3; BD; 550967
IFN-γ; APC/Cy7; 1:300; 4S.B3; Biolegend; 502530
IL-10; PE/Cy7; 1:50; JES3-9D7; Biolegend; 501420
IL-10; APC; 1:50; JES3-9D7; BD ; 554707
IL-10; PE; 1:10; JES3-9D7; BD; 559330
IL-17A; PacificBlue; 1:100; BL168; Biolegend; 512312
IL-1a; PE; 1:50; 364-3B3-14; Biolegend; 500106
IL-1R1; PE; 1:20; FAB269P; R&D; FAB269P-100
IL-4; FITC; 1:600; MP4-25D2; Biolegend; 500807
Ki-67; Brilliant Violet 421; 1:10; Ki-67; Biolegend; 350506
NALP3/NLRP3; APC; 1:50; REA668; Miltenyi; 130-111-210
RORγt; APC; 1:10; AFKJS-9; eBioscience; 17-6988-82
IL-1b; Alexa Fluor 647; 1:50; JK1B-1; Biolegend; 508207
CCR7; PE;1:50; G043H7; Biolegend;353203
CD25; BV421;1:100;BC96; Biolegend;302640
Western blot /Jess
caspase 8; 1:50 (Jess) 1:1000 (WB); 1C12; Cell signaling; 9746T;
caspase 1; 1:1000 (WB); polyclonal; Cell signaling; 2225S
b-actin; 1:2000 (WB) 1:200 (Jess); 8H10D10; Cell signaling; 3700S
caspase-3; 1:1000 (WB) 1:50 (Jess); polyclonal; Cell signaling; 9662
gasdermin D; 1:1000 (WB) 1:50 (Jess); polyclonal; Cell signaling; 96458
cleaved gasdermin D; 1:50 (Jess) 1:1000(WB); E7H9G; Cell signaling; 36425S
NLRP3;  1:2000 (WB); D2P5E; Cell signaling; 13158S
Mouse IgG; HRP; 1:2000 (WB); polyclonal; Cell signaling; 7076
Rabbit IgG; HRP; 1:2000 (WB); polyclonal; Cell signaling; 7074
IL-1α; 1:1000 (WB); EPR5103(2); Abcam; ab134908
gasdermin E; 1:50 (Jess) 1:500 (WB); EPR19859; Abcam; ab215191;
Sodium Potassium ATPase; 1:50 (Jess); EP1845Y; Abcam; ab76020
GAPDH; 1:1000 (WB) 1:100 (Jess); 6C5; MERCK; CB1001
ASC; 1:50 (Jess); B-3; Santa Cruz Biotechnology; sc-514414
NLRP3/NALP3;  1:50 (Jess); 25N10E9; Novus Biologicals; NBP2-03948;
Anti mouse detection module; HRP;  as per manufacturer`s instructions;  Protein Simple; DM-002
Anti rabbit detection module; HRP;  as per manufacturer`s instructions;  Protein Simple; DM-001

Validation

FACS antibodies validation:

Biolegend - https://www.biolegend.com/en-us/quality/quality-control

BD - https://www.biocompare.com/Antibody-Manufacturing/355107-Antibody-Manufacturing-Perspectives-BD-Bioscience/

Milteniy Biotec - https://www.miltenyibiotec.com/DE-en/products/macs-antibodies/antibody-validation.html#gref

All FACS antibodies are commercially available and verifications can be found on respective manufacturer's website.
All antibodies have in addition been tested on the cells used herein by performing titrations according to standard recommendations (Eur J Immunol. 2019 Oct;49(10):1457-1973. doi: 10.1002/eji.201970107). They were then used at the concentrations indicated herein and in the methods section of the manuscript.

Western blot/Jess antibodies validation:

Cell Signaling Technology - https://www.cellsignal.de/about-us/our-approach-process/antibody-validation-western-blotting

Abcam - https://www.abcam.com/primary-antibodies/how-we-validate-our-antibodies#Western%20blot

NOVUS Biologicals - https://www.novusbio.com/5-pillars-validation

Santa Cruz Biotechnology - https://www.labome.com/method/Santa-Cruz-Antibodies.html

MERCK - https://www.sigmaaldrich.com/DE/en/technical-documents/technical-article/protein-biology/immunohistochemistry/antibody-enhanced-validation

All antibodies  have in addition been tested on the cells used herein by performing titrations according to standard recommendations (Eur J Immunol. 2019 Oct;49(10):1457-1973. doi: 10.1002/eji.201970107). They were then used at the concentrations indicated herein and in the methods section of the manuscript.

All western blot antibodies are commercially available and verifications can be found on respective manufacturer's website. In addition, western blot antibodies were validated using genetic strategy: expression of the target protein was compared before and after knockout using CRISPR/Cas9 technology. If protein expression following knockout was substantially reduced, then antibody was considered as specific.

# Eukaryotic cell lines

Policy information about cell lines

| Cell line source(s) | Allogeneic PBMCs were used as feeder cells and were isolated from healthy donors. T cell lines and T cell clones were generated from primary human cells and kept short term in culture. |
| --- | --- |
| Authentication | does not apply |
| Mycoplasma contamination | not tested in primary human t cells. |
| Commonly misidentified lines (See ICLAC register) | does not apply |

# Human research participants

Policy information about studies involving human research participants

| Population characteristics | healthy, men and women, age: 22-65 |
| --- | --- |
| Recruitment | fresh blood from healthy anonymous blood donors and buffy coats from the blood banks of the Charite Universitätsmedizin Berlin and the Universitätsklinikum Jena were used whenever needed. Clinical blood and synovial fluid samples were obtained from Bas Vastert (University Medical Center Utrecht, Biobank). The recruitment occured based on diagnosis and the samples were stored in a biobank and selected randomly based on the diagnosis criterium only. |
| Ethics oversight | The ethics committees of the Charité Universitätsmedizin Berlin, the Technical University of Munich and the Friedrich Schiller University of Jena approved the study with with positive ethics votes. |

Note that full information on the approval of the study protocol must also be provided in the manuscript.

# Flow Cytometry

## Plots

Confirm that:

☒ The axis labels state the marker and fluorochrome used (e.g. CD4-FITC).

☒ The axis scales are clearly visible. Include numbers along axes only for bottom left plot of group (a 'group' is an analysis of identical markers).

☒ All plots are contour plots with outliers or pseudocolor plots.

☒ A numerical value for number of cells or percentage (with statistics) is provided.

## Methodology

| Sample preparation | Primary cells were isolated as described in the methods (Ficoll isolation, positive magnetic isolation using microbeads, flow-cytometry assisted cell sorting) |
| --- | --- |
| Instrument | BD FACSAria, BD LSRFortessa, Cytoflex, Cytex AuroraBD, FACSAriaTM III (BD Biosciences), BD FACSAriaTM Fusion (BD Biosciences), BD LSRFortessa (BD Biosciences), CytoFLEX Flow Cytometer (Beckman Coulter), MACSQuant Analyzer (Miltenyi Biotec) |
| Software | FlowJo Software (Tree Star Inc) for FACS analyses |
| Cell population abundance | Purity of the relevant cell populations was checked after sorting and found to be >98% |
| Gating strategy | The gating strategies are shown in the Extended Data Fig. File 3. Lymphocytes were gated by FSC/SSC and exclusion of dead cells by zombie dye, exlcusion of doublets as shown, and further gating for CD4+CD14– CD3+ T cells and subgating for memory marker CD45RA– (CD45RA– for memory T cells, CD45RA+ for naive T cells) and the differential expression of chemokine receptors for the respective T helper cell subsets as shown and explained in the methods and the results section. The positive populations were defined with the use of unstained and single-stained controls and normally were above 10³ on a log scale. |

☒ Tick this box to confirm that a figure exemplifying the gating strategy is provided in the Supplementary Information.

