## [Peer Review File · Nature Immunology]

Peer Review Information

Journal: Nature Immunology

Manuscript Title: Human Th17 cells engage gasdermin E pores to release IL-1a upon NLRP3 inflammasome activation

Corresponding author name(s): Christina Zielinski

Reviewer Comments & Decisions:

Decision Letter, initial version:
--

Subject: Decision on Nature Immunology submission NI-A33904

Message: 26th May 2022

Dear Professor Zielinski,

Your Article, "Human Th17 cells engage gasdermin E pores to release IL-1a upon NLRP3 inflammasome activation" has now been seen by 3 referees. You will see from their comments copied below that while they find your work of considerable potential interest, they have raised concerns that must be addressed. In light of these comments, we cannot accept the manuscript for publication, but would be very interested in considering a revised version that addresses these serious concerns.

We hope you will find the referees' comments useful as you decide how to proceed. If you wish to submit a substantially revised manuscript, please bear in mind that we will be reluctant to approach the referees again in the absence of major revisions.

PLEASE NOTE that we have also looked over your author response and are mostly satisfied by your plans for revision. The one issue that we somewhat disagree with you is with regards to the Schnitzler syndrome data. We agree with the reviewers that these data, although interesting and supportive are not clearly evidence of your proposed gasdermin E dependent mechanism, so we think you either need further data here or you need to de-emphasize these data. It is also not really clear to us why you have chosen to focus on this very rare disease and not provided data from other Nlrp3 or Th17 related or autoinflammatory diseases for which it might be easier to provide the reviewer requested details that add physiological relevance for your mechanism. i.e. it is not apparent to us that your mechanism is in any way Schnitzler syndrome specific. Please feel free to email me if you wish to discuss this further as this issue of physiological in vivo evidence for your mechanism is clearly one the reviewers also think is important.

If you choose to revise your manuscript taking into account all reviewer and editor comments, please highlight all changes in the manuscript text file in Microsoft Word

format.

* If you have not done so already please begin to revise your manuscript so that it conforms to our Article format instructions at <http://www.nature.com/ni/authors/index.html>. Refer also to any guidelines provided in this letter.

The Reporting Summary can be found here:
<https://www.nature.com/documents/nr-reporting-summary.pdf>

You may use the link below to submit your revised manuscript and related files:
[REDACTED]

If you wish to submit a suitably revised manuscript we would hope to receive it within 6 months. If you cannot send it within this time, please let us know. We will be happy to

consider your revision so long as nothing similar has been accepted for publication at Nature Immunology or published elsewhere.

Nature Immunology is committed to improving transparency in authorship. As part of our efforts in this direction, we are now requesting that all authors identified as 'corresponding author' on published papers create and link their Open Researcher and Contributor Identifier (ORCID) with their account on the Manuscript Tracking System (MTS), prior to acceptance. ORCID helps the scientific community achieve unambiguous attribution of all scholarly contributions. You can create and link your ORCID from the home page of the MTS by clicking on 'Modify my Springer Nature account'. For more information please visit www.springernature.com/orcid.

Thank you for the opportunity to review your work.

Sincerely,

Nick Bernard, PhD
Senior Editor
Nature Immunology

Reviewers' Comments:

Reviewer #1:

Remarks to the Author:

This paper reports that NLRP3 activates GSDME to cause IL-1alpha release in Th17 cells. Although CD4+ T-cell expression of IL-1alpha has been known for many years (e.g. PMID: 1673143; PMID: 16788102) and NLRP3 signalling has also been reported in T-cells, the finding that GSDME, not GSDMD, mediates IL-1alpha release is novel and also very much unexpected. The findings may also inform on how targeting IL-1alpha (or caspase-3/GSDME), as opposed to IL-1beta (or caspase-1/GSDMD), may be used to suppress pathogenic Th17 cell functions in autoimmune/autoinflammatory conditions. However, I still have several major concerns regarding i) whether GSDME-IL-1alpha release occurs in the absence of cell death (point 11, below), and ii) whether the proposed pathway of NLRP3-caspase-8-caspase-3-GSDME-IL-1alpha has been correctly defined (point 14, below). Further, a more detailed investigation into why Schnitzler syndrome patient Th17 cell IL-1alpha release is increased would greatly benefit from a more detailed analysis (point 15, below). Specific points:

1. Figure 2b. The percentage of Th17 cells staining positive for IL-1alpha is ~ 10% of the population. Can the authors provide an explanation as to why only a fraction of cells exhibit expression of IL-1alpha (e.g. do more detailed time courses reveal waves of expression)?
2. Figure 2f/g. Important IL-1beta and TGFbeta treatment alone controls are lacking.

3. Figure 2h. The difference in levels between CCR6⁻ and CCR6⁺ cells is not convincing. i.e. four donor CCR6⁺ cell IL-1alpha levels are equivalent to the IL-1alpha levels in CCR6⁻ cells, and only 4 are elevated. More data points are required.
4. Figure 3a. Cell lysate expression analysis of IL-1alpha (pro and cleaved levels) is required for comparison.
5. Figure 3c-e. Western blots examining the levels/release of pro and mature IL-1alpha would provide valuable information e.g. To show that a corresponding increase in pro IL-1alpha is observed upon calpain inhibition, not just an abrogation of IL-1alpha expression/levels. Calpain activity is reportedly required for maturation of IL-1alpha in macrophages following inflammasome activation (<https://doi.org/10.1016/j.celrep.2021.108887>), so one might also expect that IL-1alpha maturation to be reduced in monocytes treated with a calpain inhibitor.
6. It is written "Despite the essential role of calpain for pro-IL-1a maturation, the mechanism leading to the extracellular release of cleaved IL-1a still remains unknown." The authors should note studies, not referenced (such as <https://doi.org/10.1016/j.celrep.2021.108887>), documenting the mechanism of IL-1alpha cleavage and release, albeit in other cell types.
7. Figure 4f. Please clarify the Th17 cell stimulations and whether these are identical to Fig. 3e – e.g. presumably CD3/CD28 stimulation, but why is IL-1beta also used in Fig. 3f?
8. Figure 4f. The caspase-1 levels in the supernatant do increase in Th17 cells and I expect if more experiments were performed this would become significant.
9. Figure 4g. Controls to show functionality of YVAD-cmk are essential for the interpretation of this data.
10. Figure 5d. Results should be verified by western blot for release of mature IL-1alpha (as per figure 3a). This would also inform as to whether Th17 GSDMD/E pores contribute to calpain activity and hence IL-1alpha maturation, or not.
11. Figure 5e. LDH is not a sensitive method for detecting cell death, particularly if only a very small subset of Th17 cells contain GSDME pores. In addition, the half-life of LDH in media is only ~ 9 hours, making it an unsuitable readout for cell death at longer time points. A single cell analysis of T-cell numbers/death in control and GSDME KO is required in order to verify whether GSDME is causing cell death, or not. The analysis of IL-1alpha/Ki-67 positive cells etc, does not eliminate that potential for the small number of cells "releasing" IL-1alpha to be killed by GSDME pore formation.
12. Figure 5i. Lacks molecular weight markers which makes the data difficult to interpret.
13. Figure 5j-l etc. Cleavage of caspase-8 appears to occur upstream of NLRP3-caspase-3-GSDME. Does caspase-8 activate NLRP3 (e.g. is ASC specking reduced upon caspase-8 loss or is caspase-8 recruited into the Th17 cell ASC speck, as observed in macrophages)?
14. Does the more specific caspase-1 inhibitor VX-765 impact either caspase-3 or GSDME activation and IL-1alpha release (i.e. could the signalling pathway actually be TCR-caspase-8-NLRP3-caspase-1-caspase-3-GSDME)? I understand that YVAD treatment did not impact IL-1alpha release, but this may result from cell death caused by YVAD targeting of caspase-8 and necroptotic death signalling; as all the inhibitors used here and elsewhere, IETD, DEVD, YVAD are relatively non-specific and can efficiently inhibit many caspases e.g. see PMID: 18976637. Arguably the best way to address if caspase-1 contributes to caspase-3/GSDME activation is to CRISPR target caspase-1 and examining GSDME/caspase-3 processing and IL-1alpha release.
15. Figure 6f. Lacks mechanistic insight. E.g Do the patients display increased/altered IL-1alpha/NLRP3/caspase-1/caspase-8/caspase-3/GSDME/GSDMD levels/activity (i.e. what is the mechanism in these patients that results in increased IL-1alpha release)? Does IL-1beta treatment in vitro recapitulate what is observed in these patients receiving anti-IL-1beta therapy?

16. Can the authors comment on how NLRP3 activated in Th17 cells upon TCR signalling?

Reviewer #2:

Remarks to the Author:

Chao and colleagues identify a role for NLRP3/Casp8/Casp3/GSDME regulating IL-1a production by Th17 cells. They show that this is calpain processed IL-1a and find it secreted by Th17 cells from patients with Schnitzlers syndrome. Overall the molecular pathway may be OK, although the link to Schnitzlers syndrome is tenuous, and some details seem lacking.

Major points:

It is concerning that the details for FACS analysis of intracellular IL-1a are not provided. It states to refer reference 44, and in that paper cells were "restimulated for 5 hours with PMA and ionomycin in the presence of brefeldin A for the final 2.5 hours of culture". Is that indeed the case here, as the authors own data shows IL-1a release is not impeded by BFA?

Why does calpain inhibition or CAPN2 deletion prevent IL-1a secretion, when presumably the NLRP3/Casp8/Casp3/GSDME pathway is still functional for IL-1a release? Shouldn't it be released, just as full length unprocessed IL-1a? Perhaps the processing of IL-1a should be quantified in these experiments?

Minor points:

The legend of Figure 2 is lacking, what is inducing IL-1a in Th1/2 cells here, similar to the Th17 cells?

There is huge variation between how IL-1a ELISA data is reported: pg/ml or per 10^6 cells or fold change. Negative controls are not provided in a lot of experiments.

Some claims are demonstrably false such as "GSDME expression has never been reported in T cells before.". Please see Tixeira et al Front Imm 2018 and other papers.

4e spliced westerns.

Although IL-1a appears to have some effect on IL-10 levels, the reciprocal effect was not determined, so should not feature in speculation about the existence of feedback loops.

Some Schnitzler syndrome patients have somatic mutations in NLRP3, was that the case for the T-cell clones studied here?

I am not sure that IL-1a could be considered "the etiological disease-defining cytokine" in Schnitzler syndrome.

Reviewer #3:

Remarks to the Author:

Review Zielinski 2022 NI

In the manuscript submitted by Chao et al. "Human Th17 cells engage gasdermin E pores to release IL-1a upon NLRP3 inflammasome activation" the authors describe the expression of IL-1a by human Th17 cells. In the Th17 cells, the authors found IL-1a expression to coincide with cell viability and proliferation. Furthermore, the authors found this process to be regulated by the NLRP3 inflammasome and gasdermin E pores and the autocrine IL-1a signaling in Th17 cells to suppress IL-10 and induce a pathogenic phenotype. IL-1a is a critical regulator of Th17 differentiation and the paper shows that IL-1 produced by differentiating Th17 cells plays a critical role in their growth and differentiation. The manuscript is well written and describes an interesting novel role of the NLRP3 inflammasome and IL-1a secretion in Th17 cell biology. However, the manuscript would improve from better use of the single-cell data. In order to put the observation in context and the importance of IL-1 in T cell differentiation, it would be also important to put more emphasis on the physiological role of IL-1a secretion by Th17 cells.

Major comments:

1. It would be important to compare the level of IL-1a expression by Th17 cells to multiple other cell types that are bona fide producers of IL-1a (macrophages, DCs, neutrophils, B cells...), in Figure 2a. It is important to define how the level of IL-1a expression compares to cells known to express IL-1a in which Gasdermin D is activated for IL-1 release. Is it at a similar level and can therefore be expected to have a paracrine biological function or is it a low level in the majority of T cells, which may act as an autocrine growth without induction of much cell death? Or is cell death always associated with IL-1 release from T cells? There are emerging data showing that IL-1a can also be secreted by DCs without pyroptosis, resulting in the generation of hyperactive DCs.
2. In Figure 6 to show that the IL-1a expression by Th17 cells themselves is important for their function, CRISPR-CAS9 knockout of IL-1a should be performed in the Th17 cells to study the role of IL-1 in cell-intrinsic production leading to Th17 differentiation.
3. What is the physiological function of IL-1a secretion by Th17 cells and does IL-1 production by T cells have a role during infections or autoimmune diseases? Any evidence for that? The Schnitzel syndrome seems generally driven by IL-1a-expressing cells and that Th17 cells from these patients also express IL-1a is not entirely surprising and the fact that IL-1a blockade decreases IL-1a by Th17 cells is also expected. In the Schnitzel syndrome, how do the levels expressed by Th17 cells compare to other immune cell populations that express IL-1a? Can the IL-1a produced by Th17 cells be expected to contribute to the disease? Is there any evidence for IL-1a expression by Th17 cells in any other conditions, including infections or in response to commensal bacteria?
4. The single-cell data should be analyzed more deeply to develop a nuanced understanding of IL-1 production by T cells. What are the genes co-expressed with IL-1a (Figure 1a)? What are the differentially expressed genes in cluster 1? What does the expression of gasdermin E, NLRP3, and caspases look like on the UMAP, even though the process is supposed to be driven at the protein level but does the expression of inflammasome-specific genes change and co-expressed and co-regulated in T cells as has been observed in DCs and myeloid cells? A comparison of the gene expression of IL-1 production in DCs vs. T cells will give additional information about the common drivers and differences between the two cell types, which will give further mechanistic information

about the mechanisms by which IL-1 is produced by T cells. Are they specifically up-regulated in cluster 1?

Minor comments:

1. The introduction is too short. Inflammasomes, NLRP3, and IL-1a in general and in the context of T cells are not being introduced. The manuscript would improve if the authors would provide more background and explain the importance of the studied molecules in immune-mediated diseases, for a general reader.

2. Does cluster 1 (Figure 1C) express a higher proliferation/survival signature that would support the proposed hypothesis that IL-1 drives the proliferation and expansion of Th17 cells? In addition, what are generally the DE genes of the different clusters? The heterogeneity of human Th17 cells by scRNAseq would be of general interest to the reader.

In summary, the manuscript describes an interesting and novel finding that human Th17 cells express IL-1a and also partly addresses the mechanism of IL-1a secretion by T cells. The physiological role of IL-1a in human Th17 cells (in health or disease), however, remains unclear and how often is this mechanism used by T cells for differentiation and function. More analysis and results in that direction would significantly improve the impact and provide a mechanistic understanding of the role of IL-1 produced by T cells in Th17 differentiation and induction of tissue inflammation.

Author Rebuttal to Initial comments

Reviewer 1.

This paper reports that NLRP3 activates GSDME to cause IL-1alpha release in Th17 cells. Although CD4+ T-cell expression of IL-1alpha has been known for many years (e.g. PMID: 1673143; PMID: 16788102) and NLRP3 signalling has also been reported in T-cells, the finding that GSDME, not GSDMD, mediates IL-1alpha release is novel and also very much unexpected. The findings may also inform on how targeting IL-1alpha (or caspase-3/GSDME), as opposed to IL-1beta (or caspase-1/GSDMD), may be used to suppress pathogenic Th17 cell functions in autoimmune/autoinflammatory conditions. However, I still have several major concerns regarding i) whether GSDME-IL-1alpha release occurs in the absence of cell death (point 11, below), and ii) whether the proposed pathway of NLRP3-caspase-8-caspase-3-GSDME-IL-1alpha has been correctly defined (point 14, below). Further, a more detailed investigation into why Schnitzler syndrome patient Th17 cell IL-1alpha release is increased would greatly benefit from a more detailed analysis (point 15, below).

Specific points:

1. Figure 2b. The percentage of Th17 cells staining positive for IL-1alpha is ~ 10% of the population. Can the authors provide an explanation as to why only a fraction of cells exhibit expression of IL-1alpha (e.g. do more detailed time courses reveal waves of expression)?

AR (author response): We have reported IL-1 α secretion to represent a new/overlooked property of human Th17 cells and have demonstrated that this property is restricted to a small subset within the Th17-cell population but not any other T-cell subset (evidence given by 1. scRNA-seq data (Fig. 1a,b) and 2. Th17-cell clones (Fig. 1c)), which

distinguishes itself from the remaining Th17 cells by its enhanced proinflammatory identity (Fig. 2a-d). To the best of our knowledge, this has not been reported before, and has not been reported in the referenced manuscripts.

The requested explanation to the question of why only ~ 10% of the Th17 cell population exhibit expression of IL-1 α is due to our finding, that only a dedicated proinflammatory subset within Th17 cells expresses IL1 α (Fig. 2 NEW). Furthermore, we present new data in the revised manuscript demonstrating that IL-1 α production by Th17 cells is restricted to Th17 cells with TCR specificity for *C. albicans* (Fig. 8 NEW).

As the reviewer correctly implies, IL-1 α expression levels may be related to the timepoint of analysis after CD3 and CD28 stimulation and thus the activation state of the Th17 cells. This is indeed the case as shown 1.) by TCR signalling and calcium flux-dependent IL-1 α production (Fig. 3b, Fig. 4b, g, h) and 2.) by the requested time course experiment ("waves of expression"), which is shown in Fig. 7f.

2. Figure 2f/g. Important IL-1beta and TGFbeta treatment alone controls are lacking.

AR (author response): We now provide the requested IL-1beta- and TGFbeta-only controls in the revised version of the manuscript (Extended Data Fig. 8 NEW, Fig. 3h NEW). The reason for not having shown it in the first place was to restrict the conditions to the polarizing conditions only, which in the case of Th17-cell priming is the combination of IL-1beta plus TGFbeta. However, we realize that the single-treatment controls are very important, as pointed out by the reviewer, and that they highlight the synergetic effect of the Th17-cell polarizing cytokine combination on IL-1 α induction.

3. Figure 2h. The difference in levels between CCR6- and CCR6+ cells is not convincing. i.e. four donor CCR6+ cell IL-1alpha levels are equivalent to the IL-1alpha levels in CCR6- cells, and only 4 are elevated. More data points are required.

AR (author response): As suggested by the reviewer, more data points have now been added. We have added 3 more donors adding up to 11 tested donors in total. With the new data points, we corroborated the significant increase in IL-1alpha expression in CCR6+ compared to CCR6- T cells and even improved the statistical significance (Fig. 3i NEW). Additionally, we now also tested IL-1alpha release into the supernatant by ELISA and consistently observed a clear increase in IL-1alpha production by CCR6+ versus CCR6- T cells. We provide these additional data for the reviewer's perusal (Fig. R1 NEW).

4. Figure 3a. Cell lysate expression analysis of IL-1alpha (pro and cleaved levels) is required for comparison.

AR (author response): As suggested by the reviewer, we have now performed the suggested the cell lysate expression analysis of IL-1 α (pro- and cleaved levels) in Th17 cells for comparison to the supernatant (Extended data

Fig. 10a NEW). As expected, we observed both pro- and cleaved forms in the cell lysate and a strong bias for cleaved IL-1 α in the supernatant (due to calpain mediated cleavage) (Fig. 4a, Extended data Fig. 10a NEW). Some pro-IL1 α can be found in the supernatant due to necrosis-associated pro-IL1 α release (as demonstrated by the presence of β -actin in the cumulative 5-day supernatant, Extended data Fig. 10a NEW).

5. Figure 3c-e. Western blots examining the levels/release of pro and mature IL-1alpha would provide valuable information e.g. To show that a corresponding increase in pro IL-1alpha is observed upon calpain inhibition, not just an abrogation of IL-1alpha expression/levels. Calpain activity is reportedly required for maturation of IL-1alpha in macrophages following inflammasome activation (<https://doi.org/10.1016/j.celrep.2021.108887>), so one might also expect that IL-1alpha maturation to be reduced in monocytes treated with a calpain inhibitor.

AR (author response): We absolutely agree that it would further support the data if an intracellular increase in pro-IL1alpha is also shown upon calpain inhibition. We now provide the requested data highlighting the intracellular increase in pro-IL-1alpha upon treatment with the calpain inhibitor (Fig. 4d NEW).

We are also intrigued about the difference in the regulation of IL-1 α exit in T cells (in which it is trapped after calpain inhibition) and monocytes (in which it is not trapped) (Fig. 4d NEW). As we and others have shown before (Gross et al *Immunity* 2012, Tsuchiya et al. *Cell Rep* 2021, ref. now added following reviewer's suggestion), uncleaved IL-1alpha can also exit macrophages or BMDCs following inflammasome activation, which is consistent with our monocyte control comparison (Extended Data Fig. 10b).

6. It is written "Despite the essential role of calpain for pro-IL-1a maturation, the mechanism leading to the extracellular release of cleaved IL-1a still remains unknown." The authors should note studies, not referenced (such as <https://doi.org/10.1016/j.celrep.2021.108887>), documenting the mechanism of IL-1alpha cleavage and release, albeit in other cell types.

AR (author response): We realize that our statement must have caused some misunderstanding. It was not meant to summarize the literature. Instead, with this introductory sentence for Fig. 5, we specifically summarize our own data from the preceding section, which demonstrated an essential role for calpain in pro-IL-1 α maturation in the preceding Fig. 4. We then conclude that a mechanism for the extracellular release of this cleaved IL-1 α still needs to be identified, which is then addressed in the following section and Figures 5-6 (GSDME). We have therefore moved the mentioned sentence to the preceding section to make this clear in the revised version of the manuscript.

The suggested reference (GSDMD regulation in macrophages and role for IL-1 α release) is indeed a very good addition to our discussion chapter to stress that IL-1alpha release in T cells is executed via a different and new mechanism than the one previously shown in macrophages (see also our answer to 5.). This reference has now been included.

7. Figure 4f. Please clarify the Th17 cell stimulations and whether these are identical to Fig. 3e – e.g. presumably CD3/CD28 stimulation, but why is IL-1beta also used in Fig. 3f?

AR (author response): We have clarified the Th17 stimulatory conditions in the revised manuscript. In short, Th17 cells were always stimulated with anti-CD3 and anti-CD28 plate-bound mAbs for 48 h and then transferred to an uncoated plate before analysis on day 5 (cumulative 5-day supernatant).

In Fig. 3F (now Fig.4g) no IL-1beta was used. To the best of our knowledge, we have not indicated its use in the manuscript. In case the reviewer meant Fig. 4f (instead of Fig. 3f, which is now Fig 5f), the answer is that we added this extra condition (in addition to the no-IL-1beta condition) because we have shown before that IL-1beta and Th17-polarizing conditions (TGFbeta+IL1beta) promote IL-1alpha secretion (Fig. 3f,g,h NEW, Extended Data 8 NEW, Extended Data Fig.13 a,c NEW). Therefore, an increase in caspase-1 cleavage and secretion into the supernatant could be expected upon IL-1beta-enhanced IL-1alpha secretion if caspase-1 is involved in the regulation of IL-1alpha in T cells. We have therefore further excluded a role for caspase-1 in this context. We have clarified the use of this extra condition in the revised version of the manuscript.

We are happy to exclude this additional control condition if it causes confusion because we have clearly ruled out a role for active caspase-1 via several other experimental strategies (i.e. CASP1 Crispr-KO, Fig. 5g NEW). We have moved the previous Fig. 4g to the Extended Data. Fig. 14a.

8. Figure 4f. The caspase-1 levels in the supernatant do increase in Th17 cells and I expect if more experiments were performed this would become significant.

AR (author response): We have corroborated our finding that IL-1alpha secretion is independent of caspase-1 with multiple different and independent readouts and experimental strategies, including new data:

1. Cleaved caspase-1 was not detected by Western blot analysis (Fig. 5e).
2. IL-1alpha secretion was not reduced upon inhibition of caspase-1 (Extended Data Fig.14).
3. Caspase-1 expression was not regulated by Th17-cell priming cytokines, despite upregulation of IL-1alpha by these cytokines (Extended data Fig. 13 a, b NEW, Fig. 3h NEW, Extended Data Fig. 16 right panel NEW),
4. Negative FLICA staining indicated the absence of active caspase-1 in T cells at the single-cell level by FACS, in contrast to the case in monocytes (Extended data Fig. 13 a-d NEW)
5. In caspase-1 knockout Th17 cells (CRISPR-Cas9 KO), IL-1alpha secretion was not affected (Fig. 5g NEW).

Given all the findings, we are highly confident that caspase-1 cleavage does not have any role in the secretion of IL-1alpha in human T cells.

9. Figure 4g. Controls to show functionality of YVAD-cmk are essential for the interpretation of this data.

AR (author response): We agree and have added the requested functionality controls of YVAD-cmk in the revised version of the manuscript (Extended Data 14b NEW). YVAD-cmk strongly inhibited IL-1beta secretion (4-fold reduction) in monocytes stimulated with LPS and nigericin. We are therefore very confident that the inhibitor we used was functional, in line with previous reports (i.e. Schneider et al. *Cell reports* 2017).

10. Figure 5d. Results should be verified by western blot for release of mature IL-1alpha (as per figure 3a). This would also inform as to whether Th17 GSDMD/E pores contribute to calpain activity and hence IL-1alpha maturation, or not.

AR (author response): IL-1alpha was quantified by ELISA in this experiment after we showed in Fig. 4a (previously Fig. 3a) that the secreted IL-1alpha was present in the supernatant in its cleaved form. Even if the supernatant contained uncleaved IL-1alpha, this would not undermine our conclusion that GSDME is responsible for the IL-1alpha exit into the supernatant, given the reduction in overall IL-1alpha as assessed by ELISA. Therefore, the ELISA method is, in our opinion, a very appropriate and sensitive technique for this experimental question. An analysis by Western blot is technically challenging considering the availability of few primary Th17 cells per time point in a time-course experiment. In addition, the protein abundance in the supernatant is generally very low (compared to that in lysates), making frequent quantifications technically very difficult.

Nevertheless, the reviewer raises the very interesting new aspect of a potential cross-talk of GSDME and calpain activity. It could be possible that GSDME pores may provide an additional means by which calcium could enter and promote calpain activity since it has previously been reported that GSDMD facilitates calcium flux in mouse macrophages (Tsuchiya K et al., Cell reports 2021). We therefore investigated whether GSDME pores contribute to calpain activity and hence IL-1alpha maturation by testing calpain activity in Th17 cells with and without a CRISPR-Cas9 KO for GSDME. Our data do not show a significant change in calpain activity upon genetic depletion of GSDME. This supports the mechanism proposed in the manuscript, which is summarized in Extended Data Fig. 22. The data are provided for your perusal (Fig. R2 NEW).

11. Figure 5e. LDH is not a sensitive method for detecting cell death, particularly if only a very small subset of Th17 cells contain GSDME pores. In addition, the half-life of LDH in media is only ~ 9 hours, making it an unsuitable readout for cell death at longer time points. A single cell analysis of T-cell numbers/death in control and GSDME KOs is required in order to verify whether GSDME is causing cell death, or not. The analysis of IL-1alpha/Ki-67 positive cells etc, does not eliminate that potential for the small number of cells “releasing” IL-1alpha to be killed by GSDME pore formation.

AR (author response): We absolutely agree with the reviewer and have therefore performed the CytoTox assay with supernatants from short-term cultures after fresh medium replacement on day 4 and analysis on day 5 as indicated in the methods section of our manuscript, in accordance with previously published pyroptosis analyses with innate cells and according to the reviewer’s suggestions. We realized that our legend was mixed up with the time for CRISPR-Cas9 engineering and was therefore confusing. We have now corrected it and improved it for better clarity. Additionally, we have added a positive control condition with monocytes, which is used to demonstrate pyroptosis (Fig. 7D NEW). To further corroborate our unexpected finding that IL-1 α secretion and GSDME pore formation occur in the absence of cell death, unlike in innate cells, we have provided further evidence via multiple different experimental strategies:

1. We selected T-cell clones with varying levels of IL-1 α expression and recloned them individually to monitor their cloning efficiency in correlation to their initial IL-1 α expression on the single-cell level. We were able to exclude the possibility that IL-1 α expression correlates with cell death on the single-cell level (Fig. 7e NEW).
2. In an attempt to isolate IL-1 α ⁺ and IL-1 α ⁻ Th17 cells for single-cell cloning, we discovered that T cells, unlike innate cells, do not express IL-1 α on the cell surface (Extended Data Fig. 9d). We therefore established and validated a new cytokine secretion assay for IL-1 α , which captures IL-1 α on the outer cell membrane upon autocrine IL-1 α secretion. This allowed us to sort IL-1 α ⁺ and IL-1 α ⁻ viable Th17 cells, to clone single cells and

to monitor the cloning efficiency at the single-cell level. Again, no difference in the single-cell cloning efficiency of IL-1 α ⁺ and IL-1 α ⁻ Th17 cells was observed (Extended Data 20 NEW), excluding IL-1 α -associated cell death/pyroptosis.

3. A time-course analysis with Th17-cell clones that were repetitively restimulated, demonstrated no selective loss of IL-1 α -producing cells from the respective clonal population. IL-1 α expression was activation dependent and re-inducible upon restimulations (Fig. 7f).
4. We generated T-cell clones, selected IL-1 α ⁺ versus IL-1 α ⁻ T-cell clones and performed a transcriptomic comparison (mRNAseq), which revealed gene signatures of increased proliferation (Fig. 7g NEW) in IL-1 α -producing Th17-cell clones.
5. We also performed scRNA-seq and compared Th17 cells that expressed *IL1A* to Th17 cells that did not. This also revealed high proliferation and viability upon transcriptomic single-cell analysis (Fig. 7h NEW).
6. We also performed a new bulk mRNA-seq analysis comparing Th17 cells with and without CRISPR-Cas9-engineered GSDME knockout. This showed no change in a wide range of death pathways or death-associated genes upon comparison of both populations by GSEA (Fig. 7b NEW).
7. scRNA-seq analysis comparing *GSDME*-positive and *GSDME*-negative Th17 cells demonstrated even increased enrichment in the module for positive regulation of T-cell proliferation for *GSDME*-positive single-cells (Fig. 7c NEW).

Therefore, we have used different strategies to address the reviewer's suggestion to use a single-cell analysis approach to answer the question of whether GSDME is associated with cell death. Based on our consistent findings, we are confidently able to rule out that GSDME causes cell death in human Th17 cells, which is in contrast to the case in myeloid cells.

Overall, we are very confident that our existing data and the new experiments strongly support the conclusion that human Th17 cells do not undergo pyroptosis upon GSDME-mediated IL-1 α release but are instead maintained long-term in accordance with their adaptive memory function.

12. Figure 5i. Lacks molecular weight markers which makes the data difficult to interpret.

AR (author response): The molecular weight markers are now included in the figure (now Fig. 6e).

13. Figure 5j-l etc. Cleavage of caspase-8 appears to occur upstream of NLRP3-caspase-3-GSDME. Does caspase-8 activate NLRP3 (e.g is ASC specking reduced upon caspase-8 loss or is caspase-8 recruited into the Th17 cell ASC speck, as observed in macrophages)?

AR (author response): In order to address the reviewer's question of whether caspase-8 is upstream of the NLRP3-casp3-GSDME axis we knocked out caspase-8 by CRISPR-Cas9 in human Th17 cells and assessed inflammasome

speck formation (active inflammasome formation) as suggested by the reviewer. No difference in NLRP3-speck formation in the presence or absence of caspase-8 depletion was observed, which suggests that caspase-8 rather acts downstream of NLRP3 as an effector protease, as we have proposed in the manuscript (Fig. R7). Although caspase-8 has previously been reported to be a positive upstream regulator of NLRP3 inflammasome activation in caspase-1 intact cell types, our data are in line with previous reports in the setting of caspase-1 deficiency (Th17 cells also lack caspase-1 cleavage) (Antonopoulos C et al. 2015, Gurung P & Kanneganti TD 2015, +others).

14. Does the more specific caspase-1 inhibitor VX-765 impact either caspase-3 or GSDME activation and IL-1alpha release (i.e. could the signalling pathway actually be TCR-caspase-8-NLRP3-caspase-1-caspase-3-GSDME)? I understand that YVAD treatment did not impact IL-1alpha release, but this may result from cell death caused by YVAD targeting of caspase-8 and necroptotic death signalling; as all the inhibitors used here and elsewhere, IETD, DEVD, YVAD are relatively non-specific and can efficiently inhibit many caspases e.g. see PMID: 18976637. Arguably the best way to address if caspase-1 contributes to caspase-3/GSDME activation is to CRISPR target caspase-1 and examining GSDME/caspase-3 processing and IL-1alpha release.

AR (author response): As suggested by the reviewer, we have now used the more specific caspase-1 inhibitor VX-765 to perform the suggested experiment. In accordance with the data obtained using the previous inhibitor, we did not observe an impact of caspase-1 inhibition on caspase-3 or GSDME activation (Extended Data Fig.18 NEW). As also suggested by the reviewer, we have CRISPR-targeted caspase-1 and can definitely exclude a role for caspase-1 in IL-1 α secretion in human Th17 cells (Fig. 5g NEW, see also response to 8.). This is in accordance with our data demonstrating the absence of cleaved caspase-1 expression in human Th17 cells (Western blot, ELISA, FLICA staining). We have also monitored cell death via viability assay by FACS analysis and performed cell counting and can exclude a role for cell death in the interpretation of our findings. We have now also dedicated a separate figure to the topic of cell death/proliferation (excluding pyroptosis) (Fig. 7 NEW),

15. Figure 6f. Lacks mechanistic insight. E.g Do the patients display increased/altered IL-1alpha/NLRP3/caspase-1/caspase-8/caspase-3/GSDME/GSDMD levels/activity (i.e. what is the mechanism in these patients that results in increased IL-1alpha release)? Does IL-1beta treatment in vitro recapitulate what is observed in these patients receiving anti-IL-1beta therapy?

AR (author response): With our data from Schnitzler Syndrome patients, we intended to provide the first report on T-cell derived IL-1alpha production in a human disease. We realize it is tempting to investigate every single component of the IL-1alpha/NLRP3/(caspase-1)/caspase-8/caspase-3/GSDME/(GSDMD)-pathway in a specific clinical situation, as suggested by the reviewer. However, due to limitations of patient material and the potential existence of multiple abrogated targets, this was not possible to us. We have therefore removed our previous Schnitzler Syndrome data to de-emphasize any claims about its pathogenesis, as also proposed by the editor. Instead, we have now added new data on the more frequent autoimmune disease juvenile idiopathic rheumatoid arthritis (JIA) to the manuscript, before the mechanism is presented (Fig. 2g NEW). Interestingly, we observe strongly increased IL-1 α expression of Th17 cells in the blood of patients as compared to healthy donors and also a very high frequency of IL-1 α ⁺ Th17 cells in the synovial fluid in these patients. This will serve as the first report on IL-1 α production by T cells in a disease. These clinical data also support other new data, which we included to demonstrate the pro-inflammatory identity of the IL-1 α producing subset of Th17 cells (Fig. 2a-f NEW).

To provide more information of the physiological relevance of IL-1 α producing Th17 cells, we have added new data in the context of anti-microbial host defense. We demonstrate that the ability of T cells to produce IL-1 α is associated with their TCR specificity (Fig. 8 NEW). Th17 cells are known to enrich for antigen specificities against *C. albicans* and *S. aureus*. We demonstrated that Th17 cells with an antigen specificity for *C. albicans* produce high levels of IL-1 α , whereas Th17 cells specific for *S. aureus* do not. Furthermore, we demonstrate that Th17 cells clear *C. albicans* infections not only via their production of IL-17, as previously thought, but also to a significant extent, via their ability to produce IL-1 α . This revealed a new modality of T cell mediated anti-fungal host defense.

16. Can the authors comment on how NLRP3 activated in Th17 cells upon TCR signalling?

AR (author response): Calcium signaling induced by TCR activation, which is required for IL-1 α production (Fig. 4g, h), has previously been shown to activate the NLRP3 inflammasome (Lee GS *Nature* 2012). We comment on this in the discussion part of the current manuscript and in the summarizing graphical abstract (Extended data Fig. 22). In line with this, we provide a new experimental data set for the reviewer's perusal showing increased NLRP3 inflammasome activation by ASC speck formation in response to PMA/ionomycin restimulation (strong calcium flux) in human T cells (Fig. R3 NEW).

Response to reviewer #2:

(Remarks to the Author)

Chao and colleagues identify a role for NLRP3/Casp8/Casp3/GSDME regulating IL-1 α production by Th17 cells. They show that this is calpain processed IL-1 α and find it secreted by Th17 cells from patients with Schnitzlers syndrome. Overall the molecular pathway may be OK, although the link to Schnitzlers syndrome is tenuous, and some details seem lacking.

Major point 1:

It is concerning that the details for FACS analysis of intracellular IL-1 α are not provided. It states to refer reference 44, and in that paper cells were "restimulated for 5 hours with PMA and ionomycin in the presence of brefeldin A for the final 2.5 hours of culture". Is that indeed the case here, as the authors own data shows IL-1 α release is not impeded by BFA?

AR (author response): We have used the standard procedure for intracellular cytokine staining according to the technical data sheets from the companies selling the respective reagents and have therefore referred to published work. We have outlined the stimulatory conditions, which were correctly summarized above by the reviewer, in the legends and methods. We have now added more details from the referenced paper in the methods section. In short, presence or absence of BFA does not affect IL-1 α expression because IL-1 α is an unconventional cytokine, as shown in our data and correctly interpreted by the reviewer (Extended data Fig. 9a-c). We used BFA for intracellular cytokine staining as mentioned in the methods section because costainings with Th subset-defining cytokines (such as

IL-17, IFN- γ , and IL-4) were performed simultaneously to demonstrate coexpression, and such costainings required the use of BFA.

Major point 2:

Why does calpain inhibition or CAPN2 deletion prevent IL-1a secretion, when presumably the NLRP3/Casp8/Casp3/GSDME pathway is still functional for IL-1a release? Shouldn't it be released, just as full length unprocessed IL-1a? Perhaps the processing of IL-1a should be quantified in these experiments?

AR (author response): We showed that calpain leads to cleavage of pro-IL-1a into cleaved IL-1a, whereas the NLRP3 inflammasome–Casp8–Casp3–GSDME axis leads to extracellular release of this cleaved IL-1a via the GSDME pore. The NLRP3/Casp8/Casp3/GSDME pathway only enables the release of the cleaved but not full-length IL-1alpha, most likely due to pore size and charge effects as reported before for GSDMD by Xia et al. Nature 2021. We therefore proposed that pro-IL-1a is trapped intracellularly if cleavage by calpain is inhibited (Fig. 4d NEW).

As suggested by the reviewer, we have now provided additional evidence for this and quantified the processing of IL-1alpha upon calpain inhibition for the revised version of the manuscript. We indeed saw an increased accumulation of pro-IL-1a upon calpain inhibition intracellularly in addition to the reduced exit of IL-1 α into the supernatant, which is in line with our proposed mechanism (Fig. 4d NEW). Please also note our reply to reviewer 1 (5th point), who had a very similar comment.

Minor

point:

The legend of Figure 2 is lacking, what is inducing IL-1a in Th1/2 cells here, similar to the Th17 cells?

AR (author response): In Fig. 3a-e and h (previously in Fig.2), we used Th cell subsets isolated *ex vivo* (memory cells) without any inducing cytokines. They were isolated according to the differential expression of chemokine receptors, which enriched for the respective Th phenotypes (Acosta-Rodrigues et al Nat Immunol 2007, Zielinski et al. Nature 2012). Only Fig. 3f and g include inducing cytokines because naïve T cells were polarized into Th-cell subsets. We have provided more details in the legend to achieve better clarity in the revised manuscript.

Minor point: There is huge variation between how IL-1a ELISA data is reported: pg/ml or per 10⁶ cells or fold change. Negative controls are not provided in a lot of experiments.

AR (author response): We normalized to cell numbers if cell cultures were performed for longer time periods and with the addition of polarizing cytokines, which might have biased the proliferation of cells. In this way, we could accurately compare different conditions. All ELISA experiments included a medium control (no cells and unstimulated cells) which served as a negative control and showed undetectable IL-1alpha levels. We are happy to show this negative control, but we currently believe that this might rather confuse the reader if added to the main figure.

We have harmonized the reporting of the ELISA data throughout the manuscript as suggested by the reviewer. All raw data will also be provided in the reporting sheet (extended data).

Minor point:

Some claims are demonstrably false such as “GSDME expression has never been reported in T cells before.”. Please see Tixeira et al Front Imm 2018 and other papers.

AR (author response): We claim that GSDME has never been reported in T cells before because our diligent screen of the literature has not yielded any report of this. We have also not found any published report on TCR signaling-induced GSDME cleavage and IL-1 α secretion. Furthermore, we found no evidence of GSDME expression in T cells in the mentioned paper Tixeira et al. *Front Immunol*, as no primary T cells were tested in this manuscript (instead, the Jurkat cell line was used), or elsewhere. We have also tested our findings in Jurkat cells but do not see any evidence of IL-1 α secretion in Jurkat cells (highlighting major differences between the immortalized Jurkat cell line mentioned in the referenced paper and primary T cells, which are relevant in our IL-1 α context). These Jurkat data have been added for the reviewer’s perusal (Fig. R4). To better reflect this literature situation, we have rephrased the sentence that was pointed out by the reviewer the following way: “This was surprising considering that GSDME expression has never before been reported in *primary T cells*”.

Minor

4e spliced westerns.

point:

AR (author response): The bands come from the same blot, which we have indicated in the revised manuscript. All original blots will be provided.

Minor point:

Although IL-1 α appears to have some effect on IL-10 levels, the reciprocal effect was not determined, so should not feature in speculation about the existence of feedback loops.

AR (author response): We have also data showing that IL-10 suppresses IL-1 α expression levels and are happy to provide these data if wished. However, we referred to another feedback loop, and now realize that we might have depicted this in a confusing way (illustrated in Extended Data Fig. 22). We intended to clarify the sustained suppression of IL-10 in Th17 cells following naïve Th17 cell priming with IL-1 β , which has remained enigmatic so far. This sustained IL-10 suppression is achieved following IL-1 β induced autocrine IL-1 α production, which feeds back via IL1R to continuously suppress IL-10. Therefore, depletion of IL-1 α from the supernatant restores IL-10 expression by Th17 cells (Extended Data Fig. 4). We have de-emphasized this finding in the manuscript by removing it from Fig. 7 and shifting it into the Extended Data. We also removed this aspect from the discussion to give more space to new exciting findings about the TCR specificities of IL-1 α ⁺ Th17 cells and about anti-fungal host defense.

Minor point:

Some Schnitzler syndrome patients have somatic mutations in NLRP3, was that the case for the T-cell clones studied here?

AR (author response): Activating mutations in NLRP3 in Schnitzler syndrome are very rare and genetic screening for research and publication reasons was unfortunately not allowed due to ethical restrictions for this study. To avoid too strong expectations that defects in the whole NLRP3-casp8-casp3-GSDME-IL-1alpha pathway can be recapitulated with Schnitzler syndrome patients, we followed the editor's recommendation to deemphasize this data. We have removed the Schnitzler Syndrome data from the last figure and instead present new data with JIA patients in the context of pro-inflammatory IL-1 α ⁺ Th17 cell functions before the mechanism of IL-1 α release is presented (Fig. 2 NEW).

Minor point:

I am not sure that IL-1a could be considered "the etiological disease-defining cytokine" in Schnitzler syndrome.

AR (author response): We agree that it would have been better to rephrase this sentence given the complexity of the disease. For reasons, outlined in the previous responses we have completely removed the Schnitzler syndrome data to deemphasize any claims about the pathogenesis of this complex and rare disease.

Instead, we provide new data about the physiological and clinical role of IL-1 α -producing Th17 cells. We added new data on the TCR specificities of IL-1 α producing Th17 cells (Fig. 8 NEW). We also demonstrate that T cell derived IL-1 α is relevant for *C. albicans* phagocytosis (Fig. 8 NEW). Furthermore, we have included new data and on juvenile idiopathic arthritis (JIA) (Fig. 2 NEW).

Reviewer #3

(Remarks to the Author)

Review Zielinski 2022 NI

In the manuscript submitted by Chao et al. "Human Th17 cells engage gasdermin E pores to release IL-1a upon NLRP3inflammasome activation" the authors describe the expression of IL-1a by human Th17 cells. In the Th17 cells, the authors found IL-1a expression to coincide with cell viability and proliferation. Furthermore, the authors found this process to be regulated by the NLRP3 inflammasome and gasdermin E pores and the autocrine IL-1a signaling in Th17 cells to suppress IL-10 and induce a pathogenic phenotype. IL-1a is a critical regulator of Th17 differentiation and the paper shows that IL-1 produced by differentiating Th17 cells plays a critical role in their growth and differentiation. The manuscript is well written and describes an interesting novel role of the NLRP3 inflammasome and IL-1a secretion in Th17 cell biology. However, the manuscript would improve from better use of the single-cell data. In order to put the observation in context and the importance of IL-1 in T cell differentiation, it would be also important to put more emphasis on the physiological role of IL-1a secretion by Th17 cells.

AR (author response): We thank the reviewer for the positive feedback. We agree that we have not extensively explored the available single-cell transcriptomic data, but kept the analysis confined to hypothesis-driven questions and only used the scRNAseq data as a starting point (unbiased discovery of IL1A in Th17 cells) for subsequent in-depth mechanistic dissection of IL-1a regulation. In accordance to the reviewers' suggestions, we have now provided a more

extensive unbiased analysis of the scRNA-seq data (see below). We have also provided new data on the physiological role of IL-1a secretion by Th17 cells for anti-fungal host defense (see below).

Major comments:

1. It would be important to compare the level of IL-1a expression by Th17 cells to multiple other cell types that are bona fide producers of IL-1a (macrophages, DCs, neutrophils, B cells...), in Figure 2a. It is important to define how the level of IL-1a expression compares to cells known to express IL-1a in which use Gasdermin D is activated for IL-1 release. Is it at a similar level and can therefore be expected to have a paracrine biological function or is it a low level in the majority of T cells, which may act as an autocrine growth without induction of much cell death? Or is cell death always associated with IL-1 release from T cells? There are emerging data showing that IL-1a can also be secreted by DCs without pyroptosis, resulting in the generation of hyperactive DCs.

AR (author response): We absolutely agree that it is important to relate IL-1 α production levels in T cells to those in other cell types. We have stressed the high level of T cell derived IL-1alpha production by comparison to monocytes stimulated with LPS and nigericin, which served as a positive control for very high IL-1alpha secretion. Comparable levels were reached (Fig. 3a), demonstrating that T cells represent a relevant source of IL-1alpha. We have now also addressed the reviewer's request to compare IL1A expression by Th17 cells to multiple other cell types. We therefore performed transcriptomic scRNA-seq analyses of multiple public PBMC data sets and interrogated IL1A gene expression levels following cell type annotation by marker genes (Extended Data Figure 2 NEW). Interestingly, no IL1A expression was observed in resting blood immune cells without stimulation, stressing the importance of cell type-specific stimulation (e.g., LPS for monocytes, TCR stimulation for T cells, etc.). We have added representative data to the new version of the manuscript (Extended Data Fig. 2a,b New). Furthermore, we have provided new scRNAseq data analysis showing IL1A expression by LPS stimulated monocytes (Extended Data Fig. 2c-e NEW) and performed differential expression and co-expression analyses, which led to the conclusion that IL1A is differentially regulated by different immune cell types and upon different cell type specific stimuli (Fig. 1D New).

Regarding cell death: It is known from previously published reports that bioactive IL-1alpha can be released by dying cells (full-length form). We have now shown a molecular pathway leading to cleaved IL-1a exit via pores in the absence of pyroptosis. It is indeed possible that the release of IL-1alpha further promotes survival and growth in an autocrine manner, as implied by the reviewer, because the new data in Fig. 7b-I NEW do not only exclude cell death in the setting of membrane pore formation but even show increased proliferation. We have therefore performed more transcriptomic analyses of IL-1 α^+ vs. IL-1 α^- T cells (and GSDME $^+$ and GSDME $^-$ T cells) and confirmed the higher viability, proliferation and activation in IL-1 α^+ T cells. The new data have been included in the revised manuscript (Fig. 7 NEW).

2. In Figure 6 to show that the IL-1a expression by Th17 cells themselves is important for their function, CRISPR-CAS9 knockout of IL-1a should be performed in the Th17 cells to study the role of IL-1 in cell-intrinsic production leading to Th17 differentiation.

AR (author response): To deemphasize the topic of autocrine IL-1 α production on Th17 cell pathogenicity (IL-10 suppression), we have removed the data from the final figure and moved it to Extended Data Fig. 4, where it serves to support the pro-inflammatory identity of the IL-1 α producing subset of Th17 cells.

Role of cell-intrinsic production of IL-1 α on Th17 cells: We inhibited autocrine IL-1 α signaling with a specific IL-1 α -blocking antibody. This was done in fully differentiated memory Th17 cells to demonstrate the impact on their anti-inflammatory IL-10 production. Theoretically, this could easily be repeated in CRISPR-Cas9 KO Th17 cells. However, we believe that the neutralizing antibody was quite efficient and showed significant effects. We have not addressed the role of IL-1 α in Th17 cell differentiation (from naïve T-cell precursors) because naïve T cells do not express IL-1 α (only inducible upon TCR stimulation in the presence of the polarizing cytokines IL-1b+TGFb, previous Fig. 2f,g, now Fig.3f,g). “Wild-type” naïve T cells without any CRISPR KO of IL1A also do not differentiate into Th17 cells but require the external source of the polarizing cytokine IL-1 (either IL-1a or IL-1b via shared IL-1R) plus TGFb, which is why a CRISPR-Cas9 KO of IL1A would not be able to show a role in Th17 cell differentiation. This can be seen in Fig. 3f and Fig. 3g. We can therefore rule out that cell intrinsic production of IL-1a can lead to Th17 cell differentiation.

3. What is the physiological function of IL-1a secretion by Th17 cells and does IL-1 production by T cells have a role during infections or autoimmune diseases? Any evidence for that? The Schnitzel syndrome seems generally driven by IL-1a-expressing cells and that Th17 cells from these patients also express IL-1a is not entirely surprising and the fact that IL-1a blockade decreases IL-1a by Th17 cells is also expected. In the Schnitzel syndrome, how do the levels expressed by Th17 cells compare to other immune cell populations that express IL-1a? Can the IL-1a produced by Th17 cells be expected to contribute to the disease? Is there any evidence for IL-1a expression by Th17 cells in any other conditions, including infections or in response to commensal bacteria?

AR (author response): “Infections”&“commensal bacteria”: We have generated new data in the context of infections or commensal microbes, which we have now added to the revised version of the manuscript. We can show that *C. albicans*-specific Th17 cells have significantly higher IL-1 α expression than *S. aureus*-specific Th17 cells. This is consistent with previous reports, which show that *C. albicans*-specific Th17 cells are proinflammatory (IL-10 negative) and that *S. aureus*-specific Th17 cells are anti-inflammatory (IL-10 positive) and with reports showing that *C. albicans*-but not *S. aureus*-specific Th17 cells require IL-1 β for their generation (the IL-1 α -inducing cytokine) (Noster et al. *J Allergy Clin Immunol* 2016; Zielinski et al. *Nature* 2012) (Fig. 8 a,b NEW). Furthermore, we have added new data (FACS-based phagocytosis assays and real time imaging videos) showing that T cell derived IL-1 α promotes *C. albicans* phagocytosis/killing (Fig. 8 c,d NEW). Therefore, Th17 cells exert their anti-fungal effector function not only via IL-17, as previously thought, but, unexpectedly, also to a significant extent via IL-1 α .

“IL-1 α in disease”: We provide new data (transcriptomic and functional) highlighting the proinflammatory identity of the IL-1 α ⁺ subset of Th17 cells (Fig. 2 NEW). In particular, we provide new data showing significantly increased IL-1 α expression by Th17 cells from the blood in patients with juvenile idiopathic arthritis (JIA). We also show very high frequencies of IL-1 α -producing Th17 cells in the synovial fluid of these patients. This is the first report on IL-1 α -producing T cells in a human disease. In the previous version of the manuscript, we have highlighted how IL-1 β -induced autocrine IL-1 α leads to sustained suppression of IL-10 in Th17 cells, because it had remained enigmatic how the IL-1 β priming signal achieved long-term IL-10 suppression. We have now de-emphasized this pathway and moved it to Extended Data Fig. 4, where it serves to support the proinflammatory nature of IL-1 α ⁺ Th17 cells.

4. The single-cell data should be analyzed more deeply to develop a nuanced understanding of IL-1 production by T cells. What are the genes co-expressed with IL-1a (Figure 1a)? What are the differentially expressed genes in cluster 1? What does the expression of gasdermin E, NLRP3, and caspases look like on the UMAP, even though the process is supposed to be driven at the protein level but does the expression of inflammasome-specific genes change and co-

expressed and co-regulated in T cells as has been observed in DCs and myeloid cells? A comparison of the gene expression of IL-1 production in DCs vs. T cells will give additional information about the common drivers and differences between the two cell types, which will give further mechanistic information about the mechanisms by which IL-1 is produced by T cells. Are they specifically up-regulated in cluster 1?

AR (author response):

As suggested by the reviewer, we have added new data showing the top 20 upregulated genes in individual Leiden clusters (0-5) (Extended Data Fig. 1 NEW). We have also shown the genes coexpressed with IL1A (Extended Data Fig. 3a) as requested. We have plotted a network with the genes, which are directly or indirectly coexpressed with IL1A in Th17 cells (Extended Data Fig. 3b). Furthermore, an enrichment analysis for IL1A coexpressing genes was performed (Extended Data Fig. 3c). As further suggested by the reviewer, we have also performed the UMAP analysis for the genes proposed by the reviewer (Fig. R5). We have furthermore, in accordance with the reviewer's suggestion, performed a comparison of the IL1A gene expression of Th17 cells and myeloid cells to assess differences or common drivers between Th17 cells and monocytes. While resting monocytes did not reveal IL1A expression (UMAP, Extended Data Fig. 2 NEW), LPS stimulation leads to IL1A expression. Interestingly, a direct comparison with human Th17 cells revealed almost no overlap in the regulation of IL1A in monocytes versus Th17 cells, as also corroborated experimentally throughout our study (Figure 1d NEW). This highlights the uniqueness of the pathway that we have described for human T cells.

We have therefore provided deeper bioinformatic single-cell analyses on IL-1 α production as proposed by the reviewer. Since the process of IL-1 α secretion is regulated at the protein level in line with the reviewer's interpretation (NLRP3 inflammasome activation, enzymatic caspase cleavage cascade, GSDME pore formation), we used the exploratory scRNAseq data to highlight the discovery of IL1A as a Th17 cell property, but deemphasized the transcriptomic analyses for the mechanistic exploration of IL1A secretion.

More bioinformatic scRNAseq analyses can be shown, if requested (Fig. R6 NEW), to exploit the Th17 cell scRNAseq data set, which we also plan to deposit publicly. We also performed a weighted gene correlation analysis using all genes actively expressed in the IL1A⁺ cells. Subsequently, we used the WGCNA algorithm to identify clusters of highly correlated genes. In total, 149 modules were identified, of which one contained IL1A and 18 other genes. The IL1A-containing modules consisted of annotated genes (PTDSS2, CALCB, SUSP1, TCEAL4, FAM102A, PLAAT2, RXRG, CCDC74B, INHBE) and lncRNA (CACNA1C-AS1, THAP7-AS1, ATP6V0E2-AS1, FAM66D) and the nonannotated transcripts (AC092617.1, AL645933.2, AC078845.1, AC145343.1, AC068473.5) (Fig. R6a). From the annotated genes, the ones with the strongest correlation to IL1A were PTDSS2 (positive), RXRG (positive), INHBE (positive) and FAM66D (positive).

Additionally, we performed a differential gene correlation analysis (DGCA) between the IL1A⁺ and IL1A⁻ Th17 cells in order to investigate the global regulatory effect of the IL1A expression. From this global network we retrieved 8 genes of interest according to our experimental findings (Th17-cell identity, molecules involved in IL1a regulation), and we visualized the differences in their correlations with other genes between the IL1A⁺ and IL1A⁻ Th17 cells. Interestingly, 737 positive correlations involving the 8 genes of interest present in the IL1A⁺ cells were lost in the IL1A⁻ cells (Fig. R6b, +/-). In addition, the strength of 596 positive correlations in the IL1A⁺ cells that involved the 8 genes of interest was different in the IL1A⁻ cells (Figure R6b, +/-). Our findings therefore indicate a strong change in the expression pattern of those important genes between IL1A⁺ and IL1A⁻ Th17 cells.

Minor

comment:

1. The introduction is too short. Inflammasomes, NLRP3, and IL-1a in general and in the context of T cells are not being introduced. The manuscript would improve if the authors would provide more background and explain the importance of the studied molecules in immune-mediated diseases, for a general reader.

AR (author response): We provided more information in the introduction.

Minor comment:

2. Does cluster 1 (Figure 1C) express a higher proliferation/survival signature that would support the proposed hypothesis that IL-1 drives the proliferation and expansion of Th17 cells? In addition, what are generally the DE genes of the different clusters? The heterogeneity of human Th17 cells by scRNAseq would be of general interest to the reader.

AR (author response): As suggested by the reviewer, we tested the proliferation/survival signatures and did indeed see the expected enrichment of the proliferation/survival signature in cluster 1 (Fig. 7h NEW). This finding is in line with our transcriptomic results upon comparison of IL-1 α ⁺ and IL-1 α ⁻ Th17-cell clones. Of note, this finding supports the idea that IL1alpha-producing cells are proliferating/surviving despite GSDME pore formation, as we have also experimentally demonstrated (Fig. 7c NEW, d NEW, e NEW, f). As suggested by the reviewer, we now also provide the DEGs for the different clusters (Extended data Fig. 1 NEW) and agree that information about the heterogeneity within the human Th17-cell subset is of high interest and indeed a resource for research on more interesting Th17-cell properties and regulatory mechanisms. We have for this reason also provided results of more bioinformatic analyses on the different Th17-cell clusters including enrichment analyses with their DEGs, network analyses for IL1A expression and GSEA for further characterization of the Th17-cell heterogeneity. The raw sequencing data will also be provided and will be made publicly available. Of note, we do not address the question of how IL-1 drives proliferation and expansion of Th17 cells as this has been addressed earlier (work by W. Paul and C. Pasare, i.e. Shlomo Z, PNAS 2009). Instead, we highlight that in T cells pore formation is not associated with pyroptotic cells death in contrast to other cell types. We hypothesize the ESCRT machinery to be involved in pore repair as outlined in the discussion.

In summary, the manuscript describes an interesting and novel finding that human Th17 cells express IL-1a and also partly addresses the mechanism of IL-1a secretion by T cells. The physiological role of IL-1a in human Th17 cells (in health or disease), however, remains unclear and how often is this mechanism used by T cells for differentiation and function. More analysis and results in that direction would significantly improve the impact and provide a mechanistic understanding of the role of IL-1 produced by T cells in Th17 differentiation and induction of tissue inflammation.

AR (author response): We thank the reviewer for the positive feedback summary and the great suggestions to improve the translational value of the manuscript. The physiological role of IL-1 α in human Th17 cells has now been addressed with new data in the revised version of the manuscript (Fig. 8 NEW). We have discovered IL-1 α as a so far overlooked effector mechanism of Th17 cells for the purpose of anti-fungal host defense (TCR specificity, *C. albicans* phagocytosis/killing). Furthermore, we have clarified that autocrine IL-1 α does not contribute to Th17 cell differentiation (as mentioned above) but rather to Th17 cell pathogenicity (Fig. 2 NEW). Also new clinical data with JIA patients has been included (Fig. 2g) demonstrating significantly increased IL-1 α expression by circulating Th17 cells from JIA

patients compared to healthy controls. These patients also contained high frequencies of IL-1 α producing T cells in the inflamed synovial fluid.

Decision Letter, first revision:

Subject: Your manuscript, NI-A33904A

Message: Our ref: NI-A33904A

25th Oct 2022

Dear Dr. Zielinski,

Thank you for your patience as we've prepared the guidelines for final submission of your Nature Immunology manuscript, "Human Th17 cells engage gasdermin E pores to release IL-1a upon NLRP3 inflammasome activation" (NI-A33904A). Please carefully follow the step-by-step instructions provided in the attached file, and add a response in each row of the table to indicate the changes that you have made. Please also check and comment on any additional marked-up edits we have proposed within the text. Ensuring that each point is addressed will help to ensure that your revised manuscript can be swiftly handed over to our production team.

When you upload your final materials, please include a point-by-point response to any remaining reviewer comments and please make sure to upload your checklist.

If you have not done so already, please alert us to any related manuscripts from your group that are under consideration or in press at other journals, or are being written up for submission to other journals (see: <https://www.nature.com/nature-portfolio/editorial-policies/plagiarism#policy-on-duplicate-publication> for details).

In recognition of the time and expertise our reviewers provide to Nature Immunology's editorial process, we would like to formally acknowledge their contribution to the external peer review of your manuscript entitled "Human Th17 cells engage gasdermin E pores to release IL-1a upon NLRP3 inflammasome activation". For those reviewers who give their assent, we will be publishing their names alongside the published article.

Nature Immunology offers a Transparent Peer Review option for new original research manuscripts submitted after December 1st, 2019. As part of this initiative, we encourage our authors to support increased transparency into the peer review process by agreeing to have the reviewer comments, author rebuttal letters, and editorial decision letters published as a Supplementary item. When you submit your final files please clearly state in your cover letter whether or not you would like to participate in this initiative. Please note that failure to state your preference will result in delays in accepting your manuscript for publication.

Cover suggestions

As you prepare your final files we encourage you to consider whether you have any images or illustrations that may be appropriate for use on the cover of Nature Immunology.

Nature Immunology has now transitioned to a unified Rights Collection system which will allow our Author Services team to quickly and easily collect the rights and permissions required to publish your work. Approximately 10 days after your paper is formally accepted, you will receive an email in providing you with a link to complete the grant of rights. If your paper is eligible for Open Access, our Author Services team will also be in touch regarding any additional information that may be required to arrange payment for your article.

Please note that *Nature Immunology* is a Transformative Journal (TJ). Authors may publish their research with us through the traditional subscription access route or make their paper immediately open access through payment of an article-processing charge (APC). Authors will not be required to make a final decision about access to their article until it has been accepted. [Find out more about Transformative Journals](https://www.springernature.com/gp/open-research/transformative-journals).

If you have any questions about costs, Open Access requirements, or our legal forms, please contact ASJournals@springernature.com.

Authors may need to take specific actions to achieve [compliance](https://www.springernature.com/gp/open-research/funding/policy-compliance-faqs) with funder and institutional open access mandates. If your research is supported by a funder that requires immediate open access (e.g. according to [Plan S principles](https://www.springernature.com/gp/open-research/plan-s-compliance)) then you should select the gold OA route, and we will direct you to the compliant route where possible. For authors selecting the subscription publication route, the journal's standard licensing terms will need to be accepted, including [self-archiving policies](https://www.springernature.com/gp/open-research/policies/journal-policies). Those licensing terms will supersede any other

terms that the author or any third party may assert apply to any version of the manuscript.

Please use the following link for uploading these materials: [REDACTED]

Best regards,

Elle Morris
Senior Editorial Assistant
Nature Immunology
Phone: 212 726 9207
Fax: 212 696 9752
E-mail: immunology@us.nature.com

On behalf of

Nick Bernard, PhD
Senior Editor
Nature Immunology

Reviewer #1:

Remarks to the Author:

The authors have rigorously addressed all my concerns with a large amount of (convincing) new data and/or explanations. I'm sure that their findings will be well received by others in the field and will provide a solid foundation to be built upon further.

Reviewer #2:

Remarks to the Author:

My comments have been addressed

Reviewer #3:

Remarks to the Author:

In the revised paper, the authors added more detailed analysis of the scRNA-seq data and addressed the physiological role of IL-1a secretion by Th17 cells in the host defense against fungi. In addition, the authors compared the Th17 IL-1a levels to other immune cell populations. Thereby, the authors responded sufficiently to our requests/comments.

Final Decision Letter:

Subject: Decision on Nature Immunology submission NI-A33904B

Message: In reply please quote: NI-A33904B

Dear Dr. Zielinski,

I am delighted to accept your manuscript entitled "Human Th17 cells engage gasdermin E pores to release IL-1a upon NLRP3 inflammasome activation" for publication in an upcoming issue of Nature Immunology.

Over the next few weeks, your paper will be copyedited to ensure that it conforms to Nature Immunology style. Once your paper is typeset, you will receive an email with a link to choose the appropriate publishing options for your paper and our Author Services team will be in touch regarding any additional information that may be required.

Please note that *Nature Immunology* is a Transformative Journal (TJ). Authors may publish their research with us through the traditional subscription access route or make their paper immediately open access through payment of an article-processing charge (APC). Authors will not be required to make a final decision about access to their article until it has been accepted. [Find out more about Transformative Journals](https://www.springernature.com/gp/open-research/transformative-journals).

Authors may need to take specific actions to achieve [compliance](https://www.springernature.com/gp/open-research/funding/policy-compliance-faqs) with funder and institutional open access mandates. If your research is supported by a funder that requires immediate open access (e.g. according to [Plan S principles](https://www.springernature.com/gp/open-research/plan-s-compliance)) then you should select the gold OA route, and we will direct you to the compliant route where possible. For authors selecting the subscription publication route, the journal's standard licensing terms will need to be accepted, including [self-archiving policies](https://www.springernature.com/gp/open-research/policies/journal-policies). Those licensing terms will supersede any other terms that the author or any third party may assert apply to any version of the

manuscript.

Your paper will be published online soon after we receive your corrections and will appear in print in the next available issue. Content is published online weekly on Mondays and Thursdays, and the embargo is set at 16:00 London time (GMT)/11:00 am US Eastern time (EST) on the day of publication. Now is the time to inform your Public Relations or Press Office about your paper, as they might be interested in promoting its publication. This will allow them time to prepare an accurate and satisfactory press release. Include your manuscript tracking number (NI-A33904B) and the name of the journal, which they will need when they contact our office.

About one week before your paper is published online, we shall be distributing a press release to news organizations worldwide, which may very well include details of your work. We are happy for your institution or funding agency to prepare its own press release, but it must mention the embargo date and Nature Immunology. Our Press Office will contact you closer to the time of publication, but if you or your Press Office have any enquiries in the meantime, please contact press@nature.com.

Also, if you have any spectacular or outstanding figures or graphics associated with your manuscript - though not necessarily included with your submission - we'd be delighted to consider them as candidates for our cover. Simply send an electronic version (accompanied by a hard copy) to us with a possible cover caption enclosed.

If you have not already done so, we strongly recommend that you upload the step-by-step protocols used in this manuscript to the Protocol Exchange. Protocol Exchange is an open online resource that allows researchers to share their detailed experimental know-how. All uploaded protocols are made freely available, assigned DOIs for ease of citation and fully searchable through nature.com. Protocols can be linked to any publications in which they are used and will be linked to from your article. You can also establish a dedicated page to collect all your lab Protocols. By uploading your Protocols to Protocol Exchange, you are enabling researchers to more readily reproduce or adapt the methodology you use, as well as increasing the visibility of your protocols and papers. Upload your Protocols at www.nature.com/protocolexchange/. Further information can be found at www.nature.com/protocolexchange/about .

Please note that we encourage the authors to self-archive their manuscript (the accepted version before copy editing) in their institutional repository, and in their funders' archives, six months after publication. Nature Portfolio recognizes the efforts of funding bodies to increase access of the research they fund, and strongly encourages authors to participate in such efforts. For information about our editorial policy, including license agreement and author copyright, please visit www.nature.com/ni/about/ed_policies/index.html

Sincerely,

Nick Bernard, PhD
Senior Editor
Nature Immunology